# Grain size dynamics using a new planform model. Part 1: GravelScape description and validation

Amanda Lily Wild[1,2], Jean Braun[1,2], Alexander C Whittaker[3], and Sebastien Castelltort[4]

[1] Helmholtz Center Potsdam, GFZ German Center for Geoscience, Potsdam, Germany
[2] The Institute of Geosciences, Universität Potsdam, Potsdam, Germany
[3] Department of Earth Science and Engineering, Royal School of Mines, Imperial College London
[4] Department of Earth Sciences, University of Geneva, Rue des Maraîchers 13, 1205 Genève, Switzerland
[1] **Correspondence:** Amanda Lily Wild (awild@gfz-potsdam.de)

**Abstract.** The grain size preserved within the stratigraphic record over thousands to millions of years has several important applications. In particular, it can serve as a record of significant climatic, eustatic, or tectonic events. Here we present a new model for grain size fining predictions that combines a landscape evolution model based on the Stream Power Law but modified for sedimentation (Yuan et al., 2019) with an extension of the self-similar grain size model (Fedele and Paola, 2007). The new model, which we called GravelScape, includes the effects on grain size fining of lateral heterogeneities in deposition rate caused by dynamically evolving channels. We show that, when multi-channel dynamics (i.e. avulsions) are prevented, by reducing the planform model to a single downstream dimension, our new model can reproduce results obtained by other methods that assume that fining is controlled by subsidence only. We demonstrate that including across-basin (two-dimensional) effects can lead to deviations from previous subsidence predictions for grain size fining. The magnitude of these deviations correlates with the extent of sediment bypass and the configuration of surface topography, both of which influence the amplitude of across-basin variability within the sedimentary system.

## 1 Introduction

Grain size trends in the stratigraphic record are widely used as an important source of information about past and present depositional environments. As postulated by Allen et al. (2013) and Duller et al. (2010), the ratio of sediment flux to accommodation not only controls basin filling, deposition and fan development, but also strongly influences downstream fining for gravel and sand coarse fractions. When used within their geologic context, variations in grain sizes can be used to identify changes in past environmental conditions that have been recorded in the stratigraphic record (Armitage et al., 2011; Rice, 1999). For example, increasing precipitation within the source catchment results in a lateral shift in where coarser grains are deposited and a lengthening of the fan area in response to the accompanying increase in sediment flux (Armitage et al., 2011). Hooke (1968) suggests that higher tectonic uplift and tilting within the Panamint Range produce higher sediment flux in the western part of Death Valley leading to more extensive coarse grained alluvial fans relative to the eastern Black Mountains area. As postulated by Duller et al. (2010) and observed by Dingle et al. (2016) within the Ganga Plain, grain size fining rate is more rapid in short, rapidly subsiding systems. Multiple further examples (i.e. Harries et al. (2018); Paola et al. (1992); Whittaker et al. (2011);

Brooke et al. (2018)) highlight how a thorough understanding of grain size fining in response to external forcing has broad implications for interpreting the sedimentary record to unravel tectonic conditions and climates of the geological past.

There are a number of internal dynamical processes that are also likely to impact the coarse grain size record depending on sediment flux, discharge, subsidence rate (or the creation of accommodation space), and basin geometries (Sømme et al., 2009; Scheingross et al., 2020; Romans et al., 2016; Hajek and Straub, 2017). These internal or autogenic processes, such as basin reorganization, sediment waves, channel avulsions or knickpoint generation (Scheingross et al., 2020) arise due to strong feedbacks between topography, erosion, and sediment transport that are independent of external perturbations. Their impact on grain size fining have been, however, largely excluded from previous theoretical and modelling studies that predict grain size fining.

Modelling studies (Veldkamp et al., 2017; Carretier et al., 2016; Armitage et al., 2011; Davy and Lague, 2009; Paola and Voller, 2005) commonly predict how external forcings control the grain size in the stratigraphic record, based on reduced complexity approaches. This is because hydraulic processes controlling grain size as observed in modern systems are often too complex to model over millions of years and their characteristics are, for the most part, not preserved within the geologic record. For example, Fedele and Paola (2007) have developed a reduced complexity model of depositionally-controlled grain size fining using concepts derived from the stratigraphic record, such as time-integrated grain size self-similarity. The self-similar grain size fining model assumes that downstream deposition is the main control on the fining rate (Fedele and Paola, 2007) and, thus, does not consider feedbacks between grain size and topography.

Many past grain size modelling approaches for the stratigraphic record (Veldkamp et al., 2017; Carretier et al., 2016; Armitage et al., 2011; Davy and Lague, 2009; Paola and Voller, 2005) have in common that they consider fining along a single main channel transporting sediment from source to sink. Consequently, they do not address the complexity of most natural sedimentary systems, where transport and deposition occur along several channels that are characterised by a very dynamic behaviour that is likely to influence grain size fining and the way it is stored in the stratigraphic record.

This is the main motivation for the new method that we developed and present here that generalises Fedele and Paola (2007)'s self-similar gravel grain size model into multiple dimensions (downstream, across the basin, and overtime/depth) using a planform landscape evolution model, FastScape (Braun and Willett, 2013) that allows for erosion and deposition (Yuan et al., 2019) following the approach developed by Davy and Lague (2009). Because the model can represent deposition and erosion in two dimensions, i.e., along several dynamic channels, it can update topography at each time step, and because it does not rely on the common assumption that deposition rate must be equal to subsidence rate (Duller et al., 2010; Whittaker et al., 2011), it should allow us to investigate the impact of external forcings and internal processes on grain size fining and therefore help us to better understand the sedimentary record.

In this manuscript, we first provide a short description of Fedele and Paola (2007)'s model for self-similar grain size fining and of the landscape evolution model (FastScape) that we have used. We then describe how we have combined the two to propose a new, plan-form model for self-similar grain size fining, which we called GravelScape (see supplementary video for an example of the model). We describe its basic behaviour and compare its results to the sediment fining curves obtained by Duller et al. (2010) along a single channel, assuming an exponential shaped subsidence function simulating a flexural basin and

a constant sediment flux at steady-state into the basin. We will use these results to validate our numerical approach but also to demonstrate that differences are likely to be driven by autogenic processes that can only be modelled with a two-dimensional, plan-form approach.

This paper, (Wild et al., 2024a), is the first of a series of three that report our recent work on modelling grain size fining in two-dimensional sedimentary systems. As explained above, the first focuses on the description of the multi-channel, LEM-coupled grain size model, its general behaviour at steady-state, and its validation by comparison with Duller et al. (2010)'s single channel work. The second paper (Wild et al., 2024b) will be devoted to studying autogenic processes and how they affect preserved grain size under many different basin set-ups at steady-state. The third paper (Wild et al., 2024c) expands beyond the steady-state hypothesis to focus on foreland basin evolution and how autogenic dynamics and the resulting grain size fining are likely to behave and are stored in the stratigraphic record at various phases of basin evolution.

## 2 Previous work on which our model is based

### 2.1 Fedele and Paola (2007)'s self-similar grain size model

The underlying approach for our integration of grain size fining into a landscape evolution model is based on Fedele and Paola (2007)'s grain size solution ($D$) as a function of dimensionless distance downstream ($x^*$) within a depositional area for gravel:

$$D(x^*) = \overline{D_0} + \phi_0 \frac{C_2}{C_1} e^{-C_1 y^*} - 1 \tag{1}$$

and sand:

$$D(x^*) = \overline{D_0} e^{-C_3 y^*}. \tag{2}$$

where $x^*$ is the distance along the river profile, $x$, normalized by its total length ($x^* = x/L$). $\overline{D_0}$ and $\phi_0$ are the initial mean grain size and standard deviation at the source, respectively. $C_1, C_2$, and $C_3$ are constants that represent the change in mean grain size and standard deviation downstream. $C_v$ is defined as the ratio of the downstream change in the standard deviation relative to mean grain size, i.e., $C_v = C_1/C_2$, and referred to as the coefficient of variation. $C_1$ ranges between 0.5 and 0.9, $C_3$ between 0.1 and 0.45 and $C_v$ between 0.7 and 0.9 (Fedele and Paola, 2007). Whittaker et al. (2011) note that a $C_v$ value of 0.8 is commonly observed. Note that, to avoid repetition, this work will focus on the application of the gravel equation (1), but all methods described further below could be modified and applied to solving the sand equation (2).

As defined in Fedele and Paola (2007), $y^*$ is the dimensionless distance transformation given by:

$$y^*(x^*) = \int_0^{x^*} R^*(x^*) \, dx^* \tag{3}$$

where $R^*$ is the dimensionless downstream distribution of deposition defined as:

$$R^* = (1 - \gamma)RL. \tag{4}$$

where $\gamma$ is the porosity of the sediment deposit, and $R$ is the ratio of deposition rate, $r$, to sediment flux, $q_s$, i.e., $R = r/q_s$. At each $x^*$ point along the river profile, $R^*$ relates the sediment deposition into the substrate to the sediment flux in the river, taking into account a given length and porosity. Finally, deposition rate, $r$, is the sum of the rate of change of elevation of the landscape and the subsidence rate, $\sigma$, i.e., $r = dh/dt + \sigma$.

In non-mathematical terms, Fedele and Paola (2007)'s approach postulates that grain size fining over long time scales is controlled by 1) the source sediment supply and grain size distribution, 2) the deposition rate throughout the system length, and 3) the hydraulic mobility of different grain size types (gravel vs. sand) assuming a constant shield stress at the bed, 4) self-similarity between the subsurface and surface of the distribution of gravel grain size clasts, and 5) a mass balance of the transport, substrate, and an active layer (Fedele and Paola, 2007). We refer the reader to (Fedele and Paola, 2007) for a more detailed explanation of the assumptions behind this model and how it has been derived from these assumptions. However, the self similarity of gravel grain sizes has now been repeatedly demonstrated from field data (D'Arcy et al., 2017; Harries et al., 2018; Brooke et al., 2018) and this model has been successfully applied to stratigraphic examples (Whittaker et al., 2011; Duller et al., 2010; Garefalakis et al., 2024).

## 2.2 The Landscape Evolution Model (LEM)

We use FastScape (Bovy, 2021) as a basic Landscape Evolution Model (LEM) that solves the Stream Power Law (SPL) in two dimensions following the efficient algorithm described in Braun and Willett (2013). The SPL states that the rate of bedrock elevation change is the sum of uplift (or subsidence) rate $U$ and erosion rate assumed proportional to upstream drainage area, $A$, used as a proxy for discharge, and local slope $S$ (Whipple and Tucker, 1999) :

$$\frac{dh}{dt} = U - KA^m S^n \tag{5}$$

$K$ is a rate coefficient or erosivity that depends mostly upon lithology and precipitation rate. $m$ and $n$ are exponents that generally maintain a ratio of $m/n \approx 0.5$. Sediment transport and deposition are incorporated into FastScape by using Yuan, Braun, Guerit, Rouby, and Cordonnier (2019)'s implementation of Davy and Lague (2009)'s $\xi - q$'s algorithm, which states that the rate of sediment deposition is proportional to sediment flux and inversely proportional to upstream drainage area. This leads to the following evolution equation that integrates the processes of uplift/subsidence, erosion, transport, and deposition into a simple framework:

$$\frac{dh}{dt} = U - K\tilde{p}^m A^m \left(\frac{dh}{ds}\right)^n + \frac{G}{\tilde{p}A} \int_A (U - \frac{dh}{dt}) dA \tag{6}$$

In this formulation, $G$ is a dimensionless deposition coefficient and $\tilde{p}$ represents spatial or temporal variations in precipitation rate with respect to a mean value that is contained in both $K$ and $G$. In this work, we will vary topography through $G$ and discuss its impact on model results. $G$ controls the transport- or detachment- limited nature of the depositional system (Yuan et al., 2019). Guerit et al. (2019) have demonstrated that many natural river systems tend to be on the transport limited side. Note that, for numerical reasons, $G = 1$ is the maximum transported-limited value computed efficiently and reliably by FastScape

(Yuan et al., 2019), although larger values can be used but may require multiple iterations and reduce efficiency. Reducing $G$ reduces the transport limited nature of the system and moves into more of a detachment limited set-up.

In FastScape, in order to compute the transport of water and sediment, nodes are ordered along flow paths through the landscape. This is performed by creating a stack order using single or multiple receiver algorithm. Both algorithms allow for the formation of multiple channels, but the single receiver algorithm is limited to flow convergence while the multiple receiver algorithm permits flow divergence. Both are therefore different from the single channel approach that is common to all previous applications of the self-similar grain size fining model. In all the model results presented here, we have used the multiple receiver algorithm, in which the proportion of water and sediment that is given from one node to its downslopes neighbouring nodes is proportional to slope.

In FastScape, uplift and subsidence rate are imposed as functions of space and time, but they can also be obtained by solving the flexure equation representing flexural isostasy of the lithosphere/crust system.

## 3 A model for grain size fining including lateral heterogeneity : GravelScape

### 3.1 Coupling: Incorporating the Self-Similar Grain Size Model into FastScape

We have generalized Fedele and Paola (2007)'s self-similar approach to two dimensions, i.e., following flow path, $s$, or the stack node ordering computed in FastScape. We call our algorithm GravelScape. For this, we replace any spatial integration or differentiation along the normalized river distance by its equivalent value along the normalized steepest descent flow path, $s^* = s/L_s$, where $L_s$ is the total length of the flow path. For example, the deposition rate is the spatial derivative of the flux along that flow path:

$$\frac{\partial q_s}{\partial s} = \frac{1}{L_s}\frac{\partial q_s}{\partial s^*} \tag{7}$$

In practice, we first compute the flow path and erosion/deposition rate along the flow path using the multiple receiver routing in FastScape. Next, sediment flux is computed by summing the erosion/deposition rate, $r$, down the flow path. Note that, where deposition takes place, $r$ is negative and consequently decreases the sediment flux $q_s$ in the summation along the flow path to the limit of $q_s = 0$. Once $q_s$ is computed, the downstream distribution of deposition $R$ can be calculated according to:

$$R = \begin{cases} (1-\gamma)\frac{r}{q_s} & \text{if } r, q_s > 0 \\ 0 & \text{otherwise} \end{cases} \tag{8}$$

Next, the dimensionless distance transformation $y^*$ is computed from a weighted summation of $R$ through the nodal stack. Where flow convergences, a weighted mean is used to compute the receiver $y^*$-values, where the weights are proportional to the upstream drainage area size of the donor nodes. In this way, when multiple flow paths converge, the largest drainage area flow path will dominate the downstream grain size distribution. Such an approach matches observations that grain size distributions of larger catchment rivers tend to dominate downstream grain size distributions (Harries et al., 2019).

Finally, using $y^*$, we compute the mean grain size using equation 1 or 2. In these equations, the value for $D_0$ is needed at every grid point within the model. For this, we compute a field $D_0$ that is obtained by propagating the value of the grain size

last deposited on the bed (which we call $D_t$) down the flow path of a given time step from where the channel originates. Using $D_0$ defined in this way at each grid point, we use equations 1 or 2 to calculate a grain size value, $D_x$, at all points of the grid where deposition occurs, which we also use to update $D_t$ for the following time step. At the start of the simulation, $D_0$ and $D_t$ are set equal to the mean grain size as produced by bedrock erosion in the mountain catchment. Note that deposition does not take place at every grid point at every time step. So $D_t$ is only updated to the newly computed grain size $D_x$ where net

deposition takes place. Finally, local erosion may take place in the basin causing an increase in sediment flux, and a reduction in $D_x$ fining rates.

In summary, we carry three grain size values at every grid point: (1) the grain size of material deposited at the current time step ($D_x$), (2) the grain size at the surface of the model, i.e., of the material last deposited at that point ($D_t$) and (3) the grain size in the 'source' area, i.e., where the flow path traversing the grid point originated ($D_0$). We show in supplementary materials

Figure SM1-1 computed values of $D_0$, $D_t$, and $D_x$ for a given time step under the assumption of single and multiple receiver routing. In the work presented here, we will only show computed grain size values using the multiple receiver algorithm.

## 3.2 Model setup

To illustrate the behaviour of the new model and, most importantly, to compare it to previous studies which computed grain size fining along a single, main channel, we chose a setup that is similar to that used by Duller et al. (2010). As shown in

Figure 1, this setup is composed of two regions, one subjected to uplift (the source or mountain region of length $L_M$) and the other to subsidence (the basin region of length $L_B$). Although our model is two-dimensional, the uplift and subsidence are assumed to be functions of one of the two horizontal components only, i.e., the $x-$ coordinate. Following the setup of Duller et al. (2010), the subsidence, $\sigma(x)$, in the basin area is given by:

$$\sigma(x) = \sigma_0 e^{-\alpha x/L_M} \tag{9}$$

where $\sigma_0$ is the subsidence at the orogenic front (i.e., at the limit between the source and basin areas, where $x = 0$) and $\alpha$ is the rate of decay of the subsidence away from the front. Large values of $\alpha$ result in most of the subsidence near the source, while small values of $\alpha$ result in a more distributed subsidence across the basin. This exponential function mimics relatively well subsidence curves resulting from flexure of the underlying lithosphere/crust under the weight of the topography created by uplift in the source area, without any lateral heterogeneity.

Duller et al. (2010) does not explicitly model the source area, but assume a constant sediment flux, $q_{s,i}$, originating from it. To reproduce this setup, we will assume that the source area uplifts at a rate $U$ chosen such that $q_{s,i} = (1 - \gamma)UL_M$. In our setup, we will need to run the model for a sufficiently long period of time for it to reach steady-state so that the flux is constant, and our results become comparable to those of Duller et al. (2010). Note that the values of the various LEM parameters, i.e., $K$, $m$, $n$ and $G$, do not influence the value of the steady-state flux coming out of the source area, only the time it will take to

reach the steady-state.

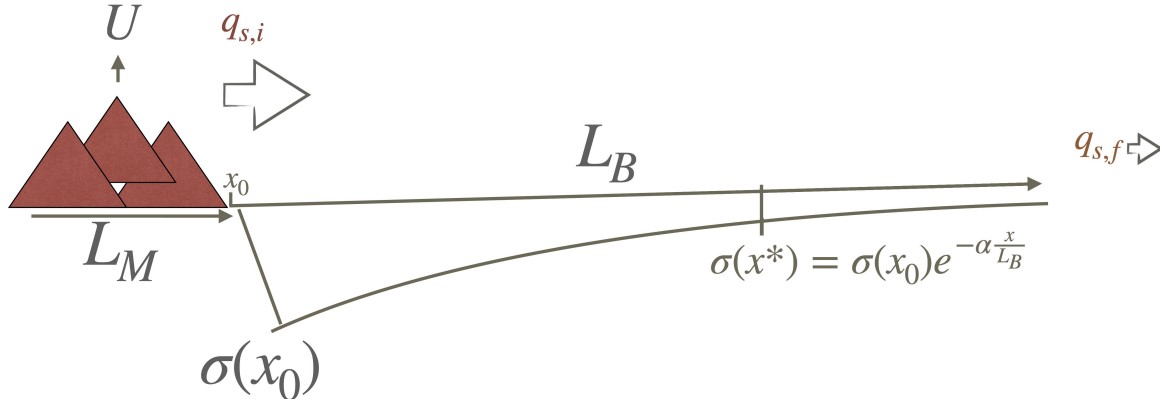

**Figure 1.** Sample basin set up using Duller et al. (2010) imposed subsidence. All figures within this work will be draining an orogenic front to the left to an eventual sink to the right even if only results within the basin are shown.

We can then follow Duller et al. (2010) and introduce a quantity, $F$ (see Figure 2), that is the ratio of the flux coming out of the source, $q_{s,i}$, to the flux of sediment that is trapped in the basin. As demonstrated by (Duller et al., 2010), $F$ and $\alpha$ are the major controls on the relative rate of grain size fining across the basin. $F$ can be written as:

$$F = q_{s,i} / \int_0^{L_B} \sigma(x)\, dx \tag{10}$$

$F$ indicates the under, over, or filled state of the basin. When $F$ is larger than 1 there is more incoming flux, $q_{s,i}$, than the basin subsidence can accommodate, producing an outflow of sediment from the basin ($q_{s,f} > 0$). In contrast, when $F$ is smaller than 1, the basin is under-filled and no sediment leaves the basin ($q_{s,f} = 0$). Here, systems that are characterised by $F$ values lower than or equal to 1 will be referred to as under-filled or limited bypass systems. Systems where $F$ is greater than 1, but less than or equal to 10, will be referred to as filling or low bypass systems (Figure2 a). Systems where $F$ is greater than 10, will be referred to as over-filled or high bypass systems (Figure2b). In Figure 2c we show how varying $\sigma_0$ while keeping both $\alpha$ and $q_{s_i}$ constant leads to different values for $F$.

From the definition of the flux at steady-state and by performing the integral of the subsidence given by Equation 9, we obtain the following relationship between $F$, $U$ and $\sigma_0$:

$$F = \frac{(1-\gamma)U L_M \alpha}{\sigma_0 L_B (1 - e^{-\alpha})} \tag{11}$$

In practice, we will impose values for $\sigma_0$, $\alpha$ and $U$ and deduce the corresponding value of $F$ from Equation 11.

Duller et al. (2010) have shown that for a given value of the incoming sediment flux from the source, $q_{s,i}$ and a given subsidence pattern, i.e., given values of $F$ and $\alpha$, the self-similar grain size fining model of Fedele and Paola (2007) predicts a

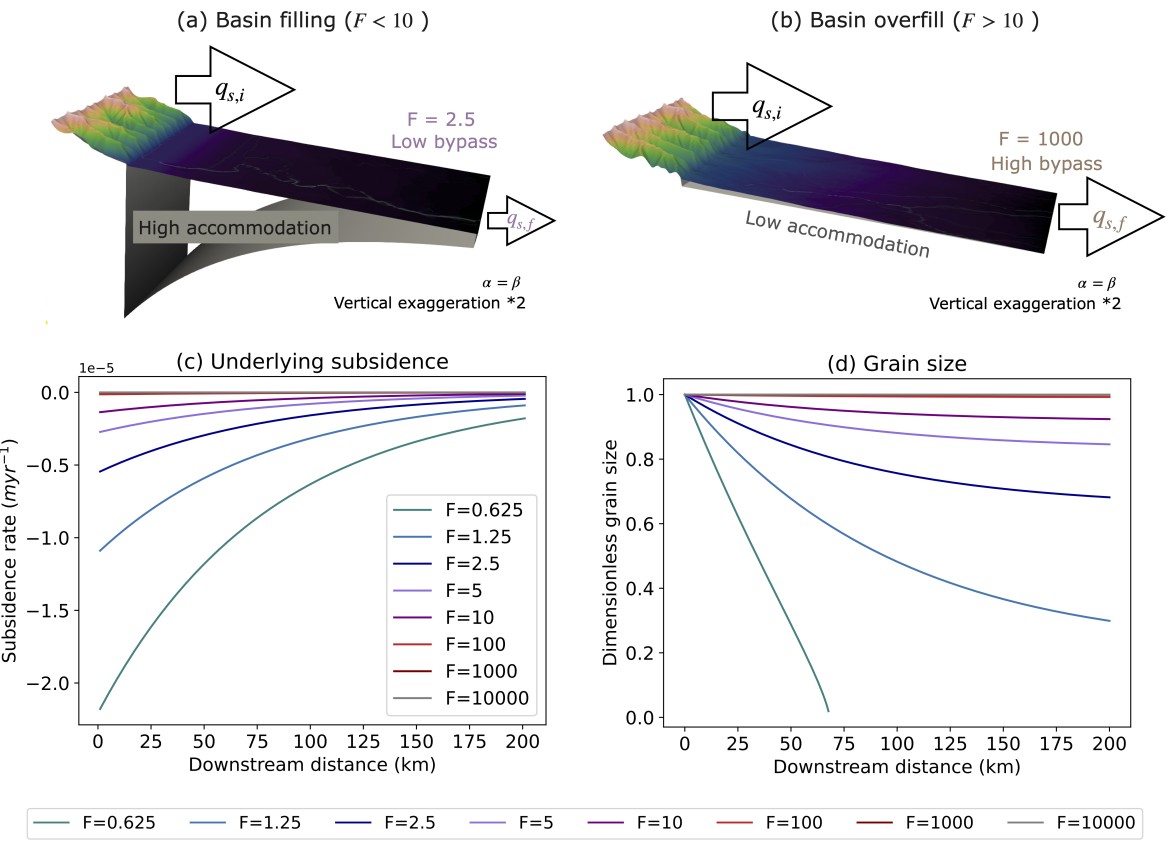

**Figure 2.** Examples of model runs that differ by their assumed subsidence $\sigma_0$ keeping both the incoming flux $q_{s,i}$, and subsidence decay rate $\alpha$ constant. (a) A high subsidence value leads to a reduced outgoing flux, $q_{s_f}$, a smaller $F = 2.5$ value, and creates a system that is in low bypass. (b) A low subsidence value leads to an outgoing flux, $q_{s_f}$, similar to $q_{s_i}$, a larger $F = 1000$ value, and creates a system that is in high bypass. (c) Subsidence curve for various values of $\sigma_0$ and thus $F$. (d) Resulting grain size trends by varying $F$ as obtained by Duller et al. (2010).

unique distribution of grain size in the basin. If we note that the sediment flux at any point, $x$, along the surface of the basin is the upstream integral of the deposition/subsidence rate, we can substitute Equation (9) into Equation (3), to obtain:

$$y^*(x^*) = -\ln\left(1 - \frac{1 - e^{-\alpha x^*}}{F(1 - e^{-\alpha})}\right) \tag{12}$$

which, in turn, can be used to obtain an analytical expression for the grain size fining curve. Examples of such solutions are given in Figure 2d. We will now compare results obtained with GravelScape with this "theoretical" solution. For this we will perform two types of model runs, one using the full two-dimensional nature of GraveScape, i.e., with a large number of nodes in the $y$-direction and one with only two nodes in the $y$-direction to mimic the behaviour of a single channel. To simplify the

description of the results we will use the following naming convention: the theoretical solutions (assuming deposition as equal to subsidence in a single channel) will be named DULLER, the full planform, multi-channel GravelScape solutions will be named GravelScapeMCH and the single channel (2 cell wide) GravelScape solution will be named GravelScape1CH. Model inputs used for standard GravelScape1CH and GravelScapeMCH set ups used to validate our model model against DULLER, are given in table 1.

**Table 1.** Reference GravelScape model parameter values for Duller Validation with 1 cell orogen and imposed subsidence. Note sometimes a range of values were tested during the validation (eg: $F$, $G$, or resolution), but the reference value has been used unless otherwise specified on the figure or in the caption.

| Parameters | Validation Set-Up (1-cell Orogen and imposed subsidence) |
|---|---|
| $K$ | $2 \times 10^{-5}$ m$^{1-2m}$/yr |
| $m$ | 0.4 |
| $n$ | 1 |
| $G$ | 0.2, 0.5, 1 (reference), 1.5, 2 |
| $Diffusion$ | 0.1 |
| $L_M$ | 1 000 m (simplified single cell orogen) |
| $L_B$ | 200 000 m |
| $L_y$ | 2 000 (GravelScape1CH) or 100 000 (GravelScapeMCH) m |
| $\Delta_x$ | 500, 1 000 (reference), or 2 000 m |
| $\Delta y$ | 500, 1 000 (reference), or 2 000 m |
| $\Delta t$ | 1 000, 10 000 (reference), 100 000 yrs |
| $\sum time$ | $25 \times 10^6$ yrs (steady-state) |
| $U$ | 0.01 m/yr |
| $D_0$ | 1 |
| $\phi_0$ | 0.75 |
| $C_V; C_1$ | 0.75 |
| $P_O$ | 20 |
| $P_B$ | 1 |
| $(P_O * L_M)/(P_B * L_B)$ | 0.1 |
| $\sigma_0$ | $-5.45 \times 10^{-5}, -2.72 \times 10^{-5}, -1.36 \times 10^{-5}$ (reference), $-1.5 \times 10^{-6}, -1.5 \times 10^{-7}, -1.5 \times 10^{-8}$ m/yr |
| $\alpha$ | 2.5 |
| $F$ | 2.5, 5, 10 (reference), 100, 1 000, 10 000 |

## 4 Results: Presenting GravelScape

### 4.1 Overview of GravelScapeMCH outputs in 2D

In figure 3 we show an example of the result of a GravelScape model run. We used model parameter values as given in Table 1) with an imposed subsidence producing a very high bypass basin ($F = 10000$) and moderately transport -limited conditions ($G = 1$). More specifically, in Figure 3, we show computed topography, erosion rate, drainage, grain size distribution over time ($D_t$), and grain size at an instantaneous time step ($D_x$) for a reference model experiment at a series of arbitrary consecutive time steps selected once the system has reached flux steady-state, i.e., the computed values of $q_{s,i}$ and $q_{s,f}$ do not change significantly with time. Note that the red colors in the erosion rate map correspond to areas where net erosion takes place, whereas the blue areas correspond to areas where net deposition takes place. By comparing panels b and e, we see that GravelScape predicts instantaneous grain size ($D_x$) only in regions of net deposition. We see that, even though the model has reached a quasi steady-state, important fluctuations in topography, discharge and grain size take place between time steps in response to what appears to be large avulsions. We note that, on average, the grain size tends to decrease from the mountain front to the base level. We also note that the predicted grain size is coarser in channels (regions of high discharge) and is highly variable spatially and temporally. Since the largest channel is often coarser than the smaller channels, the fining is greater when averaged across all channels (in the $y$-direction) than when considering the largest channel only.

This strong variability implies (1) that the predicted grain size in the channels under high bypass should be used to compare our model's predictions to the predictions of Duller et al. (2010)'s model where predictions are limited to a single, main channel, and (2) that the grain size predictions must be averaged over many time steps (eg: $> 10$ kyr) for comparison with Duller et al. (2010)'s model, which assumes that deposition rate is equal to subsidence rate and is thus a very smooth function of $x$ and $t$.

We also note that there is a small, yet non-negligible, grain size fining in the mountain area, such that the mean grain size of the material leaving the mountain front is not exactly at a value of unity, corresponding to the assumed grain size at the source. This also implies that to compare our grain size predictions to those of Duller et al. (2010), we must scale our predicted values in the basin area by the relative grain size predicted at the exit of the source area.

### 4.2 Model validation and comparison to past approaches

In Figure 4, we compare grain size predictions made by DULLER, GravelScape1CH, and GravelScapeMCH. In the top panels (a and b), we show the predicted surface and bedrock topographies obtained in GravelScapeMCH. In the middle panels (c, d,e , and f ) we show the predicted grain sizes and in the bottom panels (g and h), we show the predicted standard deposition rate for GravelScapeMCH compared to the imposed subsidence rate. In the three left panels (a, c, e, and g), we show the results of different model runs performed by varying the subsidence rate, $\sigma_0$, leading to different values of the parameter $F$ ranging from 2.5 to 100, i.e., from low to high by-pass. In the three right panels (b, d, f and h), results are shown for three values of the transport parameter $G$, i.e., 0.5, 1 and 1.5. .

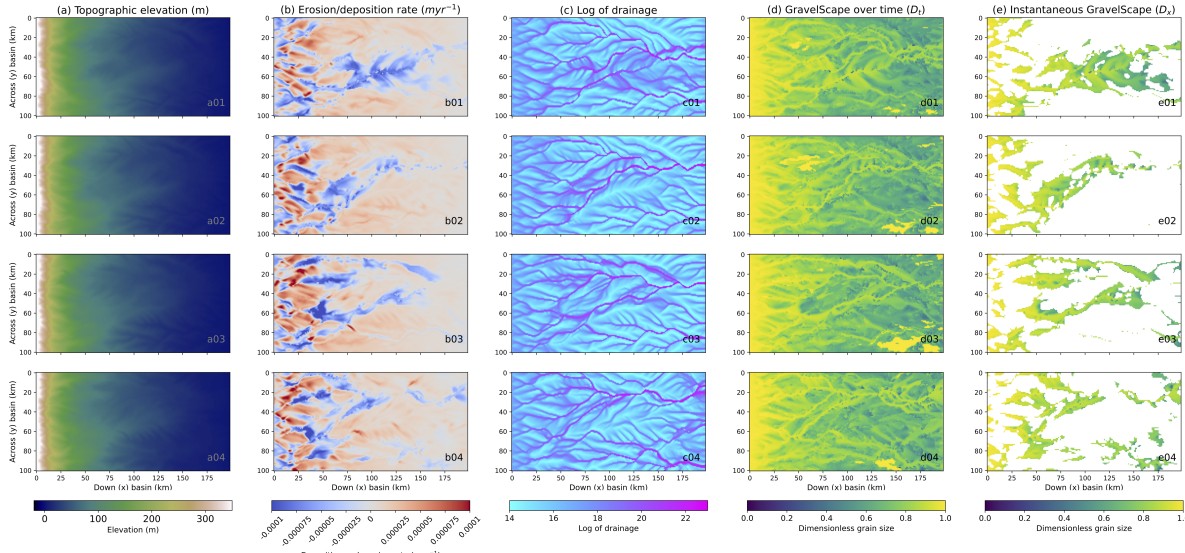

**Figure 3.** Steady-state predictions from GravelScape with multiple channels (2D) under high bypass (F=10 000). The four rows correspond to four successive time steps. Plan views of a)Topography, b) Erosion/deposition rate, c) Log of drainage area, d) surface grain size updated over time ($D_t$), and e) grain size ($D_x$) of material deposited in current time step.

We see that the GravelScape1CH predictions are, within numerical precision, identical to DULLER for all values of $F$ and $G$ (Figure 4 c and d), implying that when reduced to a single channel, GravelScape reproduces exactly the predictions of the self-similar model of Fedele and Paola (2007) when deposition rate is equal to subsidence rate, as done by Duller et al.

(2010). On the contrary, the GravelScapeMCH predictions deviate markedly from DULLER and the difference between the two model predictions increases with $F$ (Figure 4e) and $G$ (Figure 4f). With a constant $G = 1$, we tested additional high bypass values of $F = 10 - 10000$ and observed the similar fining trends as the $F = 100$ scenario in Figure 4e. For values of roughly $F > 10$, GravelScapeMCH predictions converge to a total grain size fining across the basin of approximately 15% (i.e., from 1 to 0.85 in relative grain size) regardless of the value of $F$, whereas DULLER predicts less and less fining as $F$ increases

to the point where for $F = 100$, there is almost no fining across the basin. Similarly, varying $G$ (with a constant $F$ value) in GravelScapeMCH leads to grain size fining that is two to three times larger compared to DULLER (or GravelScape1CH).

We also see that varying $F$ has little to no effect on the surface topography at steady-state (Figure 4a), whereas changing $G$ strongly affects it (Figure 4b). This is because, for values of $F \gtrsim 1$, the surface topography is set by surface processes almost regardless of the basement subsidence as shown in Braun (2022). $G$, on the other hand, affects the distance over which

deposition takes place and therefore strongly influences the height and slope of the depositional system (Equation 20 in Braun (2022)). We also see that varying $G$ causes not only variations in predicted surface topography but also in the standard deviation in deposition rate within the multi-channel model (Figure 4h), implying that steeper sedimentary systems are affected by higher

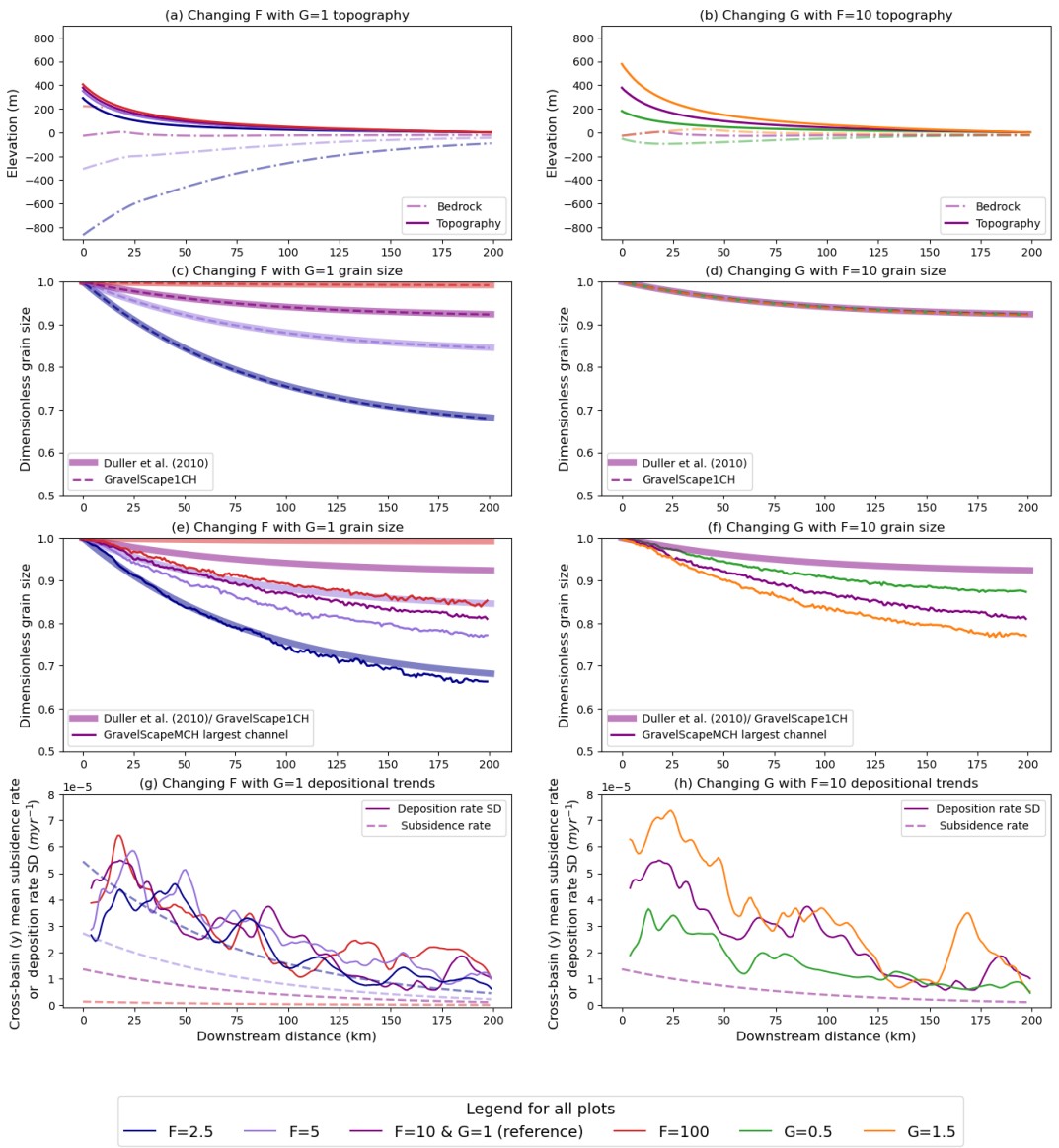

**Figure 4.** GravelScape model runs for different values of $F$ (left columns varying red to blue) that represent the degree of by-pass of the system and $G$ (right columns varying green to orange) that alters the topographic profile. Rows a)and b) show predicted topography (solid) and basement (dashed) geometry. c) and d) depict predicted grain size fining trends limiting the model width to single channel conditions (GravelScape1CH as dashed lines) compared to DULLER's (solid-opaque lines) solution. e) and f) contain predicted grain size trends for the largest channel under multiple channel conditions (GravelScapeMCH as solid-bold lines) compared to DULLER's (solid-opaque lines) solution. Rows g) and h) show the standard deviation of the deposition rate as solid-bold lines and the mean subsidence rate as dashed.

amplitude spatial variations in deposition rate. Recall that deposition rate is a direct parameter within the (Fedele and Paola, 2007) equations such that any changes in deposition rate will be reflected in grain size fining.

We can therefore conclude that the deviations in grain size fining from DULLER observed in GravelScapeMCH are related to the degree of by-pass of the system, when varying $F$, or to the shape of the surface topography changing the variation in deposition rate, when varying $G$. Further analysis on the internal dynamics of the system are conducted in Wild et al. (2024b).

### 4.3    Effect of local minima

Using GravelScape with values of $F < 1.25$ leads to predicted surface geometries that are so flat (less than a few tens of meters
over a 200 km distance) that local depressions appear in the topography that lead to the formation of lakes where the modified SPL equation cannot be used to predict deposition or erosion. In the depressions, the SPL is replaced by a simple depression filling algorithm, and the resulting deposition rate cannot be used to compute grain size fining according to Equation 1. We checked that these local depressions or minima do not influence our findings by computing the frequency of occurrence of such minima. This frequency is computed by counting the number of local minima in the main channel at a given position $x$ over $N$
time steps and dividing it by $N-1$. We found that the number of local minima is negligible for all model runs where $F > 1.25$. In Figure 5 we show the computed frequency of the local minima as a function of the $x$-coordinate for the model runs shown in Figure 4. We see that this frequency varies with the amplitude of the topography but also that it remains relatively low and does not vary significantly between the different model runs shown in Figure 4.

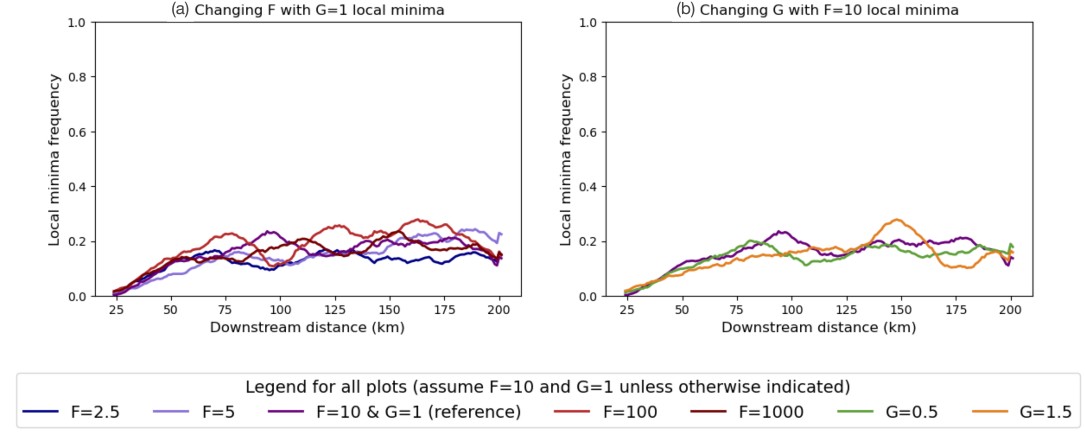

**Figure 5.** Computed local minima frequency for the GravelScapeMCH model runs shown in Figure 4.

### 4.4    Changing Spatial and Temporal Resolution

We performed another set of model runs to verify the sensitivity of GravelScapeMCH to the assumed spatial and temporal resolution as well as the time scale over which the results are averaged to compute the grain size that we compare to DULLER.

The results are shown in Figure 6. In panel a, we see that the grain size fining computed by GravelScapeMCH depends on the spatial resolution but that the results converge with increasing resolution. This mesh dependence is related to the mobility of channels that strongly affects the standard deviation in deposition rate. We will discuss this point in greater detail in the discussion. We also see that there is little dependence of the solution on the temporal resolution (value of the time step, $\Delta t$ - panel b) or on the time scale over which the grain size are averaged in GravelScapeMCH to be compared to DULLER, whether we perform this averaging over 5 Myr but with a varying number of time steps (panel c) or over 25 time steps of varying length (panel d) as long as this time scale remains larger than a characteristic time scale for channel avulsion ($\approx 250$ kyr).

## 5    Discussion

Gravel1CH can match the simple theoretical fits of DULLER, but we have also seen from the results presented above that grain size fining trends predicted by GravelScapeMCH differ markedly from DULLER although both models are based on the self-similar grain size fining model of Fedele and Paola (2007) that assumes that fining is controlled by deposition rate (scaled by surface sediment flux). Two main differences can be identified between the two models. First, DULLER assumes that deposition rate is equal to subsidence rate, while GravelScapeMCH computes deposition rate from an independent equation, namely the modified SPL or Equation 6. Second, in GravelScapeMCH, solving the SPL equation provides an estimate of the surface topography that seems to influence the fining, independently of the value of the imposed subsidence rate.

The most likely reason for the difference in grain size predictions between the two models is that, in GravelScapeMCH, although mean deposition rate must be equal to the imposed subsidence rate, local deposition rate is never equal to subsidence rate due to across basin variation (SD) (see figure 4 g and h). This is relatively easy to demonstrate if we look at the contour maps of deposition/erosion rate in Figure 3 that show great variability in that rate. More importantly, areas where grain size is computed correspond to areas of net deposition. This implies that grain size is, on average, computed in areas where deposition rate is larger than subsidence rate, as the mean deposition rate must be equal to the subsidence rate. This explains why, in a model, where deposition is restricted to narrow channels, the rate of fining must be greater in these channels than in the adjacent areas that experience less deposition or even erosion. This explains qualitatively why GravelScape predicts faster fining trends than Duller et al. (2010)'s model when the multiple channel dynamics is included (GravelScapeMCH) but similar fining trends when single channel is imposed (GraveScape1CH).

The results from our validation (Figure 4) indicate that single-channel solutions are most applicable under more uniform flow and early basin filling states (low $F$) where our GravelScape multi-channel solution showed little deviation. Such is likely the case in the Pobla Basin, Montsor Formation where Duller et al. (2010) applied the Fedele and Paola (2007) grain size fining model assuming subsidence is equal to deposition rate. The Montsor formation is described as a progradation of extensive alluvial fans filling a wedge top basin during a period of intense thrust activity and subsidence in the southern Pyrenees Axial zone (Duller et al., 2010). However, a more complex, multi-channel, lateral model that decouples deposition from subsidence rate is justified, if not necessary, to simulate grain size in systems in high bypass (high $F$), with steeper topography, or with more diverse geomorphology and stratigraphy (e.g. variations in channel dynamics, fan, and floodplain).

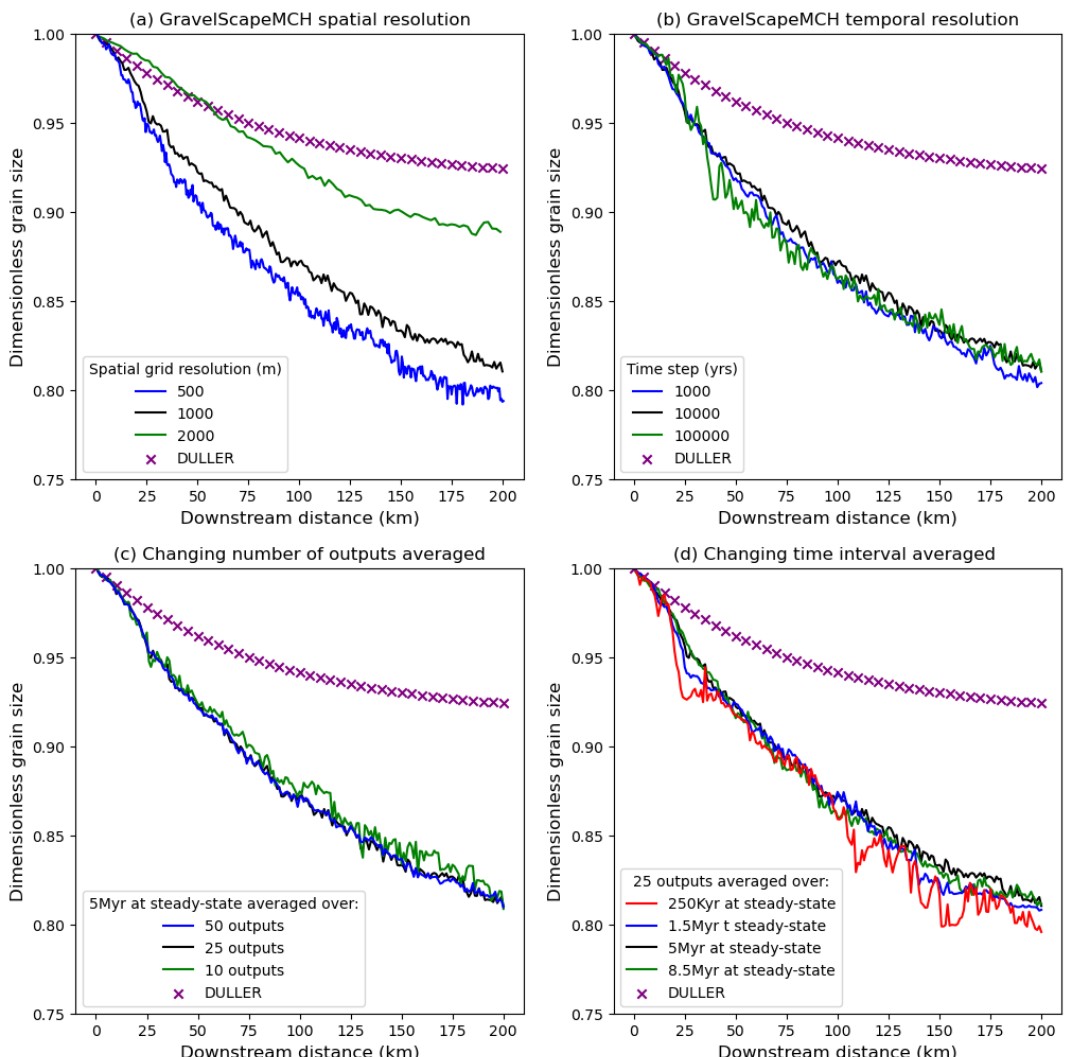

**Figure 6.** Effect of varying the spatial and temporal resolution in GravelScape for a moderatly high bypass $F = 10$ and $G = 1$ set-up: a) Varying spatial resolution $\Delta x$ and $\Delta y$; b) varying temporal resolution $\Delta t$; c) varying the number of times steps used to average the solution at steady-state over 5 Myr d) varying the length of time used to average the solution at steady-state over 25 time steps.

Additional factors (e.g. slope) that influence grain size fining in alluvial fans aside from subsidence and mean deposition rate, have long been debated in the literature (Stock et al., 2008).The difference between local deposition rate and subsidence rate led D'Arcy et al. (2017) to introduce a correction factor to Duller et al. (2010)'s model to interpret grain size data from two fans systems in Death Valley. To match the observed grain size trend, they multiplied the subsidence rate by the width of the fan as a function of distance from the fan apex. In doing so, they implicitly acknowledged the difference between deposition rate in

the fan active channel and the mean, across fan, deposition rate or the fact that to create the fan, channels have to move across

it and therefore locally match the subsidence rate with a deposition rate that is inversely proportional to the frequency at which the channel covers any given point of the fan surface. D'Arcy et al. (2017)'s correction factor is one example that justifies the need for our multi-channel grain size model that predicts lateral depositional variations, especially in systems where grain size fining cannot easily be explained through subsidence alone.

In GravelScape, this difference in deposition rate, and thus in grain size fining rate, between active channels and other parts of the depositional system, are taken into account "dynamically", as we compute the local variations in deposition rate from the modified SPL equation. This approach is not only based on a better physical (and mathematical) representation of intra-basinal processes but also yields estimates of the spatial grain size variability (between channels and interfluves) and thus an estimate of the temporal variability in grain size within a given stratigraphic section.

The relationship between grain size fining and surface topography is less obvious to decipher. We see (Figure 4h) that as $G$ increases and the topography becomes steeper, the amplitude of the variations in deposition/erosion rate also increases. Yet, the self-similar model of Fedele and Paola (2007) that we have incorporated in GravelScape does not explicitly use the shape of the surface topography to compute grain size fining. So the link that we evidence in Figure 4f between surface topography and grain size fining must be indirect and is therefore, in our opinion, also related to the difference in local deposition rate

between the two models as shown in Figure 4h. This topographic influence on grain size implies that factors that can increase topography, such as initial topography or certain basin geometries, could impact the grain size fining when multi-channel solutions are considered. This warrants further study that we present in Wild et al. (2024b) along with further applications of the multi-channel model.

Computing the surface topography as done in GravelScape also provides additional constraints when inverting grain size

data. As shown by D'Arcy et al. (2017), observed grain size fining trends are often affected by a strong local spatial variability in grain size, which limits our ability to unequivocally determine the shape of the subsidence function. In other words, inverting grain size data usually leads to a trade-off between $\sigma_0$ and $\alpha$. Using GravelScape, topographic information can be used as an additional constraint that must be satisfied by the model, potentially resolving the trade-off between $\sigma_0$ and $\alpha$. This statement must, however, be tempered down by two considerations. First, as we have shown above, topography is only sensitive to

subsidence rate when $F \lesssim 5$. Second, the shape of the surface topography is dependent on other, poorly constrained parameters that have been introduced in the modified SPL, such as $G$ or $K$. These findings add a new dimension to our understanding of grain size fining trends in sedimentary systems and to our use of grain size data to extract from the stratigraphic record useful information about past tectonic or climatic events.

## 6  Conclusions

We have incorporated the self-similar grain size fining model into a plan-view landscape evolution model. We named the resulting coupled model GravelScape. Results show that although our model can reproduce the solutions of Duller et al. (2010), when the assumption of a single channel, where deposition equals the subsidence rate is removed, GravelScape results differ and can show higher fining. The difference arises from local departures in deposition rate from the imposed subsidence rate that

arise from the channelized nature of the water flow and sediment transport as predicted by GravelScape. We have demonstrated that these local variations in deposition rate lead to a more rapid fining trend across the sedimentary system and should therefore be considered when interpreting grain size data. We have also demonstrated that the amplitude of grain size deviation from the single channel solution depends on the value of the by-pass parameter $F$. As $F$ increases, the differences between the GravelScape predictions and those from the one-dimensional model increase to the point where for values of $F$ larger than 100, grain size fining becomes independent of $F$. We have also shown that topography appears to controls fining through the parameter $G$ where steeper topographies produced greater depositional variation across the basin and more deviation from the single channel fining solutions. It also appears that these deviations in predicted grain size fining rate predicted by GravelScape (compared to single channel models) are in proportion to the standard deviation in deposition rate caused by channel avulsions and deposition/incisional pulses within the landscape.

These findings demonstrate the need to better understand what controls the amplitude and frequency of fluctuations in sedimentation rate. This is our objective in the second paper (part 2 (Wild et al., 2024b)) where we define in a quantitative manner the conditions under which grain size fining is principally controlled by subsidence or by internal processes.

# 7 Funding Declaration

This project received funding from the European Union's Horizon 2020 research and innovation program under the Marie Sklodowska-Curie grant agreement No 860383.

*Code availability.* Python Code for GravelScape is available on demand from the authors. The code will be made available on a public repository upon paper acceptance.

*Author contributions.*

. Amanda Wild: Conceptualization, formal analysis, investigation, methodology, software, validation, visualization, writing-original draft preparation, writing-review and editing

. Jean Braun: Supervision, resources, software, conceptualization, methodology, validation, visualization, writing-original draft preparation, and writing-review and editing

. Alexander Whittaker: Supervision, conceptualization, methodology, validation, and writing-review and editing

. Sebastien Castelltort: Supervision, conceptualization, and writing-review and editing

*Competing interests.* The authors declare that there are no competing interests.

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

# Grain size dynamics using a new planform model. Part 2: determining the relative control of autogenic processes and subsidence

Amanda Lily Wild[1,2], Jean Braun[1,2], Alexander C Whittaker[3], Marine Prieur[4], and Sebastien Castelltort[4]

[1] Helmholtz Center Potsdam, GFZ German Center for Geoscience, Potsdam, Germany
[2]The Institute of Geosciences, Universität Potsdam, Potsdam, Germany
[3]Department of Earth Science and Engineering, Royal School of Mines, Imperial College London
[4]Department of Earth Sciences, University of Geneva, Rue des Maraîchers 13, 1205 Genève, Switzerland
[1]**Correspondence:** Amanda Lily Wild (awild@gfz-potsdam.de)

**Abstract.** The interpretation of grain size trends within the stratigraphic record has a wide range of applications, including the identification of external forcing events. Within fluvial systems, it is not yet well constrained how autogenic processes, i.e. those internal to the basin, influence grain size signatures. Using a recently developed model, GravelScape (Wild et al., 2024a), that couples the self-similar fining model (Fedele and Paola, 2007) to a landscape evolution model, we investigate what controls the importance of autogenic processes and, in turn, their influence on grain size fining. For this, we perform a large number of numerical experiments by varying 1) the ratio between the incoming sediment flux and integrated subsidence rate ($F$), which characterizes the degree of bypass of the system, 2) the ratio of the discharge leaving the mountain to the discharge generated within the subsiding basin ($\beta$), which controls the shape of the topography of the basin, 3) the erodibility ($K$), which impacts the steady-state or transient nature of the basin, and 4) the transport coefficient ($G$), which determines the transport- vs detachment- limited behaviour of the depositional system that also influences the topography. We demonstrate that there exist two differing regimes for long term grain size fining: one dominated by autogenic processes, and one dominated by underlying subsidence. The subsidence-dominated regime occurs when the mean deposition matches the underlying subsidence which is typical of low bypass (filling) and low slopes systems (i.e. low values of $F$, high values of $\beta$, and low values of $G$). The autogenic-dominated regime occurs mostly under high bypass with steep topography when local variability in deposition rate is important (i.e. high $F$, high $G$, and low $\beta$). We also show that there is a strong correlation between the intensity of autogenic processes and the surface slope and across basin topographic variability (rugosity). We introduce a framework in which we map the different regimes for grain size fining as a function of by-pass ($F$) and surface geometry ($\beta$). We finally illustrate its use for the proper interpretation of grain size fining trends by positioning a series of natural systems within this framework.

## 1 Introduction

Sedimentary systems are an essential source of information regarding the nature, duration and amplitude of past tectonic and climatic events (Castelltort et al., 2023; Romans et al., 2016; Carretier et al., 2020; Armitage et al., 2011; Sømme et al., 2009). The fidelity of the sedimentary record is however affected by autogenic processes, i.e., that are internal to the sedimentary

systems (Scheingross et al., 2020; Hajek and Straub, 2017; Jerolmack and Paola, 2010). These autogenic processes involve sediment recycling at the intra-basin scale and have characteristic times that are usually smaller than those associated with the external perturbations (Scheingross et al., 2020). Despite this, autogenic processes can impact the preservation of longer time scale, externally-driven signals (Scheingross et al., 2020). This impact can be important and counter-intuitive. For example, Hill et al. (2012) has shown that, in certain cases, external signals may even be better recorded by lower preservation (subsidence) systems with little autogenic shredding rather than in a higher preservation, but highly reworked stratigraphic section.

While preservation is a function of the rate of creation of accommodation (mostly through subsidence), the amplitude of autogenic processes is mostly controlled by surface processes such as channel avulsion or depositional pulses that generate variability within the system. To quantify the importance of autogenic processes, Straub and Esposito (2013) as well as Jobe et al. (2016) have both described ratios of vertical aggregation relative to lateral mobility and variability appear to control stratigraphic completeness. Similarly, Ganti et al. (2011) have reported from analysing laboratory scaled experiments that autogenic processes affect the stratigraphic record on time scales that are smaller or equal to the channel avulsion time scale. Furthermore, Straub et al. (2009) have described autogenic processes as linked to the filling of topographic lows that can be described through an associated compensation timescale explained through avulsions and temporal variability in deposition rate. Toby et al. (2022) relate autogenic processes and compensation infilling of topographic lows as dependent on the system's surface active layer [1] .

Within the stratigraphic record, grain size fining observations have been commonly used to constrain subsidence patterns in space (Duller et al., 2010; Whittaker et al., 2011) and time (D'Arcy et al., 2017) or to document tectonic or climatic events (Armitage et al., 2011). Many of these studies have used the self-similar grain size fining approach of Fedele and Paola (2007) to interpret grain size data as primarily controlled by deposition rate, which they have equated to subsidence rate. However, in addition to external forcings, grain size trends are known to be influenced by topography and autogenic processes (such as avulsions or drainage re-organization) that will alter local erosion and depositional patterns (Hajek and Straub, 2017). The importance of channel mobility on grain size fining in particular has not been addressed in past applications of Fedele and Paola (2007)'s grain size self-similar model.

We have recently developed a planform grain size fining model (GravelScape) by coupling Fedele and Paola (2007)'s self-similar algorithm to a landscape evolution model (LEM) that predicts the spatial and temporal evolution of the surface topography from alluvial fan to plain environments and simulates processes such as rugosity (across basin variability in topography) and channel avulsions (Wild et al., 2024a) (also shown in the supplementary video). We used it to demonstrate, that the grain size fining trends predicted by Fedele and Paola (2007)'s approach under the assumption that deposition rate is equal to subsidence rate (Duller et al., 2010) are not valid in multi-channel landscapes with topography that are under a state of high by-pass, i.e., when incoming sediment flux is large compared to the basin-integrated subsidence rate. We have also shown (Wild et al., 2024a) that topography exerts a significant control on grain size fining under certain conditions, a conclusion that can only be reached with a model that predicts both grain size fining and topographic evolution.

---

[1] reworked layer equal to the depth between fluvial channels and interfluves, also can be referred to as the variability in topography across the basin or the rugosity

Here, we propose to use the coupled model to better quantify and parameterize the autogenic controls on grain size fining and determine under which topographic and subsidence conditions grain size fining trends can be used to constrain subsidence patterns as suggested by Duller et al. (2010). We will test a wide range of model parameters to determine what controls the amplitude of autogenic processes in sedimentary systems and their subsequent impact on stratigraphic grain size fining. More specifically, we quantify the difference in grain size fining between multi-channel and single channel approaches (creating a parameter called grain size deviation) and attempt to explain it through correlations with basin internal dynamic parameters. From this we develop a conceptual framework describing under what basin conditions autogenic vs. subsidence dynamics dominate the grain size record in the stratigraphy.

## 2 Methods

### 2.1 GravelScape

To study the importance of autogenic processes relative to external forcings on grain size fining trends, we use a coupled model (GravelScape) that is fully described in (Wild et al., 2024a). Here, we will only give essential elements and introduce equations that are necessary for the comprehension of the work presented here. The model comprises a landscape evolution model (LEM) solving the Stream Power Law (SPL) enhanced for the effect of sediment transport and deposition (Davy and Lague, 2009; Yuan et al., 2019):

$$\frac{dh}{dt} = U - K\tilde{p}^m A^m \left(\frac{dh}{ds}\right)^n + \frac{G}{\tilde{p}A} \int_A (U - \frac{dh}{dt})dA \tag{1}$$

where $h$ is surface topography, $U$ is surface uplift or subsidence, $K$ is the erodibility parameter, $G$ a dimensionless depositional parameter, $\tilde{p}$ represents variations in precipitation rate around a mean value that is included in the definition of $K$ and $G$, $A$ is drainage area, $\frac{\partial h}{\partial s}$ is topographic slope in the direction of water flow, and $m$ and $n$ are the area and slope exponents, respectively. $G$ controls whether the system is transport-limited ($G > 0.4$) or detachment-limited ($G < 0.4$). Its value is not well known, but it has been constrained to be moderately transport limited of the order of 0.7 (Guerit et al., 2019) from a wide survey of sedimentary fans. Values of $K_f$ are poorly constrained as its units and value strongly depend on the slope exponent $m$. The direction of water flow is computed from the surface topography and allows for flow divergence by assuming that, at every node of the model, discharge is distributed to all lower elevation neighbouring nodes in proportion to slope.

Grain size is computed using Fedele and Paola (2007)'s self-similar grain size fining model for gravel, which assumes that mean grain size, $\overline{D}$, and its standard deviation, $\phi$, vary in a constant ratio during fining that is, in turn, controlled by the ratio between local deposition rate, $r$, and sediment flux, $q_s$, according to:

$$D(x^*) = \overline{D_0} + \phi_0 \frac{C_2}{C_1} e^{-C_1 y^*} - 1 \tag{2}$$

where $x^*$ is a dimensionless distance along flow path and:

$$85 \quad y^*(x^*) = \int_0^{x^*} R^*(x^*) \, dx^* \tag{3}$$

with the primary components controlling the fining are $R = r/q_s$ and $\overline{D_0}$ is the mean grain size where flow initiated. See Wild et al. (2024a) for a more detailed version of the coupling of the two equations and see Fedele and Paola (2007) for a description of the coefficients $C_1$ and $C_2$. Note that, in GravelScape, the deposition rate $r$ and sediment flux $q_s$ are obtained from the solution of the LEM, contrary to most previous uses of Fedele and Paola (2007)'s model that have made the simplifying

assumption that deposition rate can be directly equated with subsidence rate (e.g., Duller et al., 2010; Whittaker et al., 2011).

We will use a controlled setup similar to that of Duller et al. (2010) and also used in Wild et al. (2024a), in which sediment is produced in an orogenic area of width $L_M$ uplifting at a rate $U_0$ resulting in a sedimentary flux, $q_{s,i} = U_0 L_M$. Subsidence rate, $\sigma$, in the adjacent basin of width $L_B$ is assumed to vary as an exponential function of distance, $x$, from the mountain front:

$$95 \quad \sigma(x) = \sigma_0 e^{-\alpha x/L_B} \tag{4}$$

simulating flexural isostasy under the weight of the adjacent mountain. The elevation at the opposite side of the mountain front (i.e. right hand-side, edge of the basin in our set-ups) is assumed to be held at a constant elevation which we will refer to as the base level. As proposed by Duller et al. (2010), we introduce the parameter $F$:

$$F = q_{s,i} / \int_0^{L_B} \sigma(x) \, dx \tag{5}$$

that measures the degree of by-pass of the system. Small $F$ values (i.e., $1 < F < 10$) correspond to low bypass systems where most of the sediment coming from the mountain is trapped in the basin, whereas large $F$ values (i.e., $F > 10$) correspond to high bypass systems where most of the sediment coming from the mountain leaves the basin at its outer end. Under these conditions, one can derive an analytical solution to Fedele and Paola (2007)'s fining model (Wild et al., 2024a), which we will use to estimate, by comparing it to GravelScape's predictions, the contribution from autogenic processes to the grain size fining

trend, relative to that resulting from the imposed basement subsidence.

## 2.2   Controls on surface topography in a sedimentary system

As shown in Braun (2022), fan extent and the subsequent foreland basin long-profile is mostly controlled by the distribution of rainfall between the mountain (source) area and the basin (sink) area, with basement subsidence only playing a secondary role in low bypass (low $F$) systems. In high bypass systems, the sedimentary flux remains relatively constant across the basin.

At steady-state using the stream power law (Equation 1), sedimentary flux is equal to the product between drainage area (to power $m + 1$) and slope (to power $n$), which implies that the slope must vary as the inverse of discharge. Near the mountain front, discharge is relatively constant and equal to the product of the mountain surface area by the assumed precipitation rate.

The slope must therefore be relatively constant, which leads to the formation of a sedimentary fan. Away from the mountain front, rainfall in the basin substantially contributes to the discharge, which therefore increases and causes the slope to decrease to form an alluvial plain. Therefore, the transition between the steep, constant slope fan and the alluvial plain takes place where the contribution to discharge from rainfall in the basin equates the discharge from the mountain area. This explains the broad one-to-one relationship between upstream catchment area and fan area across many scales (Bull, 1962; Blair and McPherson, 1994). It also implies that one of the main controls on the shape of the topography in the basin area is the difference in precipitation rate between the mountain and basin areas (Braun, 2022). To illustrate this point, we can derive the parameter $\beta$:

$$\beta = \frac{\nu_M L_M \alpha}{\nu_B L_B} \tag{6}$$

where $\nu_M$ and $\nu_B$ are the relative precipitation rates in the mountain and basin areas, respectively. $\beta$ is the ratio of the contribution to discharge from the mountain area, $\nu_M L_M$, and from precipitation in the basin, $\nu_B L_B$, multiplied by the relative wavelength of the subsidence function, $\alpha$. $\beta$ is in fact the ratio of the length/size of the fan, $\nu_M L_M / \nu_B$ to the size of the subsidence function, $\alpha / L_B$.

In short, $\beta$ is a measure of the difference in area (extent) and precipitation rate between the orogen catchment and the sedimentary basin. Combinations of high precipitation and drainage area in the orogen with low basin length and basin aridity result in high, orogen dominant, $\beta$ values. Inversely, large basin areas, especially with higher precipitation relative to the orogen, result in low, basin dominant, $\beta$ values. We keep $\alpha$ constant in all our simulations and change only $v_m$ to increase or decrease $\beta$. To emphasize the impact of changing $v_m$ on $\beta$, within figures and referring to specific values within the figures, we normalize $\beta$ by $\alpha$ (Figure 1).

Different topographic profiles predicted by GravelScape for different values of the parameter $\beta$ are shown in Figure 1 . We see that as $\beta$ increases, the transition point between the steep fan and the curved alluvial plain moves towards the edge of the basin and the surface topography evolves from concave and steep near the mountain front (low $\beta$ values) to convex and flat (high $\beta$ values). High $\beta$ values correspond to the "constrained systems" in which the distance from the mountain front to the edge of the basin is smaller than the natural width of the fan, i.e., the width it would occupy if it was allowed to develop beyond the base level.

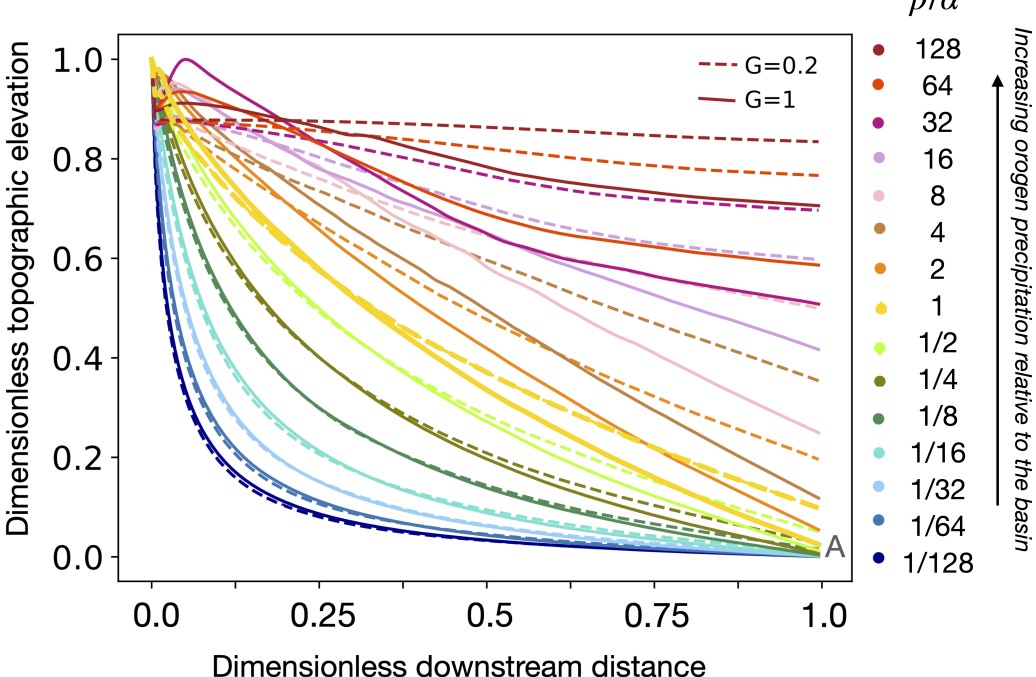

**Figure 1.** Surface topography averaged across the basin at steady-state predicted by GravelScape for various values of the parameter $\beta$ defined in Equation 6 and for different values of the depositional parameter $G$ (transport ($G = 1$) vs detachment ($G = 0.2$) limited). All topographic profiles have been normalized to 1 at the mountain front.

Note that $\beta$, through its control on discharge partitioning between the mountain and basin contributions only affects the position of the transition from steep to curved segments (and subsequent channel profile), whereas the slope at the mountain front (and therefore absolute topographic height) is controlled by additional factors (e.g., $G$ and $K$) and by the magnitude of

the sediment flux from the mountain, $q_{s,i}$, as explained by the analytical solution for the slope at $x = 0$ given in Braun (2022):

$$S(x = 0) \propto \left( \frac{G q_{s,i}}{K L_M^{m+1}} \right)^{1/n} \tag{7}$$

for high by-pass systems (i.e., when subsidence can be neglected). Essentially, the higher the value of $G$ (and the lower the value of $K$), the steeper the fan with the same fan extent as shown in Braun (2022); Wild et al. (2024a). Note that the dependence of

basin slope on $G$ and $K$ is not apparent in Figure 1 where we chose to normalize the topographic profiles. In the case $n = 1$, $K$ also controls the response time of the system (i.e., the time it takes to reach its final steady-state height) but $G$ does not (Braun, 2022).

In situations where the surface topography predicted by GravelScape is characterized by very low slopes (eg: underfilled basins with low $F$, low $G$, or high $\beta$), local minima can develop that affect the computation of the flow routing needed to

solve the modified SPL equation (Equation 1). In these situations, we use the method developed in Cordonnier et al. (2019) to

adjust the flow routing and compute the geometry of the resulting lakes forming around each local minimum. In these filled lake, Yuan et al. (2019)'s algorithm to solve equation (1) cannot be used and sediment is uniformly dumped as a first-order attempt to represent lacustrine deposition and no grain size can be accurately computed using Fedele and Paola (2007)'s grain size fining model. We checked that all model runs presented in this work were not strongly influenced by the presence of local minima.

In this work, we will vary model parameters $G$, $K$, $F$, and $\beta$ to assess the impact of subsidence and topography on grain size fining under a near constant orogen flux at steady-state. For simplicity, we will refer to the respective parameters $G$, $K$, $F$, and $\beta$ as the depositional, erodibility, bypass, and orogen discharge efficiency.

## 2.3 Modeled Autogenic Dynamics

Within our model, the river planform changes over time and space despite constant forcing conditions (see supplementary video) and we refer to this as model autogenic dynamics. These changes arise from the interactions between the depositional and erosional terms in Equation 1. Erosion leads to the formation of channels and deposition to their progressive infilling, which, in turn, affects local slope and the relative distribution of water flow between a node and its neighbours. This may lead, through downstream cascading, to discrete events that reorganizes large parts of the drainage network similar to avulsions that have been observed in laboratory experiments (Clarke et al., 2010)) and natural systems (Smith et al., 1989).

Hajek and Straub (2017) describe many autogenic processes, and their associated landforms, on a basis of spatial and temporal scale. Our landscape evolution model can only reproduce autogenic processes that occur over long timescales ($10^2 - 10^7$ years) as well as large lateral ($10^1 - 10^4$ km) and vertical spatial scales ($10^0 - 10^2$ m). This scale matches with the descriptions of Hajek and Straub (2017) for autogenic dynamics such as 1) the regrading of the depositional surface (longitudinal river planform changes), 2) avulsions, and 3) channel convergence, divergence, and, to a limited extent, bifurcations. All of these are observed in the model (as described above). Smaller scale autogenic processes described in Hajek and Straub (2017), with vertical scales under $10^0$m, such as those that involve bedforms (e.g. dunes or bars) or channel reach dynamics (e.g. riffle and pools; cut banks and point bars; meanders dynamics), cannot be reproduced in a landscape evolution model based on Equation 1.

## 3 Model Results

### 3.1 Grain size fining deviation, $\Delta D$

We now present results obtained with the coupled model to quantify the relative contributions from external forcings and autogenic processes to the control of grain size fining trends. For this, we define $\Delta D$ the difference at the basin outlet between the multi-channel (2D) grain size GravelScape solution computed in the largest channel and that predicted for a single channel using Duller et al. (2010)'s method, i.e., assuming that deposition rate is equal to the imposed subsidence rate. We will call this

quantity the grain size fining deviation and define it as:

$$\Delta D = < \overline{D}(x_{max}, y = y_{MC}) - \overline{D}_D(x_{max}) >_{SS} \tag{8}$$

where $\overline{D}(x_{max}, y = y_{MC})$ is the mean grain size computed by GravelScape at the exit of the basin within the largest main channel (i.e., at the $y$-location of the maximum discharge), $\overline{D}_D$ is the mean grain size predicted by Duller et al. (2010) and the symbols $<>_{SS}$ indicating a temporal average, once the system has reached steady-state. We computed $\Delta D$ for a large number of simulations varying $G$, $K$ and $\beta$, which control the surface topography, and varying $F$ (through $\sigma_0$ while keeping a constant $U$), which controls the degree of bypass of the system. The results are shown in Figure 2 where each panel shows how $\Delta D$ varies as a function of $\beta$ and $F$. The dependence on $K$ and $G$ can be appreciated by inspecting the different panels from left to right and top to bottom, respectively. All values are computed when the system has reached steady-state (constant apex topography) or is approaching it (i.e., within 90 percent of their steady-state topography, for model runs characterized by a low value of $K$). The regions of model space where local minima dominate are left blank to exclude them from our interpretation of these results.

We see that the grain size deviation, $\Delta D$, is controlled by all four factors, with $\Delta D$ increasing with increasing values of $F$, $\beta$ and $G$ but decreasing with increasing values of $K$. If we discard the model runs with low values of $K$ that have not yet reached their steady-state at the end of the experiments, i.e., those shown on the left column in Figure 2, we see that the dependence on $K$ is minimal.

In absolute term, the maximum values of $\Delta D$ are around 15 to 20% and are reached for high $F$ and high $G$ values, corresponding to systems in high bypass and in transport-limited conditions. Conversely, systems that are in low by bypass (low $F$) or in detachment limited conditions (low $G$) show less than a few percent deviation in grainsize fining compared to the predictions of a one-dimensional model assuming that fining is controlled by subsidence only.

In Figure 3, we show the grain size predicted by GravelScape at the exit of the basin, which we call $\overline{D}_e$. A value of 1 indicates no fining and a value of 0.5 corresponds to 50 percent fining from the original source distribution. Each panel corresponds to the same specific values of $K$, $G$, $\beta$ and $F$ as in Figure 2. As demonstrated by Duller et al. (2010), we see a strong dependence of grain size fining on $F$, with high by-pass (high $F$) systems showing the least fining (smaller value of $\overline{D}_e$). Contrary to Duller et al. (2010), we also see a strong dependence on $\beta$ and $G$ and thus on topography. Finally, we see little to no dependence of $\overline{D}_e$ on $K$, demonstrating that the apparent dependence of $\Delta D$ on $K$ is indeed an artifact due to the fact that the model experiments with low $K$ values have not reached steady-state. Figure 3 also demonstrates that similar downstream final values of fining can be observed by changing multiple parameters (e.g., $\beta$ and $G$) for the same $F$ values.

For completeness, we show in supplementary material, Figures 3 and 4, plots of predicted grain size and other autogenic quantities derived from the model as a function of downstream distance.

Combining the results from the Figures 2 and 3, we see that the greatest grain size deviation is produced under high bypass (high $F$) because GravelScape predicts much more fining than expected from Duller et al. (2010). This is because of a strong dependence of grain size fining on topography, which is not predicted by Duller et al. (2010), as high values of $G$ corresponding to more transport-limited systems producing higher fans, and low $\beta$ values producing shorter and steeper fans cause more fining

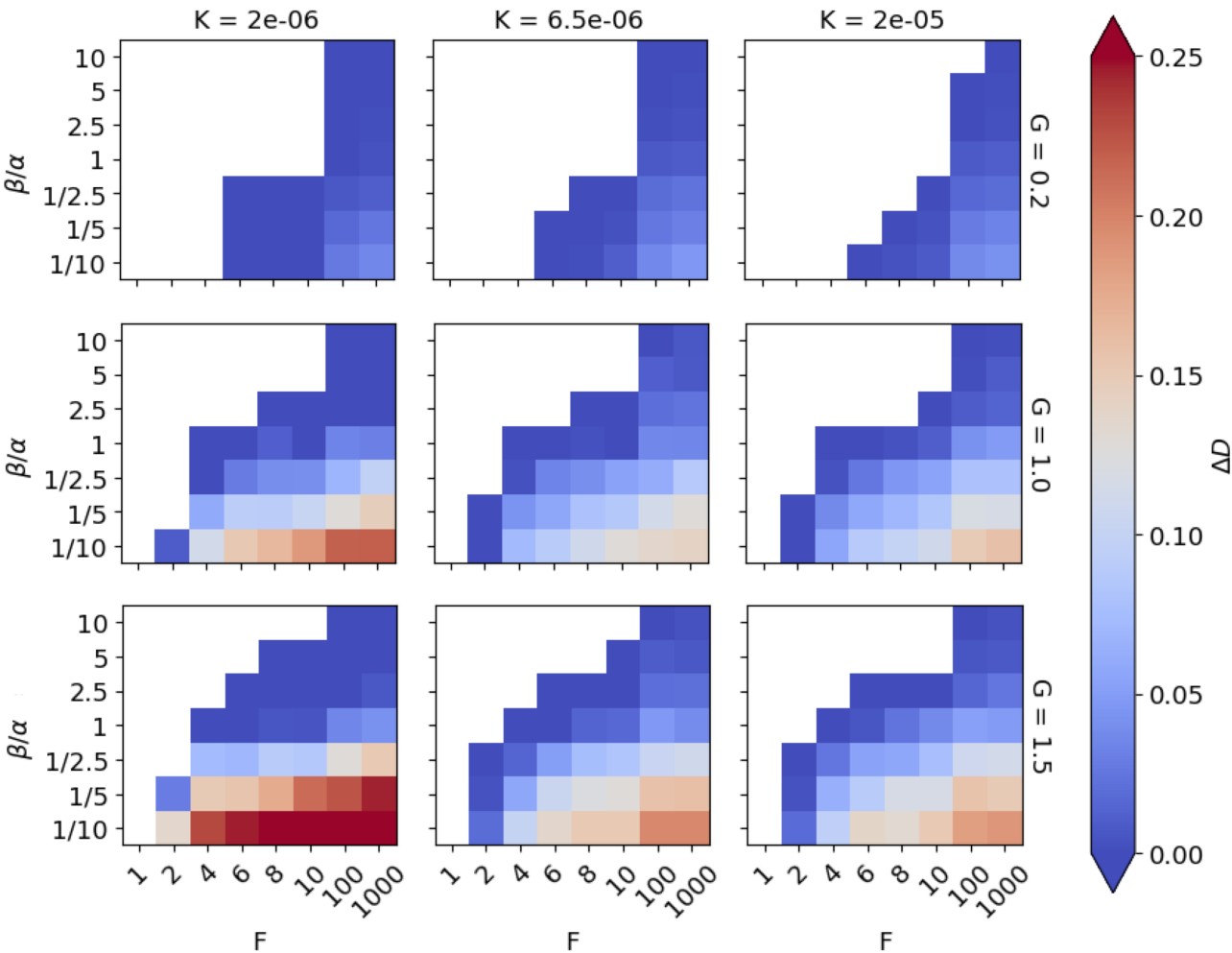

**Figure 2.** Grain size deviation of the GravelScape multi-channel solution relative to the subsidence controlled Duller et al. (2010) solution with changing model parameters $F$, $\beta$, $G$, and $K$. Areas in red highlight high internal dynamics (eg: topography, channel dynamics, etc) control on grain size fining.

215    and thus grain size deviation from Duller et al. (2010). The dependence on $K$ is less important, except that systems that are characterized by low values of $K$ will take longer to reach steady-state and are therefore likely to produce more fining than expected from their basement subsidence.

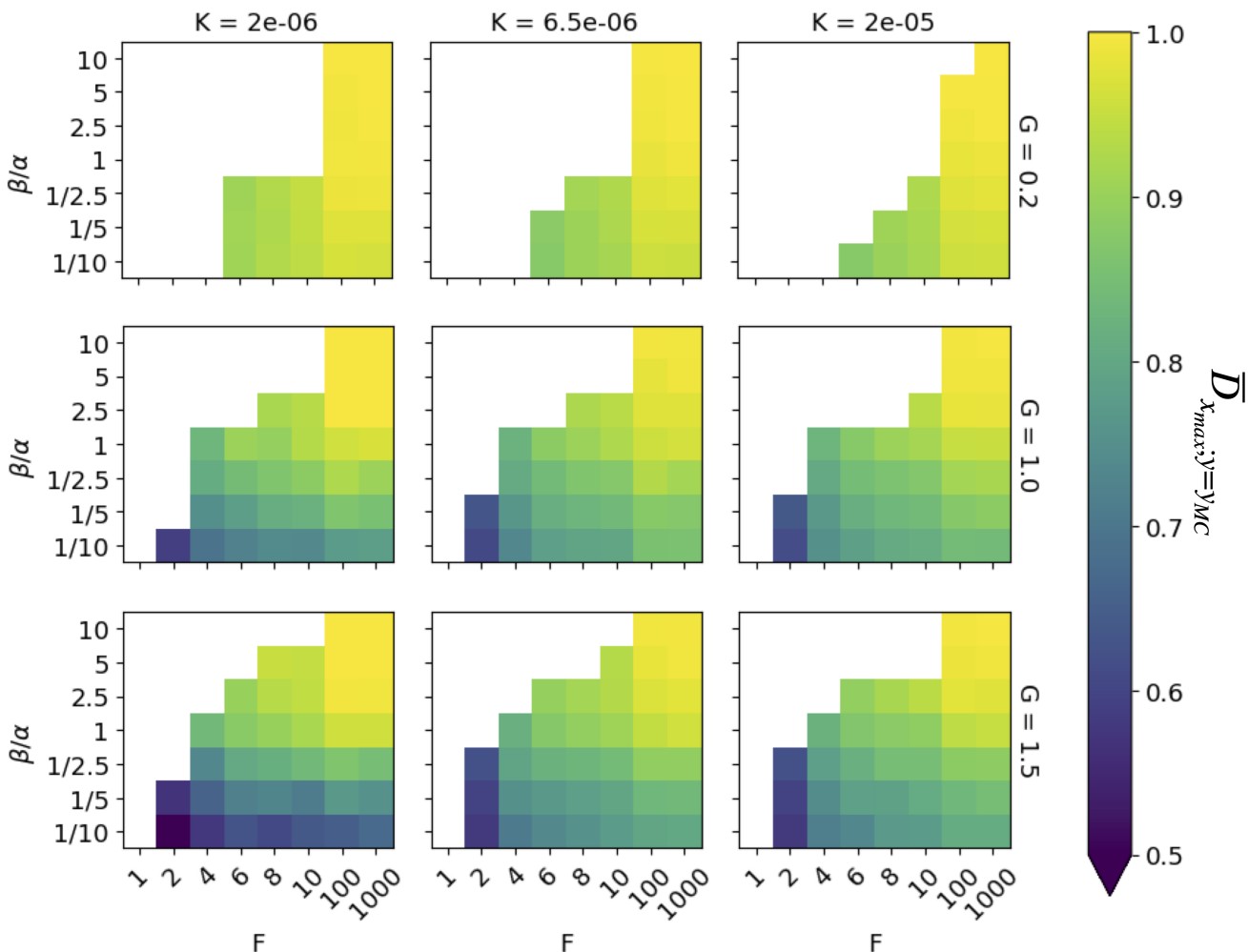

**Figure 3.** Computed grain size at the outlet of the basin within the largest channel of the GravelScape multichannel solution for various values of the model parameters $F$, $\beta$, $G$, and $K$ at or near (within 90% of) steady-state.

To demonstrate this last point, we ran the low $K$ experiment (i.e., where $K = 2 \times 10^{-6} \ m^{1-2m} yr^{-1}$) for longer than the reference 25 Myrs used in all model runs presented in Figures 2 and 3. We show the results in Figure 4. In Figure 4a we show the evolution of the basin apex topography as a function of time with the different time periods over which we computed the deposition rate and grain size fining shown in different colors: dark blue when the system has reached 90% of its final topography, light blue when it has reached 95% and orange and yellow when it has reached steady-state. In Figure 4b and c, we compare the predicted deposition rate (averaged in the $y$-direction and over the time span indicated in panel a) to the imposed

subsidence rate, and the predicted grain size fining (similarly averaged) to Duller et al. (2010)'s predictions, respectively. We

also show in Figure 4c the grain size fining obtained at steady-state with a larger value of $K$ (i.e., $K = 2 \times 10^{-5} m^{1-2m} yr^{-1}$) for reference. We see that the mean deposition rate converges towards the subsidence rate and the fining rate converges towards the solution predicted with a higher $K$ value, as the system moves towards steady-state (i.e., from dark blue to yellow). This clearly demonstrate that $K$ has no influence on the grain size fining at steady-state, but shows an impact during the transient steady-state (i.e. even with values within 90% of steady-state showed deviation) where more fining was produced than expected

(similar to a lower $F$) and more deviation from the constant, imposed subsidence rate. This is because, before the system reaches steady-state, the erosion rate in the mountain is smaller than the imposed uplift rate and, therefore, the flux coming out of the orogen, $q_{s,i}$, is smaller than the value, $U \times L_M$, we have used to compute $F$ and impose constant subsidence rate. In Wild et al. (2024b), we will further develop this point for natural systems and, in particular, for foreland basins, where subsidence rate and erosion rate in the source area are intimately linked by flexural isostasy.

## 3.2  The link between internal dynamics and grain size deviation

### 3.2.1  Depositional divergence ($d_v$)

We now proceed to determine the link between divergence in grain size fining, $\Delta D$, and autogenic processes. To compute grain size fining, both GravelScape and Duller et al. (2010)'s approaches use the model of (Fedele and Paola, 2007) , which assumes that fining is in proportion to deposition rate (scaled by sediment flux). The difference between the two methods (multiple

vs. single channel) is therefore likely to be explained by differences in the effective deposition rate they predict as mentioned in (Wild et al., 2024a). To quantify this difference, we define a parameter sensitive to local deposition rate fluctuations. We explicitly remove the background mean deposition rate, induced by basement subsidence, to isolate the amplitude of depositional variability. We call this parameter the depositional divergence rate, $\dot{d}_v$, defined as:

$$\dot{d}_v = \frac{< (\dot{d}_b - \sigma) < 0 >_{x,y,SS}}{< \dot{e}_o >_{x,y,SS}} \tag{9}$$

where $\dot{d}_b$ is the deposition rate (units of $m/yr$) computed by GravelScape (negative where/when there is deposition and positive where there is erosion), $\sigma$ is the subsidence rate ($m/yr$), $(< 0)$ means that only deposition is considered, and $<>_{x,y,SS}$ indicates an average in the $x$ and $y$-directions and over time (once the system has reached steady-state). The denominator ($\dot{e}_o$) is the mean erosion rate (in $m/yr$) in the source area (the mountain). To illustrate this concept, we show in figure 5a the patterns of deposition/erosion rate predicted by GravelScape in an arbitrary model run at an arbitrary time step. We see that the system

is dominated by deposition (because the basin basement is forced to subside) but that large variations in deposition rate appear in response to the channelized nature of transport in GravelScape. In Figure 5b, we show profiles of the deposition/erosion rate obtained by averaging values obtained by GravelScape in the $y$-direction for different values of the model parameters $F$ and $G$. We see that the deposition rate follows the trend of the imposed subsidence rate but that relatively large variations in erosion rate are predicted, even after averaging in the $y$-direction. The resulting values of the depositional divergence are shown in

Figure 5c.

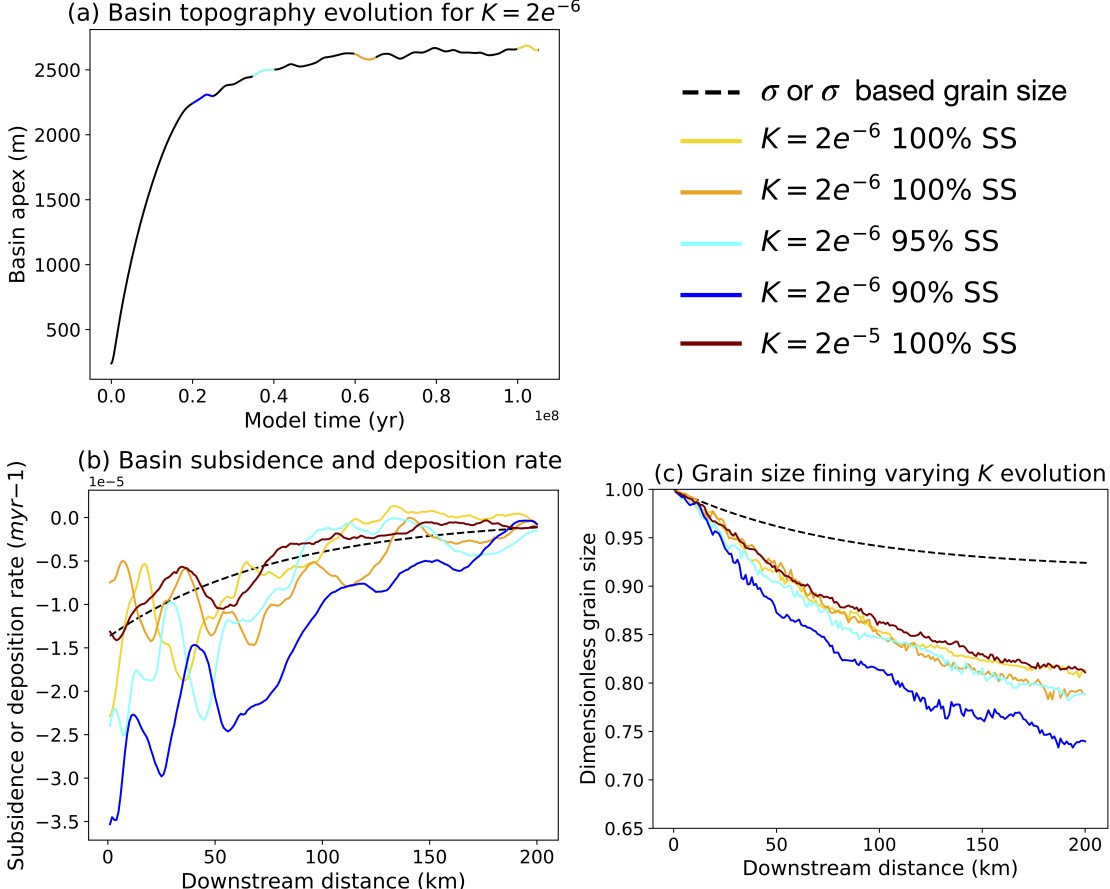

**Figure 4.** Deposition rate and grain size fining on the way to steady-state. a) Time evolution of the maximum topography in the basin with 2.5 Myr time intervals over which averaging is performed in panel b and c indicated in different colors. b) Imposed subsidence rate (dashed line) and deposition rate averaged in the $y$-direction and over the time span indicated in panel a. c) Grain size fining predicted by Duller et al. (2010) in response to subsidence only (dashed line) and grain size predicted by GravelScape at different times in the evolution of the system towards steady-state. The brown line corresponds to a solution with a high value of $K$ that has reached steady-state. All model runs assume moderate by-pass ($F = 10$), transport-limited conditions ($G = 1$) and low mountain precipitation $\beta/\alpha = 10$. All solutions shown are not affected by local minima.

At steady-state (Figure 5), $\dot{d}_v$ is a direct measure of the relative amplitude of autogenic processes around the mean, i.e., the processes that cause deposition and erosion events unrelated to the external forcing, in our case, the sediment flux from the mountain and the basement subsidence in the basin. Indeed, if the autogenic processes are negligible, deviations in local deposition rate are small compared to the imposed subsidence rate and $\dot{d}_v$ is small in comparison to the incoming sediment flux. Alternatively, if local deviations in deposition rate become more important than the incoming sediment flux, $\dot{d}_v$ is larger

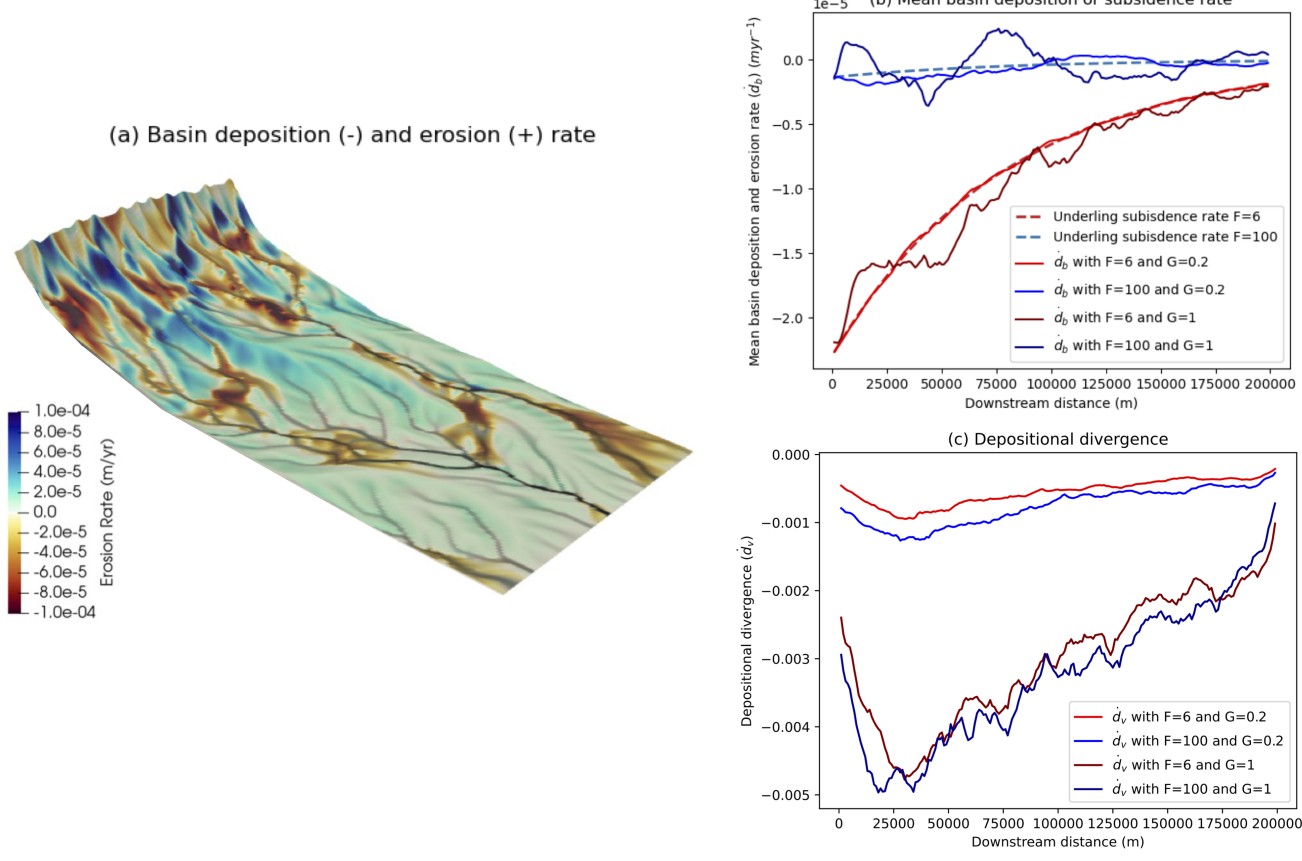

**Figure 5.** a) Example of variations in deposition and erosion rate across the basin at steady-state computed by GravelScape. b) Comparison between the subsidence rate and the computed steady-state deposition/erosion rate averaged in the $y$-direction for different values of the model parameters $F$ and $G$. c) Corresponding values of the depositional divergence, $\dot{d}_v$.

than one. Further analysis shows that when orogen discharge, $\beta$, is high or the system is more 'detatchment limited' (i.e., low G), then the magnitude of depositional divergence and the variability are relatively reduced. When $\beta$ is low or the system approaches transport limited conditions, for both high and low bypass conditions, we see a much greater magnitude of depositional divergence, particularly near the mountain front with a larger variability in the down-system direction.

### 3.2.2 Rugosity ($\eta$)

In Duller et al. (2010)'s approach, deposition is equated with subsidence and no surface topography is neither needed nor computed. On the contrary, in GravelScape, the deposition (and erosion) of sediment is a function the shape of the surface topography as shown by the form of Equation 1. Therefore, any difference in grain size fining trend between the two approaches

is therefore likely to be related to the shape of the surface topography, i.e., its mean slope and the rugosity of the surface. Based on previous work (Braun, 2022), we have already explained how the slope is function of the model parameters ($K$, $G$, $\beta$ and $F$) but little is known about the model controls of the surface roughness (or rugosity) that is caused by internal processes and, in particular, the presence of multiple channels.

To quantify the rugosity, we define a rugosity parameter, $\eta$, as the standard deviation of the topography in the $y$-direction, averaged in the $x$- and over time at steady-state:

$$\eta = <\sigma_y(h)>_{x,SS} \tag{10}$$

Defined in this way, the rugosity parameter, $\eta$, can also be regarded as the average height difference between the interfluves and the channels or as the thickness of the sediment active layer that is reworked (incised and infilled) over multiple steady-state time steps. This is also similar to the concept of the active layer described in (Toby et al., 2022). To illustrate this point, we show in Figure 6 cross-sections in the $y$-direction of the surface topography predicted by GravelScape for an arbitrary model run and time step. We see that at all three locations, the surface topography fluctuates by tens of meters, with the lows corresponding to channels and the highs to interfluves.

### 3.2.3   Links between $\Delta D$ and $\dot{d}_v$, $\eta$, and $S$

We now proceed to further analyse the model runs we have performed with GravelScape using a wide range of model parameters (as shown in Figure 7) by searching for relationships that may exist among the grain size deviation, $\Delta D$ or the depositional divergence, $\dot{d}_v$, and the surface rugosity, $\eta$, the slope, $S$, or $K$ across all model set-ups. Here, the slope, $S$, is the derivative of the surface topography in the $x$-direction averaged over the $x$- and $y$-directions and over time, at steady-state. The results are shown in Figure 7.

In each of the diagrams shown in Figure 7, each model experiment is summarized by a single point, averaged across the entire basin (for the slope) or at the basin outlet (for the grain size). The range of model parameters ($F$, $G$, $\beta$ and $K$) we consider are the same as those used in Figure 2 and 3 and not considering the models affected by local minima.

We see that there exists a strong correlation ($r^2 > 0.75$) between the depositional divergence and the grain size deviation (Figure 7a). This is a direct consequence of the assumption made in Duller et al. (2010)'s model for grain size fining that fining is in proportion to deposition rate relative to sediment flux. This, indeed, implies that where the deposition rate diverges most from the subsidence rate (large values of $\dot{d}_v$), the departure from the single channel model of Duller et al. (2010) is largest. In other words, grain size fining exceeds what is expected from a one-to-one relationship with subsidence in regions of enhanced deposition that is caused by the heterogeneity in deposition rate inherent to a system that transport and deposit sediment in distinct, multiple channels.

In Figure 7b we see that the depositional divergence is, in turn, related to the product of the rugosity by the erodibility. This product, $\eta K$, has units of m/yr (in cases where $n$ in the slope exponent in the SPL is 1) and can be regarded as a measure of the rate at which the rugosity or the side of a channel is eroded away and is therefore a proxy for the rate of the across-system (or interfluve) reworking. Depositional divergence, at steady state, describes the amplitude of erosion and deposition

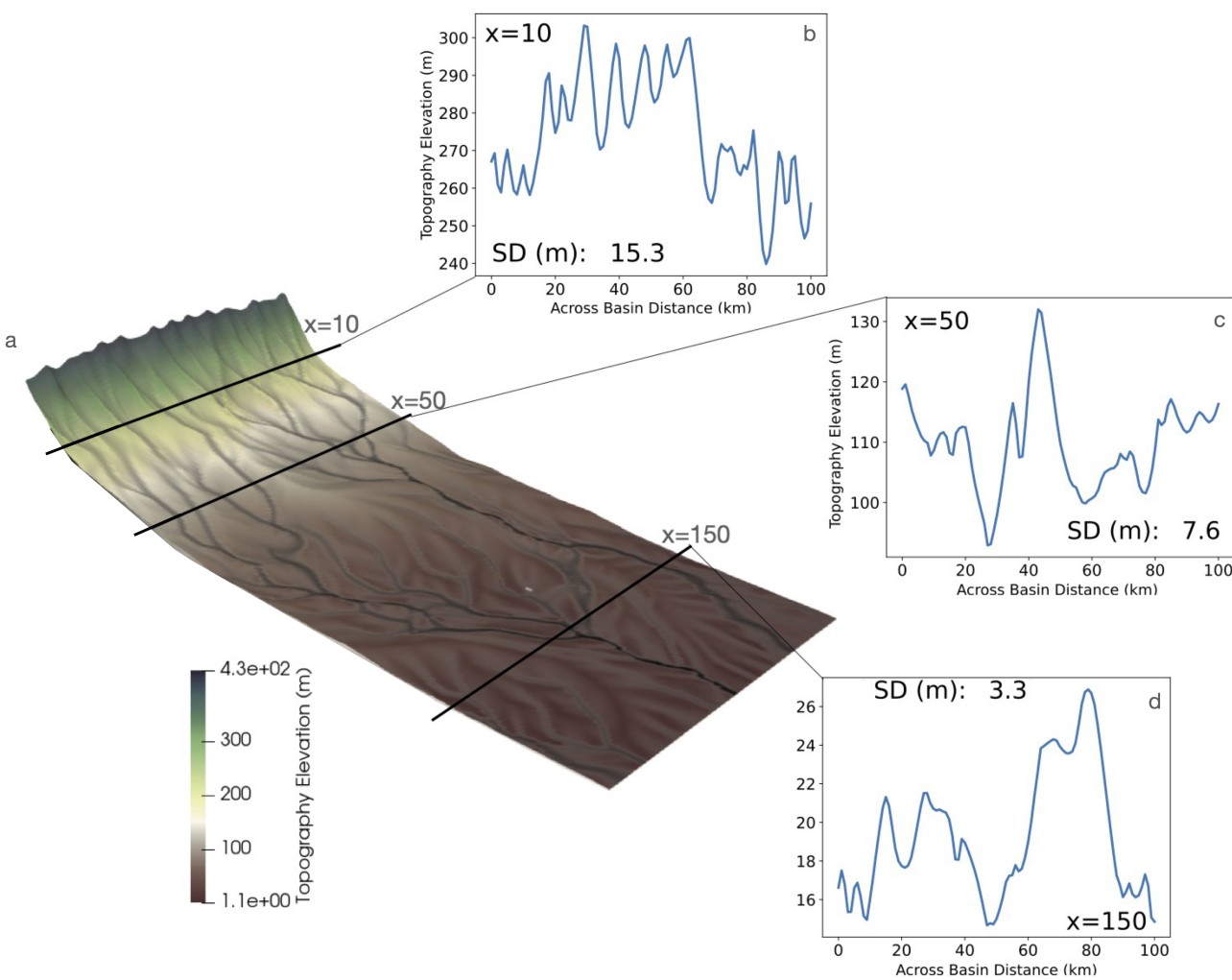

**Figure 6.** a) Example of a GravelScape predicted topography. b), c) and d) topographic profiles at three locations ($x$) distant from the mountain front by 10, 50 and 150 km, respectively. The across basin distance is the $y$ direction in the model.

around the mean rate due to local erosion followed by subsequent local infilling during the next depositional event. When the basin experiences local erosion (e.g., especially under high bypass) depositional divergence therefore controls the rate at which channels change their shape and direction, and can subsequently impact channel avulsion and mobility. The correlation showed on Figure 7b demonstrates that the autogenic processes leading to the depositional divergence are physical (and not random or numerical in nature) as they are directly related to the rate of change of channel geometry, as shown through rugosity, predicted by the basic equation (the modified SPL) at the core of the sediment transport model of GravelScape.

Interestingly, the rugosity itself appears to be strongly correlated with the slope (Figure 7c). This is easily explained when considering that it is the ratio between the slopes along and across a channel that determines the stability of any given channel.
If the across-channel slope becomes smaller than the along-channel slope, an avulsion takes place. As the rugosity controls the across-slope and the slope in the $x$-direction is approximately equal to the along-channel slope, the two should therefore be correlated. This is further proof that the deviation in grain size fining predicted by GravelScape with fluvial channel dynamics result from deterministic, physical reasons and not random (or numerical) artefacts.

Finally, we show in Figure 7d a relationship between grain size deviation and the product of slope and $K$. This is a direct consequence of the relationship between depositional divergence and the product of rugosity by $K$ and the strong correlation between rugosity and slope. This demonstrates that, although the correlation of the grain size deviation is much higher when directly related to the autogenic depositional deviation parameter, higher autogenic grain size deviation tend to occur under landscapes with steeper slopes.

In summary, we have seen that under certain circumstances, namely high bypass efficiency ($F$-values) low orogen discharge efficiency ($\beta$-values) or high depositional efficiency ($G$-values) the grain size fining predicted by GravelScape deviates markedly from predictions made assuming a single channel and a deposition controlled by subsidence only (we refer to this difference as the grain size deviation, $\Delta D$). We have shown (see Figure 7) that grain size deviation is proportional to the contribution to sedimentation from autogenic processes, which we quantified by introducing a depositional divergence factor, $\dot{d}_v$. Furthermore, we have shown that the magnitude of these autogenic processes, especially under high bypass, is correlated to surface rugosity, $\eta$, and slope, $S$.

## 4 Model Synthesis and Discussion

### 4.1 A new generalized framework to interpret grain size fining data

To further synthesise our results and facilitate their use for the interpretation of grain size fining data, we have developed a generalized framework to determine under which basin-wide configurations grain size fining is dominantly controlled by subsidence (i.e., mean deposition rate) or by autogenic dynamics (i.e., depositional divergence). The framework is based on two maps shown in Figure 8a and b, one of the grain size fining (8a) and one of the grain size fining deviation (8b) as a function of two variables, $F$ and $\beta$. These maps are obtained by averaging the results shown in Figures 3 and 2 over $G$ and $K$, respectively. This averaging is justified by the low dependence of these results on $K$ and the fact that the value of $G$ is difficult to assess, but likely to be relatively close to 1 as many alluvial, continental sedimentary systems are predominately, moderately transport-limited (Guerit et al., 2019). In Appendix B and Figure B1, we show maps similar to those shown in Figure 8 but using slope, $S$, along the vertical axis.

In these maps (Figure 8a and b), we define three main regimes for grain size fining based on the grain size divergence computed by GravelScape: 1) a subsidence-dominated regime where grain size deviation is insignificant (under 5%), 2) an autogenic-dominated regime (including transient and steady-state systems) where grain size deviation is larger than the fining induced from subsidence (i.e. that predicted from Duller), and 3) a mixed regime, where grain size divergence is non-negligible

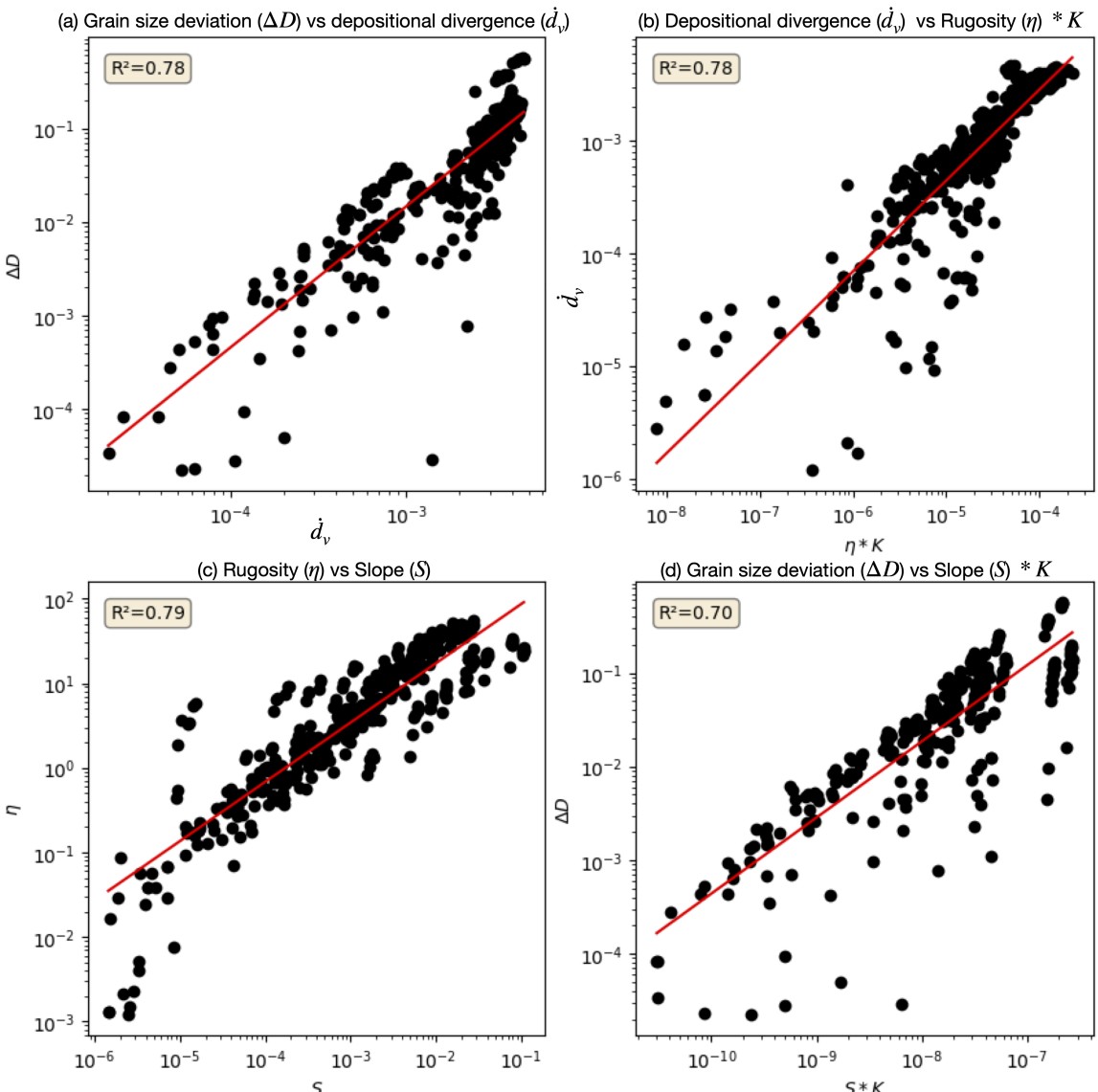

**Figure 7.** Correlations between grain size deviation, depositional divergence, surface rugosity and slope for all model experiments performed in this study obtained by varying model parameters $F$, $K$, $G$ and $\beta$. Model runs affected by local minima have been neglected. a) Grain size deviation, $\Delta D$, against depositional divergence, $\dot{d}_v$; b) depositional divergence, $\dot{d}_v$ against the product of surface rugosity, $\eta$ and erodibility, $K$; c) surface rugosity, $\eta$ against surface slope, $S$; and d) grain size deviation, $\Delta D$ against the product of slope, $S$ by $K$. In each panel, the red line shows the trend of the least square regression in log-space and the resulting coefficient of variation is given in the insert.

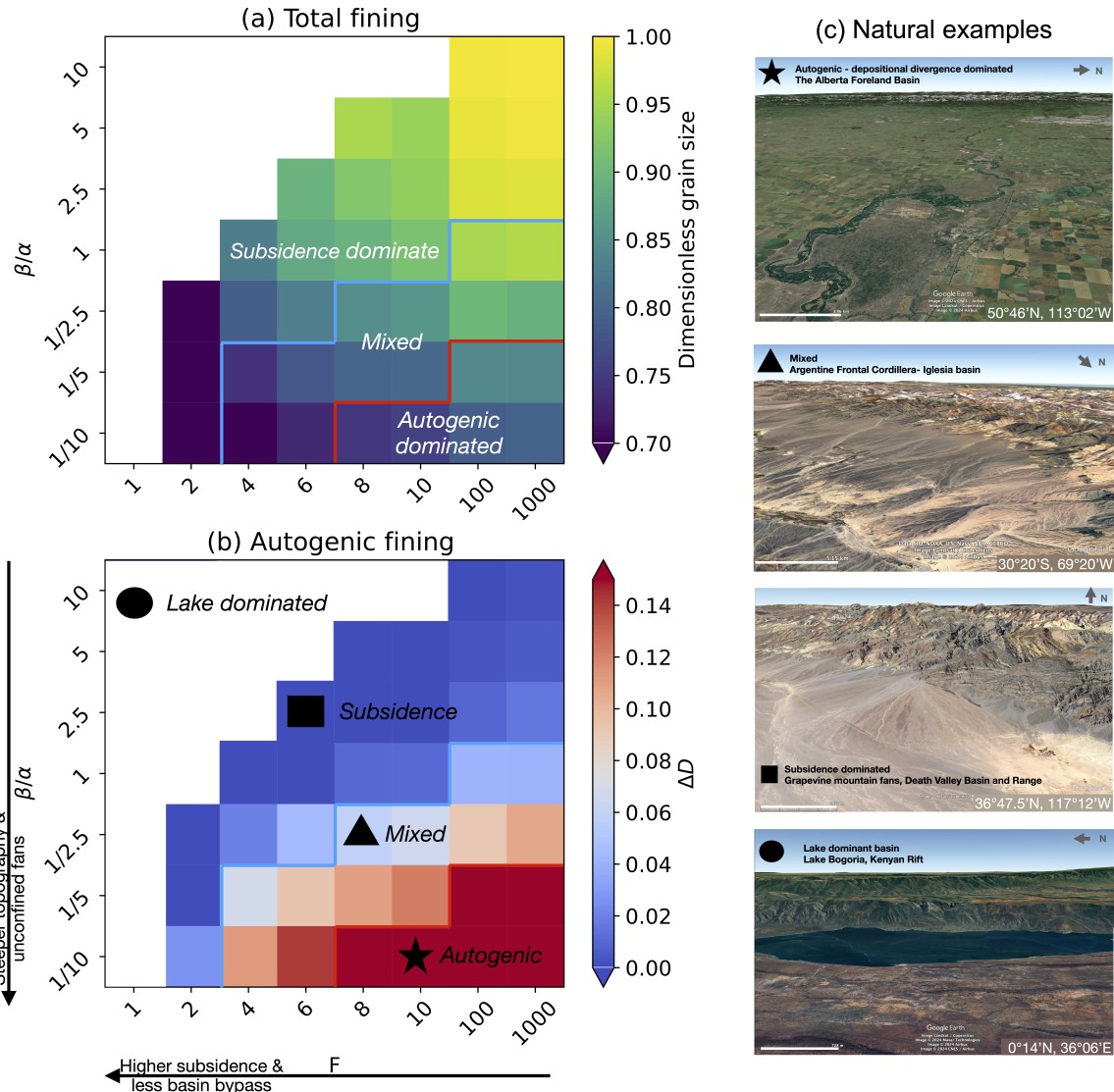

**Figure 8.** Proposed framework to interpret grain size fining trends. Maps of a) total grain size fining and b) grain size deviation as a function of $F$ and $\beta$ and c) select natural examples from the literature that we have located in our framework. Red (star) areas in plot b indicate conditions where autogenic ($\beta$) driven grain size fining is indistinguishable from (see plot a) or exceeds any subsidence ($F$) driven fining. Darker blue (square) areas in plot b indicate that subsidence dominated fining is distinguishable with limited autogenic ($\beta$) influence.

(over 5%), but not greater than the subsidence induced fining. Note that the top left region in both panels a) and b) of Figure 8 corresponds to model parameter values that lead to unrealistically flat topographies such that the solution is dominated by

the presence of numerous local minima. We do not include modelling results in this region in our framework as it represents situations that are inadequate for measuring grain size.

We see that where total fining is larger than 30% (bottom-left corner), the system is consistently in the subsidence-dominated regime. In other words, autogenic grain size fining alone cannot explain fining above 25-30%. These subsidence-dominated systems are in low by-pass and and steep slopes (low $\beta$ values). At the diagonally opposite side of the map, i.e., in systems characterised by high by-pass and low slopes (high $\beta$), there is little to no fining (less than 2.5%, top-right corner) as neither subsidence nor autogenic processes can produce fining. In our new framework, the subsidence-based interpretation for grain

size fining of Duller et al. (2010) is equivalent to a trajectory in the subsidence-dominated regime from these two locations, i.e., from bottom-left to top-right. This means that in order for grain size to be interpreted as function of subsidence only, increasing values of $F$ must be accompanied by decreasing values of slopes or increasing $\beta$.

The presence of an autogenic-dominated regime in the bottom-right corner of the Figure 8 map, corresponding to high by-pass and low $\beta$ (high slopes), has two implications. Firstly, the one-to-one relationship between grain size fining and $F$ breaks

down in systems that are characterized by high slopes. In other words, following a horizontal trajectory in our framework at constant, high slope values, we do not observed the decrease in grain size fining with increasing $F$ due to the importance of autogenic processes. Secondly, high by-pass systems may experience substantial fining in proportion to their slope. This situation corresponds to a trajectory along a vertical path in our framework in the high by-pass regime, where progressive fining corresponds to increasing slopes or an increasing importance of autogenic processes, in systems that are all characterised by

high by-pass.

These findings have important implications for the interpretation of grain size fining. Firstly, there is a danger to interpret grain size fining trends only as a direct measure of present-day or past subsidence only, whereas some of the fining may be due to autogenic processes. To avoid this over-interpretation of data, one should compare the subsidence derived from the grain size fining trend to preserved sedimentary thickness, where possible. Another option suggested by our work is to consider the

365 topography, and in particular, the surface slope. Secondly, we postulate that in high slope, high by-pass system grain size fining is likely to be dominated by autogenic processes and should therefore not be used to constrain subsidence patterns.

## 4.2   Illustrating our framework with natural examples in modern systems

To illustrate the use of the framework, we have compared our modelled sedimentary systems to the natural systems (Figure 8 c) according to their by-pass characteristics and their surface slope and confinement, that we assume indicative of $\beta$.

Extension in the Basin and Range area (Norton, 2011) has lead to high subsidence and accommodation rate in the Death Valley graben (Hammond et al., 2012) resulting in an under-filled sedimentary basin with an active depositional surface that presently lies 86 m below sea-level (Burchfiel and Stewart, 1966) and where large lakes frequently form (Blackwelder, 1933; Grasso, 1996). This indicates low $F$ conditions. The relatively large size of the catchment, especially along the western side of Death Valley, compared to the extent of the fans, as well as the relatively linear slopes of the fans are both indicative of a

$\beta/\alpha$ value near or just above 1, which positions the Death Valley fans in the subsidence dominate regime (the square in Figure

8b). This is consistent with the fining observed by D'Arcy et al. (2017) and interpreted as reflecting the spatial distribution of subsidence suggested by the underlying stratigraphy (D'Arcy et al., 2017).

Megafans exiting the Himalayas (e.g., the Kosi megafan) are of similar extent than the flexure wavelength of the underlying crust resulting in a large $\beta$ values with low slopes. This is confirmed by their convex or linear surface topography (Chakraborty et al., 2010). The resulting low slopes and high $\beta$ are likely to minimise autogenic deviation and positions them in the upper portions of the framework, where there is little fining unless there is subsidence. This is in agreement with Dingle et al. (2016)'s findings of fining trends limited to parts of the Kosi fan characterized by higher subsidence. However, (Dingle et al., 2017) attribute some of the fining observed to abrasion, a process not yet included in our model set-up.

The Iglesia basin is a piggy-back basin fed by crustal shortening and uplift in the Argentine Frontal Cordillera (Beer et al., 1990). The basin is in moderate to high bypass (Harries et al., 2019). The flexurally-controlled subsidence is likely to be much greater than the fan extent leading to small values of $\beta$. This is confirmed by the fan concave up topography (Harries et al., 2019). This should position the system in the mixed to autogenic-controlled regime (the triangle in Figure 8b). This is consistent with the internal reworking described by Harries et al. (2019).

High by-pass systems can also be found near mature orogenic settings such as the Alberta Basin of southern Canada where subsidence and in-filling rate has greatly decreased since the onset of collision in the Jurassic and again in the Cretaceous (Mossop and Shetsen, 1994). The sedimentary fans that form adjacent to the Canadian Cordillera have much smaller extent than the flexural wavelength of the underlying old cratonic lithosphere (Koohzare et al., 2008), implying a small $\beta$ value and high slope. These systems are likely to be in the autogenic-dominated fining (the star in Figure 8b) regime as suggested by the well-documented importance of autogenic processes within the post-glacial fans of southern Alberta Campbell (1998).

For completeness, we also positioned the Kenyan rift as a lake-dominated, under-filled system where deposition is dominated by lacustrine processes in the local minima-dominated regime (the circle in Figure 8b) where the self-similar grain size fining model of Duller et al. (2010) does not apply.

## 4.3 Implications for the stratigraphic record

Past studies (D'Arcy et al., 2017; Whittaker et al., 2011) have used grain size fining trends extracted from the stratigraphic record as a tool to estimate past histories of subsidence in sedimentary basins. This can be effective in situations where topography is low (high $\beta$) or the basin is filling (low $F$), but we have also shown that this can lead to overestimating the subsidence rate in transient basins or systems where autogenic processes contribute significantly to grain size fining. Thus, there is a need to identify when autogenic induced grain size fining is most likely dominating the record.

The general relationship we have evidenced between depositional divergence, rugosity, and slope (Figure 7b and c) is useful for this, as it could be used by field geologists to estimate the magnitude of the autogenic processes controlling the depositional divergence by measuring rugosity (or channel depth). In turn, because we have shown that, in most situations, grain size deviation is proportional to depositional divergence (Figure 7a), a measure of rugosity can be used to estimate whether grain size fining is affected by autogenic processes and/or whether a grain size fining trend can be used to constrain basement subsidence.

A combined approach of estimating stratigraphic thickness ($F$) while also considering paleoslope, assessing reworking, or measuring maximum channel to interfluve depth (rugosity) would be most ideal to indicate the general relevance of autogenic induced fining within the system (because slope and rugosity correlate with grain size deviation as shown in Figure 7b). This means that if a system has high paleo channel depths and slopes, combined with evidence for reworking, it is likely to be strongly influenced by autogenic processes and grain size fining estimates may overestimate the subsidence rate. Conversely, paleo systems with thick stratigraphic packages characterised by low paleo channel depths and slopes with relatively uniform infilling would have grain size fining rates more closely controlled by basement subsidence.

Further consideration should be taken specific to grain size, and its response and recovery to perturbations. Whipple and Meade (2006) have described how sediment flux returns back to the value set by the tectonic forcing after a climate perturbation and that tectonics therefore determine the underlying sedimentary record over long enough time scales and constant conditions. The same trend is predicted for grain size signals (Armitage et al., 2011) with subsidence rate controlling the long-term trend and climate-driven perturbation producing only relatively short-lived deviations from that trend. Our new findings show that, firstly, in high bypass systems, long-term grain size fining can be set by the autogenic dynamics especially in systems characterized by steep surface topography (low $\beta$) and in a transport-dominated state (high $G$). Secondly, we have also shown that, under high bypass, grain size fining becomes a function of $\beta$, which, in turn, is related to the size of the source catchment (or the sedimentary fan) relative to the subsidence pattern (or flexure wavelength), weighted by the relative precipitation rate in the source and basin areas. This implies that variations in precipitation between the basin and catchment (in space or time) could impact the long-term sediment recorded through grain size fining trends beyond a short-lived perturbation.

## 5   Conclusions

Our main findings can be summarised as follows:

- deviations from a subsidence-based interpretation of grain size fining trends are controlled by the intensity of autogenic processes;

- the magnitude of those autogenic processes, measured by introducing the depositional divergence, $\dot{d}_v$, are proportional to surface slope and rugosity, and are therefore the result of a physical process at play within the model and not the result of numerical instabilities;

- different model parameters, namely the shape parameter $\beta$, the bypass parameter $F$, the erodiblity parameter $K$, and the depositional parameter $G$, impact basin grain size fining and only select combinations promote either subsidence or autogenic dominated grain size fining;

- we proposed an averaged framework (Fig. 8) to help interpret grain size fining data that maps grain size fining and deviations from a subsidence-based interpretation of grain size fining as a function of by-pass (or $F$), and slope (or $\beta$);

 – the framework helps defining the conditions for using grain size fining trends to infer subsidence patterns, as well as the conditions where autogenic proesses dominate grain size fining, i.e., high bypass (high $F$) and steep slopes (low $\beta$);

– we have demonstrated its usefulness by positioning various natural systems into the framework and shown how this can help determine whether, for each of them, subsidence or autogenic processes dominate grain size fining.

In the third paper (Wild et al., 2024b) of this series of three, we propose to use the framework and what we have learned from
 the theoretical work presented in the first two to interpret stratigraphic transects in a synthetic foreland basin. For this, we will couple GravelScape to a simple model of the isostatic flexure of the crust/lithosphere. In doing so, we will produce a system where subsidence is in proportion to the weight of the evolving orogen and is therefore in constant transient evolution towards steady-state. The next step will consist in studying, in a source-to-sink approach, the response of such a coupled system to imposed perturbations in climate or tectonic activity as was done previously without considering the effect of autogenic
 processes on grain size fining (e.g., Armitage et al., 2011).

Another obvious extension of our theoretical work will be to use it to interpret grain size data from well-documented sites such as Death Valley, the Himalayan Foreland or the Iglesia basin of Argentina. The work presented here suggests that a joint inversion of the grain size data and topography (slope and extent of the fan) could yield constraints on the value of model parameters (such as $G$ or $K$) and subsequently allow us better assess the contribution of autogenic processes to grain size
 fining before using such data to infer subsidence patterns.

Finally, additional developmental work could involve further exploring sand fining allowing, for example, for bimodal distributions, adding an abrasion component to fining, or incorporating a feedback between grain size and the transport/erodibility parameters of the LEM component of GravelScape.

**Appendix A: Input parameters**

 **Appendix B: Alternate Framework of Slope vs $F$**

The framework maps in B1 are similar to those shown in 8, but with slope, $S$, replacing $\beta$ along the vertical axis. In Figure B1a and b, we used the slope averaged over the entire basin $L_B$. In Figure 8c and d, we used the slope averaged over, $L_f$, the theoretical size of the fan according to Braun (2022), i.e., the size of the upstream mountain catchment $L_M$ weighted by the ratio of precipitation rates in the mountain and in the basin, i.e., $\nu_M/\nu_B$. Grain size fining and grain size fining deviation are
 measured over the same distances, i.e., $L_B$ for panels a and b and $L_f$ for panels c and d.

We considered slope as a general alternative to $\beta$, since there was a similar pattern of grain size deviation in the framework and a general high correlation between slope and grain size deviation. Within the main text, we prioritized $\beta$ configurations as one approach to inducing higher slopes and more autogenically dominated conditions, due to $\beta$' s measurability at the landscape scale. However, our results also showed how transient conditions (lower $K$) and higher $G$ can increase slope and autogenic
 dynamics. With limited subsidence, any initial topography present within the basin could perpetuate increased slope, rugosity, and autogenic fining conditions. However, under high subsidence conditions, impacts of initial topography in a basin would

**Table A1.** Reference GravelScape model parameters. Note that we often tested a range of values. Unless stated on the figures, the model had the following inputs used.

| Parameters | Validation Set-Up (1-cell Orogen and imposed subsidence) |
| --- | --- |
| $K$ | $6.5\ e-6$ (reference) $m^{1-2m}yr^{-1}$ (all assuming mean annual precip. of 1 m/yr) |
| $m$ | 0.4 |
| $n$ | 1 |
| $G$ | 1 (reference) |
| $Diffusion$ | 0.1 |
| $L_M$ | 1 000 m (simplified single cell orogen) |
| $L_B$ | 200 000 m |
| $y$ | 100 000 (GravelScapeMCH) m |
| $\Delta_x$ | 1 000 m |
| $\Delta y$ | 1 000 m |
| $\Delta t$ | 10 000 yrs |
| $\sum time$ | $25e^6$ yrs (steady-state) |
| $U$ | 0.01 m/yr |
| $D_0$ | 1 |
| $\phi_0$ | 0.75 |
| $C_V; C_1$ | 0.75 |
| $\nu_M$ | 20 (reference) |
| $\nu_B$ | 1 |
| $(\nu_M L_M)/(\nu_B L_B)$ | 0.1 (reference) |
| Imposed $\sigma_0$ | -1.36e-5 (reference) m/yr |
| Imposed $\alpha$ | 2.5 |
| Imposed $F$ | 10 (reference) |

likely be rapidly buried, leading to flatter slopes, low across basin topographic variability, and subsidence dominated fining conditions. There are many more scenarios that could impact slope and subsequent autogenic fining conditions that warrant further study.

**Funding Declaration**

This project received funding from the European Union's Horizon 2020 research and innovation program under the Marie Sklodowska-Curie grant agreement No 860383.

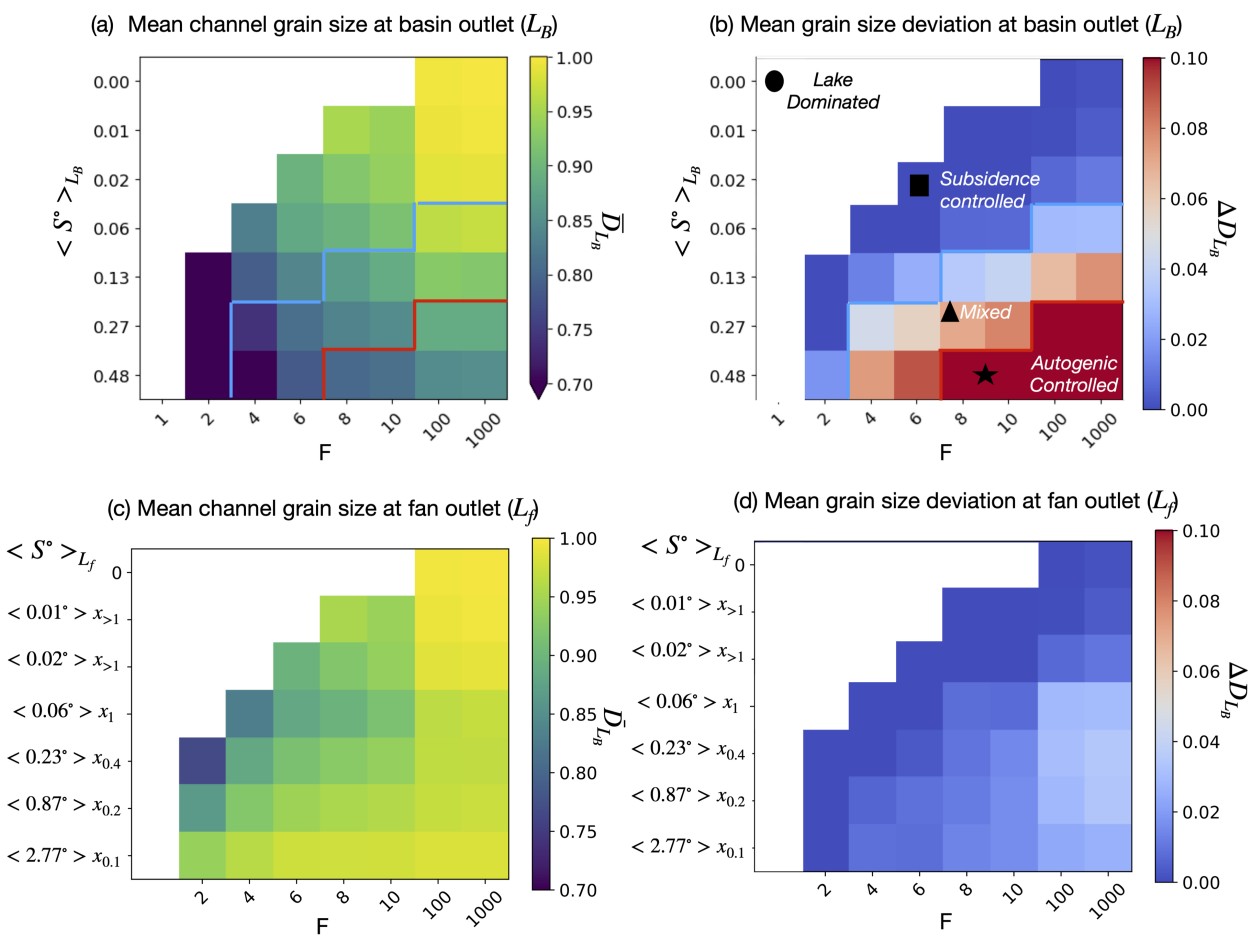

**Figure B1.** Proposed framework to interpret grain size fining trends. Maps of a) total grain size fining and b) grain size deviation as a function of $F$ and $S$. c) and d) Same as a) and b) but averaging slope and computing total fining and deviation at the end of the fan, i.e., at $x = L_f$.

*Code availability.* Python Code for GravelScape is available on demand from the authors. The code will be made available on a public repository upon paper acceptance.

*Author contributions.*

.  Amanda Wild: Conceptualization, formal analysis, investigation, methodology, software, validation, visualization, writing-original draft preparation, writing-review and editing

. Jean Braun: Supervision, resources, software, conceptualization, methodology, visualization, writing-original draft preparation, and writing-review and editing

. Alexander Whittaker: Supervision, conceptualization, methodology, and writing-review and editing

. Sebastien Castelltort: Supervision, conceptualization, and writing-review and editing

. Marine Prieur: Conceptualization, and writing-review and editing

*Competing interests.* The authors declare that there are no competing interests.

*Acknowledgements.* The authors thank Benoit Bovy for general help with xarray-simlab and FastScape curation. We would also like to thank
Charlotte Fillon for her comments during committee meetings and the earlier phases of this research. We would also like to thank scientists within the Earth Surface Process Modelling Section at the GFZ Potsdam and members of the S2S-Future Marie Curie ITN for their general feedback and discussions. The project has received funding from the European Union's Horizon 2020 research and innovation programme under the Marie Sklodowska-Curie grant agreement No 860383.

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

# Grain size dynamics using a new planform model. Part 3: Stratigraphy and flexural foreland evolution

Amanda Lily Wild[1,2], Jean Braun[1,2], Alexander C Whittaker[3], and Sebastien Castelltort[4]

[1] Helmholtz Center Potsdam, GFZ German Center for Geoscience, Potsdam, Germany
[2]The Institute of Geosciences, Universität Potsdam, Potsdam, Germany
[3]Department of Earth Science and Engineering, Royal School of Mines, Imperial College London
[4]Department of Earth Sciences, University of Geneva, Rue des Maraîchers 13, 1205 Genève, Switzerland
[1]**Correspondence:** Amanda Lily Wild (awild@gfz-potsdam.de)

**Abstract.** Within the stratigraphic record, grain size fining has been commonly used to infer subsidence rate and its variability has been interpreted as a signature of external forcing events. We have recently developed a model (Wild et al., 2024a) that predicts grain size fining within a two-dimensional landscape evolution model to predict the effect of autogenic processes on grain size fining. Here, we couple it to a flexural model to predict the stratigraphic evolution of a foreland basin, the distribu-

5 tion of grain size fining, and which of subsidence or autogenic processes dominates in controlling the fining. We show that, throughout its evolution, the foreland basin experiences a gradual increase in the by-pass ratio, $F$, that provokes a gradual shift from subsidence-dominated to autogenically-dominated grain size fining, but also progressively alters stratigraphic preservation. The amplitude, and therefore efficiency, of autogenic processes in controlling grain size fining processes are modulated by the shape of the surface topography that we control by changing the rainfall gradient and extent of the basin confinement compared to the orogen. We also show how the evolution of the basin can be mapped in the framework we recently developed

(Wild et al., 2024b) to interpret grain size fining data. Finally, we demonstrate how the model results and our findings can be used to interpret the stratigraphy and grain size information stored in a real foreland basin, namely the Alberta Basin of Western Canada.

## 1 Introduction

Foreland basins are large scale geological features, i.e., 10 to 100s of kilometers in size, that develop by flexure of the lithosphere in the vicinity of a mountain belt (Beaumont, 1981). Their evolution and stratigraphy are therefore closely linked to the uplift and erosional history of the mountain. The depth of the basin is set by the height of the mountain, while its width mostly depends on the effective elastic thickness (EET) of the lithosphere (Turcotte and Schubert, 2002). The magnitude of the incoming sedimentary flux from the mountain is a function of the uplift rate and erosional efficiency of surface processes

(Beaumont, 1981), which may be modulated by climate and rock strength during landscape evolution (Allen and Densmore, 2000; Armitage et al., 2011; Leonard et al., 2023) . As collision progresses and the mountain grows, the incoming flux and subsidence within the basin therefore change with time (Covey, 1986; DeCelles and Giles, 1996; Catuneanu, 2004) and the basin tends to evolve from under-filled to by-pass conditions (Allen and Homewood, 2009; Allen et al., 2013). Initial topography can

impact the timing of basin infilling and, when initial conditions raise elevation, promote more continental opposed to marine
dominated infilling conditions (Gérard et al., 2023). When sediment flux from the orogen exceeds the rate of accommodation
space by subsidence, the basin widens by flexure under the weight of its own sedimentary fill (Flemings and Jordan, 1989;
Beaumont, 1981; Catuneanu, 2019). This behaviour is mostly applicable to the retro-foreland basin, i.e., which forms on the
stable side of the mountain, while the pro-foreland basin is constantly fed back into the orogen or, in part, into the subduction
channel (Naylor et al., 2008).

This complex evolution of foreland basins is recorded in the depositional facies and thickness of stratigraphy, but also by the
distribution of grain size within the stratigraphic record (DeCelles and Giles, 1996; Duller et al., 2010; Armitage et al., 2011).
Grain size data is often used to describe depositional environment and the by-pass state of the basin. For example, Cant and
Stockmal (1989); Mossop and Shetsen (1994); Poulton et al. (1990) interpreted the evolution of the Western Canada foreland
basin based, in part, on the distribution of grain size. They noted an early transition from flysch (typically finer grained and
related to deeper marine turbites) to molasse (typically coarser grain continental derived sediments) and, later during the more
mature stages of the basin evolution, the outwards migration of the locus of deposition of the coarsest sediments as the basin
progressively fills and transitions into bypass. Allen and Homewood (2009) also note that many foreland basins show a similar
evolution trend towards high bypass and the bypass state of the basin has subsequent implications on grain size fining (Duller
et al., 2010).

Numerical models have been extensively used in the past to study the evolution of foreland basins and reproduce their stratig-
raphy (Beaumont, 1981; Flemings and Jordan, 1989; Sinclair et al., 1991; Johnson and Beaumont, 1995; García-Castellanos
et al., 1997; Tucker and van Der Beek, 2013). Models have also been extensively used to study how foreland basins react to cli-
matic and tectonic events (Allen and Densmore, 2000) and how these external signals are stored in the basin stratigraphy (Allen
et al., 2013). Special emphasis has been put on studying and reproducing the distribution of grain size in the stratigraphic record
(Armitage et al., 2011, 2013), mostly using Fedele and Paola (2007)'s self-similar grain size fining model, which assumes that
fining is proportional to deposition rate, scaled by sediment flux.

However, many models often used a very simple representation of how sediment is deposited in the basin, based on simple
geometrical or mass conservation arguments (Flemings and Jordan, 1989; Armitage et al., 2011), or a simple diffusion equation
(Sinclair et al., 1991). Also, many used a 1D approach (Johnson and Beaumont, 1995; Beaumont, 1981; Flemings and Jordan,
1989; Sinclair et al., 1991; García-Castellanos et al., 1997; Naylor et al., 2008) that prevents the simulation of planform river
dynamics that controls, in part, the amplitude of autogenic processes.

Yet, internal processes can also dominate the stratigraphic record (Scheingross et al., 2020; Hajek and Straub, 2017; Harries
et al., 2019). For example, using a coupled model for river incision in the mountain and sediment transport (diffusion) in the
basin, Humphrey and Heller (1995) showed that periodic oscillations in sediment flux from the mountain may arise that are
recorded in the foreland basin stratigraphy independently of any external (tectonic or climatic) periodic forcing.

In a series of companion papers (Wild et al., 2024a, b), we have developed a new model, GravelScape (the video in the
supplementary materials shows an example model run) that combines a planform landscape evolution model with Fedele and
Paola (2007)'s grain size fining model, and used it to show that local deposition, driven by topographic variations, can alter

grain size distribution, causing greater background fining down-basin driven by autogenic dynamics under select conditions
(high bypass with steep topography). As shown in Wild et al. (2024a, b), autogenic processes also affect the rate of grain size
fining, which can only be properly assessed by two-dimensional, plan form models where the first-order physics of channel
dynamics is properly represented.

Here, we propose to use this model to simulate the evolution of grain size fining and how it is recorded in the stratigraphy
within a multi-dimensional (x, y, and z) orogenic system, taking into account the potential effect of autogenic processes on the
basin stratigraphy and the distribution of grain size. For this we will couple GravelScape to a flexural model that will provide
a tight coupling between uplift and erosion in the mountain and subsidence and sediment deposition in the foreland basin.

The results of this paper will be presented in three parts. In the first part, we will show how GravelScape can be used to predict
the stratigraphy and grain size distribution within a sedimentary basin using an imposed sedimentary flux and subsidence
function. In doing so, we will highlight the relative contributions of tectonics and autogenic processes within stratigraphic
profiles under imposed states of bypass and with different basin configurations that affect its surface topography. These results
will then be used, in the second part, to interpret the predictions of a more complex, fully coupled model run, i.e., which
includes both the mountain and basin areas such that sediment flux and subsidence are no longer imposed but rather result
from uplift and erosion in the mountain and flexure in the basin in proportion to the height of the mountain. In the third part,
we will reinterpret the stratigraphic record of the Alberta foreland basin in Western Canada by using the results of the previous
two sections to demonstrate the importance of autogenic processes in controlling stratigraphy and grain size distribution in a
natural system.

## 2 Methods

In Wild et al. (2024a), we developed a new model, GravelScape, that combines a landscape evolution model, or LEM,
(Fastscape (Bovy, 2021)) to Fedele and Paola (2007)'s self-similar model for grain size fining that assumes that fining is a
function of deposition rate normalized by sediment flux. The LEM predicts erosion in the mountain or source area and depo-
sition and erosion in the basin or sink area according to a modified version of the SPL that considers erosion, and sediment
transport and deposition (Davy and Lague, 2009; Yuan et al., 2019):

$$\frac{dh}{dt} = U - K\tilde{p}^m A^m \left(\frac{dh}{ds}\right)^n + \frac{G}{\tilde{p}A} \int_A \left(U - \frac{dh}{dt}\right) dA \qquad (1)$$

where $h$ is the surface topography, $U$ the rock uplift rate, $A$ the drainage area and $\tilde{p}$ the dimensionless spatial or temporal
variations in precipitation rate from a reference value contained in the coefficients $K$ and $G$. $K$ is the erodibility coefficient, $m$
and $n$ are the area and slope exponents and $G$ is the dimensionless transport coefficient. $t$ is the time variable and $s$ the spatial
variable in the direction of steepest slope, such that $\frac{\partial h}{\partial s}$ is the slope in the direction of water flow. Even though our model can
predict three dimensional stratigraphy, the basic equations it solves are two-dimensional, i.e., depend on only on $x$ and $y$ at
each time step. It is therefore more appropriate to call our model two dimensional that stacks planform solutions in the third
dimension to produce a stratigraphy (z).

In this work, we use two different modeling approaches. First, we impose subsidence and precipitation gradients similar to Wild et al. (2024b) to generate stratigraphy at steady state, and later we compute foreland basin evolution with flexure. In all setups, we use a value of $G = 1$, which corresponds to a moderately transport-limited system with $m = 0.4$ and $n = 1$. In this case, $1/K$ controls the response time of the system. We selected high-magnitude values of $K$ in the range $10^{-5} < K < 10^{-4} \ \mathrm{m}^{1-2m} \ \mathrm{yr}^{-1}$, so that steady state is reached within 20–25 Myr. We also assume a value of $U = 10$ mm/yr and a uniform precipitation rate such that $\tilde{p} = 1$. See Table 1 for a complete list of all other parameters.

**Table 1.** Model inputs and parameter values with definitions in the text or directly in the table.

| Parameter | Imposed Subsidence Setup | Flexure Setup |
|---|---|---|
| $K$ | $2 \times 10^{-5} \ m^{1-2m} yr^{-1}$ | $7.5 \times 10^{-5}$ (Fig.3) to $2 \times 10^{-5}$ (Fig. 4) $m^{1-2m} yr^{-1}$ |
| $m$ | 0.4 | 0.4 |
| $n$ | 1 | 1 |
| $G$ | 1 | 1 |
| $L_M$ (Orogen downstream length) | 1 km | 50 km |
| $L_B$ (Basin downstream length) | 200 km | 200 km |
| $L_y$ (Model across length) | 100 km | 100 km |
| $\Delta x$ (Spatial resolution) | 1 km | 1 km |
| $\Delta y$ (Spatial resolution) | 1 km | 1 km |
| $\Delta t$ (Time resolution) | 10 kyr | 10 kyr |
| $\sum$ time (Simulation duration) | $25 \times 10^6$ yrs (final 5 Myrs steady-state shown) | $20 \times 10^6$ yrs (entire evolution shown) |
| $U$ | 0.01 m/yr | 0.01 m/yr |
| $\overline{D}_0$ | 1 | 1 |
| $\phi_0$ | 0.75 | 0.75 |
| $C_V; C_1$ | 0.75 | 0.75 |
| $\nu_M$ | 20 (Fig.2); 200 (Fig.2) | 0.8 (Fig.3); 10 (Fig.4) |
| $\nu_B$ | 1 | 1 |
| $\beta/\alpha = \dfrac{\nu_M L_M}{\nu_B L_B}$ | 0.1 (Fig.2); 1 (Fig.2) | 0.2 (Fig.3); 2.5 (Fig.4) |
| $h_0$ (Initial topography) | $1 + \left(1 - \frac{\max(0, x-L_M)}{L_B}\right) \times 250$ | $1 + \left(1 - \frac{\max(0, x-L_M)}{L_B}\right) \times 250$ |
| Imposed $\alpha$ (subsidence decay) [1] | 2.5 | – |
| Imposed $\sigma_0$ (initial subsidence rate) [1] | $-5.45 \times 10^{-5}$ (Fig.2C) or $-1.5 \times 10^{-5}$ (Fig.2A,B) $m/yr$ | – |
| $F$ | 2.5 (Fig.2C) or 100 (Fig.2A,B) | Evolves |
| $\rho_a$ | – | 3200 kg/m$^3$ |
| $\rho_s$ | – | 2800 kg/m$^3$ |
| $T_E$ | – | 20 km |

In GravelScape, we compute the flow of water assuming that water can be passed from one node to any of its downhill neighbours on a rectangular grid in proportion to local slope. This allows the formation of multiple channels and the convergence or divergence of flow paths, in response to changes in topography related to erosion or deposition. We have shown inWild et al. (2024b) that this will lead to channel avulsions, the rate of which is controlled by internal and external parameters of the model,

---

[1] See Wild et al. (2024a) or Wild et al. (2024b) for the full equation for imposed subsidence rate as an exponential function of downstream distance.

and, in particular, $K$, the erodibility coefficient. We have also shown that these avulsions leads to local variability in deposition and erosion rate that have an amplitude proportional to surface slope and surface rugosity (a proxy for channel depth).

We coupled *GravelScape* to a flexure model that computes the surface deflection, $w$, caused by isostatic flexure under the weight of topographic variations, $\Delta h$, resulting from uplift, erosion, and sedimentation. This is governed by the following biharmonic equation (Turcotte and Schubert, 2002):

$$D\nabla^4 w + (\rho_a - \rho_s)gw = \Delta h \tag{2}$$

where $D$ is the flexural rigidity, a function of the effective elastic plate thickness $T_e$, $\rho_a$ and $\rho_s$ are the densities of the asthenosphere and surface rocks, respectively, and $g$ is the acceleration due to gravity. The resulting deflection, $w$, is added to the uplift term in Equation 1. In all model simulations, we use constant values of $T_e = 20$ km, $\rho_a = 3200 \, \text{kg/m}^3$, and $\rho_s = 2800 \, \text{kg/m}^3$. We imposed a slight initial topography ($h_0$) in the model to promote continental conditions in the foreland basin, where we can compute grain size (see Gérard et al. (2023) for a description of how initial topography impacts foreland basin evolution).

As explained in Wild et al. (2024a), *GravelScape* also includes a two-dimensional version of the self-similar grain size fining model proposed by Fedele and Paola (2007). In this model, grain size $D$ is a function of a dimensionless variable $y^*$, expressed as:

$$D(y^*) = \overline{D_0} + \phi_0 \frac{C_2}{C_1} \left( e^{-C_1 y^*} - 1 \right) \tag{3}$$

This expression is obtained by integrating the deposition rate normalized by sediment flux along flow paths. Here, $\overline{D_0}$ is the mean grain size in the source area, and $\phi_0$ is its standard deviation. According to the self-similar model, both the mean and standard deviation in grain size decrease proportionally along the flow path. For a detailed explanation of the grain size model, see Fedele and Paola (2007); for its implementation in the two-dimensional landscape evolution model, see Wild et al. (2024a).

Past applications of the self-similar grain size model have assumed that deposition is equal to basement subsidence rate (e.g. Duller et al., 2010; Whittaker et al., 2011). However, Wild et al. (2024a) and Wild et al. (2024b) have shown that this need not be the case when deposition rate is calculated as an independent output of the LEM. In this case, Wild et al. (2024b) have shown that autogenic processes can lead to substantial grain size fining in high by-pass systems where little to no subsidence takes place. They have also shown that the amplitude of this autogenic grain size fining is in proportion to surface slope and rugosity.

In the remaining part of this manuscript, we will use a set of variables and quantities to describe the sedimentary system that were introduced in Duller et al. (2010), Wild et al. (2024a) and Wild et al. (2024b). They include:

- $F$, the by-pass parameter, defined as the ratio of orogen sediment flux entering the basin relative to the total integrated vertical subsidence rate within the basin (Duller et al., 2010);

- $\beta$, the ratio of the size of the sedimentary fan that forms at the foot of the mountain to the wavelength of the basin subsidence function (or flexural wavelength in the case of a foreland basin); the size of the fan is, in turn, controlled by the width of the mountain ($L_M$) catchments feeding the fan, weighted by the relative precipitation rate between the

basin ($v_B$) and the mountain ($v_M$) areas (see Wild et al. (2024b) for an exact definition of $\beta$); Braun (2022) and Wild et al. (2024b) have shown that $\beta$ controls the shape of the surface topography of the basin, with low $\beta$ values producing concave topographies made of a steep and short fan connecting smoothly to an alluvial plain, and large $\beta$ values leading to linear or convex topographies with more extensive fans. Within figures and specific $\beta$ values within the text have been normalized by/excluding $\alpha$ such that they can be directly compared, despite different flexural conditions and $\alpha$, within the framework figures of (Wild et al., 2024b) and Fig. 1. All changes in $\beta$ discussed in the text have been induced by changing orogen precipitation ($v_m$) relative to the basin ($v_b$).

– $\eta$, the surface rugosity defined as the standard deviation of the topography in the $y$-direction (Wild et al., 2024b); it can be regarded as a proxy for channel depth;

– $\dot{d}_v$, the depositional deviation, defined as the mean of the positive departure in sedimentation rate from the basement subsidence rate (Wild et al., 2024b); we have shown that it is a good measure of the amplitude of autogenic processes caused by channel avulsions;

– $\mu$, the channel mobility frequency, that we defined here as the number of times the largest channel has changed its position between two time steps over a given length of time, $\tau = 4 - 5 Myr$, divided by the number of time steps over the same period; a mobility frequency of 1 entails that the channel moved at every time step and a frequency of 0 entails no movement over the period $\tau$.

Finally, (Wild et al., 2024b) introduced a framework to facilitate the interpretation of grain size fining data. It explains under which conditions basement subsidence or autogenic processes dominate grain size fining. This framework is summarized in Figure 1 as a phase diagram in the $F$-$\beta$ space. Four domains have been defined, including (1) an autogenic-dominated domain (in black color) at high $F$ (high bypass) and low $\beta$ (high slope) values, (2) a subsidence-dominated domain (middle grey color) in a band that goes from low $F$, low $\beta$ values to high $F$, high $\beta$ values, (3) a mixed regime, where both subsidence and autogenic processes equally contribute to fining, that is comprised between the first two, and (4) a region in the upper-left corner of the parameter space (low $F$, high $\beta$ values) where surface topography is so subdued that local minima (or topographic depressions) form such that the self-similar grain size fining model is not applicable.

## 3 Imposed Subsidence Stratigraphic End Members

In figure 2, we show the stratigraphic grain size profiles predicted by GravelScape for three model experiments with an imposed incoming sedimentary flux and an imposed exponential subsidence function, as done in (Wild et al., 2024a, b). These plots differ by the value of the parameters $F$ and $\beta$. In panel A, the system is in high by-pass ($F = 100$) and has a concave topography ($\beta/\alpha = 1/10$). In panel B, the system is in high bypass ($F = 100$) and has a convex topography ($\beta/\alpha = 1$) where the orogen drainage and fan area dominates over the entire basin. In panel C, we show a basin in low by-pass ($F = 2.5$) with a concave topography ($\beta/\alpha = 1/10$). In each panel of Figure 2, we show contours of predicted grain size values along a vertical cross

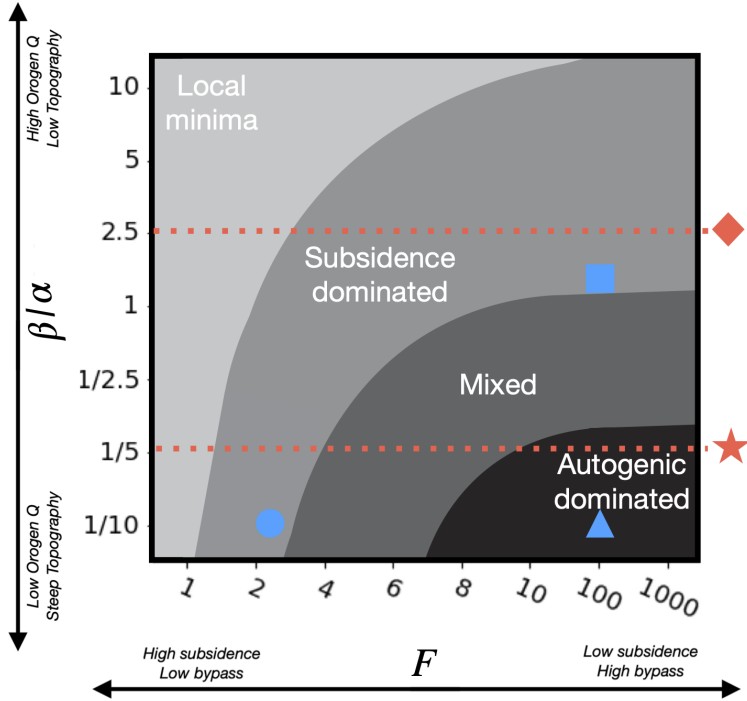

**Figure 1.** Conceptual framework developed in Wild et al. (2024b) in the $F$-$\beta$ space to express under which conditions autogenic processes, subsidence or local minima dominate the grain size fining rate. Blue symbols represent model experiments shown in panels A (triangle) ,B (square), and C (circle) of Figure 2. The lower red dashed line (with the star) corresponds to the path followed by the system in the flexure-driven model experiment shown in Figure 3. The upper red dashed line (with the diamond) corresponds to a similar experiment performed with a high $\beta$ value and shown in Figure 4.

section from the mountain front to the system base level, along the center of the model (sub-panel 1); along the entire surface
of the model at the last time step (sub-panel 2); and along two vertical cross-sections (sub-panels 3 and 4) perpendicular to
the first one at two locations along the $x$-axis as indicated by red lines in the first cross section and the top surface view. All
stratigraphic cross-sections in Figure 2 cover time steps (and thus depth) from the final 5 Myrs after steady-state conditions
have been reached and do not show the underlying bedrock. When generating stratigraphic profiles, we have plotted grain
size values only in 'major channels', i.e., defined as where discharge (or drainage area) is greater than a threshold value ($A_c$)
and filled the remaining areas or 'floodplains' in dark blue. This is done to emphasize major channels, channel mobility, and
channel reworking. In the plan views at the last time steps (sub-panels 2), channel mobility can be estimated from the shape of
the channels and the proportion of the basin covered by flood plains. Within the stratigraphic profiles that cut across the basin,

rugosity (across basin variation in topography) can be estimated by considering surface topographic relief and the amplitude of the cross-cutting patterns of infilling and incision within the across basin stratigraphy.

When vertical accommodation is minor (Figure 2A and B), stratigraphic results show generally less ($< 30\%$) grain size fining within the channel and much less stratigraphic thickness relative to Figure 2C since there is less overall accommodation/sedimentation. Sedimentation that does occur is controlled by local lateral and longitudinal ($x$ and $y$) accommodation within the topographic profile that responds over time to pulses of incision and deposition leading to strong depositional divergence. Rugosity, cross-cutting, and interfluve reworking of channels within the stratigraphic packages is high (Figure 2A3). Comparing Figures 2A and B, we note that topography, rugosity and cross-cutting are greater with lower $\beta$.

When subsidence dominates the stratigraphy (Figure 2C), it controls the rapid ($> 30\%$) grain size fining rates within the channels (Figure 2C1 and C2), preservation is high (thick stratigraphic layers in Figure 2C1), the across basin rugosity is low (no reworking, little topographic variation, or incising/cross-cutting of channels in Figure 2C3 and C4), and the channel mobility is high, producing many course grain channels that deposit ample sediment and are well preserved (Fig. 2C3). Relative to the high bypass example, the final topography in the fan is only marginally depressed (the apex is at 250 m in panel C compared to 350 m in panel A) despite the much higher subsidence rate, demonstrating that the shape of the topography is controlled by the size of the upstream catchment and the flux of sediment coming out of the mountain and, to a much lesser degree, by the subsidence, as shown in Braun (2022).

Wild et al. (2024b) have shown that grain size fining is dominated by autogenic dynamics under high bypass (high $F$ values) when surface slopes are high (low $\beta$ values) such as in Figure 2A or by subsidence under low bypass (low $F$ values) such as in Figure 2C, in agreement with previous studies (Duller et al., 2010; Whittaker et al., 2011; Armitage et al., 2013) where autogenic processes were neglected. High by-pass (high $F$ values) and low surface slope systems (high $\beta$ values) such as in Figure 2B see less or no grain size fining. The results presented here therefore confirm that our new model produces stratigraphy records that are in accordance with the basic findings of Wild et al. (2024b).

They also illustrate the extent to which subsidence is required to preserve grain size fining information in stratigraphy. In high by-pass systems ($F \gtrapprox 100$), time is highly condensed in the stratigraphic column (Figure 2A) and the patterns of deposition are mostly controlled by autogenic processes rendering the preservation of external signals very unlikely. This can be appreciated by considering the surface rugosity in Figure 2 A3 and 4 that is of the same amplitude as the thickness of the layer deposited over the last 5 Myr of the model run. The rugosity can be seen as a proxy for the depth of the active layer and thus indicates that the entire sediment package deposited during the last 5 Myr is still being reworked and that any temporal information about the evolution of the system over these last 5 Myr has been lost. To the contrary, the surface rugosity, and thus thickness of the active layer, in experiments shown in Figure2B and C is much smaller than the thickness of the sedimentary package deposited over the last 5 Myr. In these situations, the time resolution of the stratigraphic record is much finer and can be estimated as the ratio of the rugosity (approximately 5 m and 50, respectively) to the thickness deposited (30 and 900 m, respectively) multiplied by the time span of deposition (5 Myr), which gives approximate temporal resolutions of 800 kyr and 300 kyr for experiments in Figure 2B and 2C, respectively.

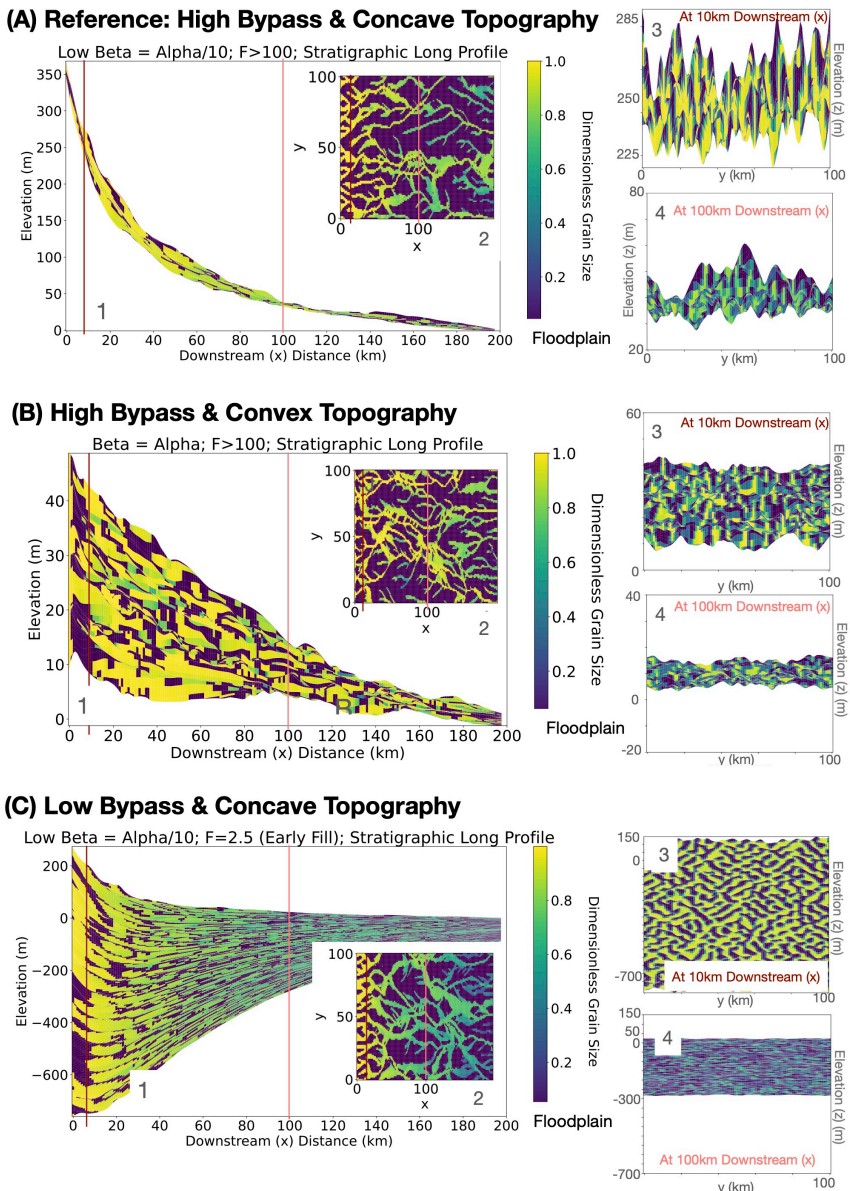

**Figure 2.** Model results with imposed subsidence. Stratigraphic section computed through the basin and along its surface under (A) high bypass and with a concave topography, (B) high bypass and with a convex topography, and (C) low bypass and with a concave topography. In each panel, predicted grain size is shown along a vertical section in the $x$-direction (section 1) and two sections in the $y$ direction (sections 3 and 4) as well as along the model final topography (inset labeled 2). Grain size is shown where large channels were actively depositing, elsewhere a dark blue color is used to represent what we will refer to as flood plain areas. Wherever large channels were not depositing coarse gravel grains during a given time step was filled in dark blue as floodplain. All model parameters are given in Table1

.

This model confirms that preservation of potential external signals is optimal or that the relative amplitude of autogenic processes is minimal in systems characterized by fast subsidence (high $F$ values) and low surface slope/rugosity (high $\beta$ values). This is in line with the theoretical predictions of Ganti et al. (2011) derived from scaled laboratory experiments.

## 4  Fully coupled foreland basin evolution

### 4.1  Flexural foreland evolution under a low $\beta$ set-up

We now present results of the fully coupled model, i.e., combining GravelScape with the flexural model and considering an evolving orogen and its retro-foreland basin. Although we only plotted the basin evolution from the mountain toe (0 km) to 200 km, we also modeled the evolving orogen to calculate the flexural response of the basin dynamically. For the main focus of this paper (figure 3), the value of the EET, $T_e = 20$ km (appendix Table A-1), that we have chosen is such that the wavelength of the resulting flexural defection ($\approx 200$ km), is much greater than the size of the fan. The size of the fan ($\approx 20$ km) is set by the size of the orogen (50 km) and the relative precipitation product in the orogen (0.8) compared to the basin (1) (Braun, 2022). In our model setup, annual precipitation is incorporated into the definition of $K$, so relative precipitation values are dimensionless factors that determine the proportion of precipitation above or below the mean annual rate falling in the basin versus the orogen. In this situation, the surface topography is mostly controlled by surface processes and only by the subsidence at very low $F$ values (Braun, 2022). This leads to a relatively small value for $\beta/\alpha = 1/5$ (see (Wild et al., 2024b) for an exact definition of $\beta$), which is typical of many foreland basins, with the exception of mega fans, which could extend beyond the lithosphere flexural wavelength, or in situations where the mountain is very close to an ocean (such as in the Southern Alps of South Island, New Zealand). This choice of a low $\beta$ value leads to relatively high surface slopes in the basin, which helps prevent the formation of local minima or lake depressions, that affect the grain size computations as discussed in (Wild et al., 2024a, b). The choice of low $\beta$ also allows for autogenic dynamics to potentially impact grain size fining as predicted in the (Wild et al., 2024b) framework (also shown in figure 1) that were otherwise reduced in high $\beta$ settings.

In Figure 3 we present a series of stratigraphic sections through the foreland basin for the duration of the model. The various panels in Figure3 show: a) the stratigraphy where colors are proportional to the value of $F$ at the time of deposition, b) the grain size in the main channel, c) the grain size along a vertical section at the center of the basin, d) deposition or erosion averaged in the $y$-direction and e) the surface rugosity, at the time of deposition. The dark blue areas (low grain size) in panel c correspond to regions where deposition took place along channel interfluves or what can be associated with flood plains. In two small insets, we show, in panel (a), the time evolution of the maximum topography (apex) in the orogen and in the basin, and, in panel (e), the mobility frequency computed over five time intervals in the basin evolution as a function of $x$. Within the inset of panel a (a2), we also show the e-folding time towards topographic steady-state in the orogen and in the basin. These times are also indicated in all stratigraphic sections of panels a to e as a blue (mountain steady-state) and orange line (basin steady-state). The channel mobility shown as an inset in figure 3e was calculated as the frequency of movement of the largest channel over 4 Myr time windows. These time intervals are also indicated as dashed black lines in the stratigraphic sections. In the following description of these results, we have divided the foreland basin evolution into four phases based primarily on its

degree of evolution towards steady-state and, in particular, the bypass parameter, $F$. Note that exact value for $F$ during each stage of the basin evolution may vary depending on the basin configuration. However, the general trend for $F$ is independent of model parameters as is its effect on the evolution of grain size trends and autogenic dynamics.

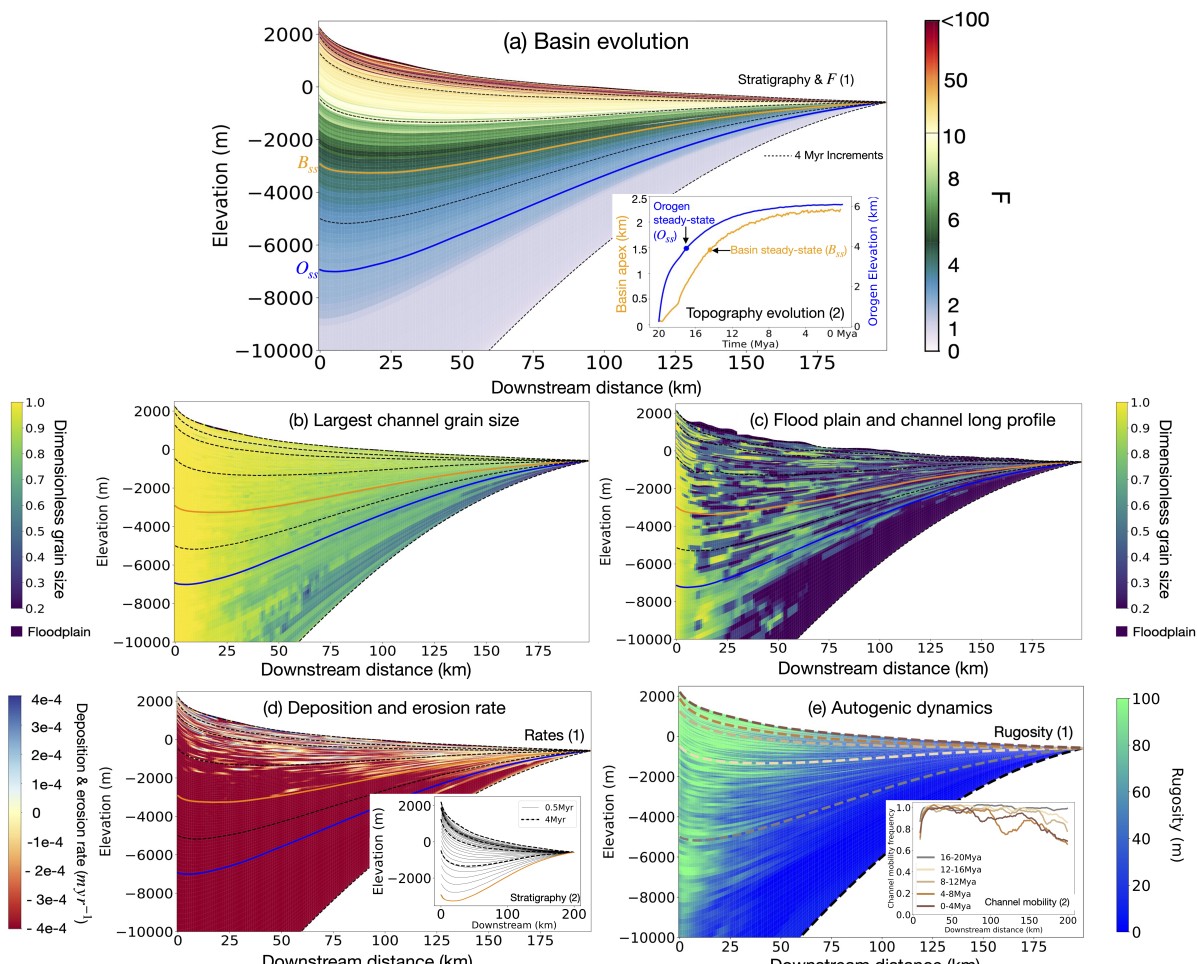

**Figure 3.** Model results using fully coupled model. Vertical cross-section through the model stratigraphy computed at the last time step of the 20 Myr model duration. a) Computed $F$ value; b) computed grain size deposited along the main channel; c) computed grain size at a given cut ($y = 10$) section through the basin (dark blue areas correspond to flood plains); d) deposition rate averaged in the $Y$-direction and e) rugosity. In all panels, the blue line correspond to the time when the mountain height approaches its steady-state value; the orange line is the equivalent for the basin maximum height and the black dashed lines correspond to time markers every 4 Myrs in the evolution of the model. In the small inset of panel a we show the time evolution of the maximum topography in the mountain and in the basin. the small inset of panel e, we show the computed channel mobility frequency as a function of the $x$-position for different time intervals as indicated. See Table1 for model input parameters.

Early in the collision, during what we shall call Phase 0 (light blue-white section with $F < 1$ in Figure 3a), the orogen grows, basin subsidence is fast but little sediment is produced yet; this leads to a mostly 'starved' foreland basin. During this phase, the basin is so underfilled that it becomes a large local depression. In this situation, our algorithm to solve the modified SPL cannot be used and sediment is deposited uniformly across the basin, simulating deposition in a lacustrine environment where the self-similar grain size fining model cannot be used. In other real life settings, this would be a marine basin in to which turbidites or marine stratigraphy might be deposited.

During phase 1 (darker shades of blue for $1 < F < 4$ in Figure 3a), the basin and orogen are in a transient state on their way to the steady-state topographic height, and the subsidence is largely driven by flexure in response to the loading of the growing orogen. The growth of the orogen and of its mean slope result in a steadily increasing erosion rate in the mountain area, and sediment flux into the basin. This leads to a progressive increase in $F$ to values that rapidly exceed 1, as the basin fills and enters a by-pass state. As $F$ increases, grain size fining in the main channel shows coarsening upwards, i.e., less fining (Figure 3b). During this transient phase towards steady-state, rugosity is relatively low, except in the fan area, but channel mobility is high across most of the basin (Figure 3e). Deposition rate is relatively uniform across the basin in response to the rapid subsidence (Figure 3d). This generates thick stratigraphic packages (approximately 15km and 7km deposited in the first two 4 Myr intervals). The strong fining trend that is observed during most of this phase is indicative of a system where sedimentation rate is dominated by subsidence ($F < 4$) and the effect of autogenic processes is minimal (low rugosity) (Wild et al., 2024b). During phase 1, the system can be position in the low $\beta$ vs low $F$ (bottom left) corner of Wild et al. (2024b)'s framework (Figure 1).

We define phase 2 (shifting from blues to greens for $4 < F < 9$ in Figure 3a) as the period when topographic steady-state in the mountain has been reached (the blue line in the stratigraphic sections) until topographic steady-state in the basin has also been reached (the orange line in the stratigraphic sections). During this phase, basin subsidence transitions from predominantly driven by the load of the mountain to the weight of the sediment itself. In this maturing phase of evolution, the system transitions into a progressive higher by-pass regime in which the value of $F$ increases to very large values. Despite this increase in $F$, we see that the fining trend in the main channel remains relatively constant (Figure 3b). We also see an increase in flood plain deposition (Figure 3c), an increase in rugosity and a decrease in channel mobility (Figure 3e) that indicate that autogenic processes become more important. This is confirmed by an increase in variability in deposition rate (Figure 3b). This strongly suggests that grain size fining is becoming progressively dominated by autogenic processes and less by subsidence. During phase 2, the figure 3 system can therefore be positioned in the low $\beta$ vs.intermediate to high $F$ region or mixed regime of Wild et al. (2024b)'s framework (figure 1).

Phase 3 is defined as the period after basin topographic steady-state has been reached (yellow to red contours corresponding to $F > 10$ values in Figure 3a). With a constant flux from the mountain, the subsidence generated due to sediment loading continues but at a much slower pace than in phase 2. The level of by-pass increases with time (less than a few hundred meters of sediment are deposited in every 4 Myr intervals), which explains the rapidly decreasing subsidence rate (less sediment deposited cause less subsidence). During this phase, erosional unconformities can be observed that can extend across the entire

basin or occur locally despite constant external conditions. These unconformities are better expressed in the model run shown in figure 4 with higher precipitation in the mountain area (and thus higher $\beta$ and higher $F$ bypass values).

During phase 3 (high bypass), grain size in the main channel is relatively coarse but some fining still takes place (Figure 3b).
Channel mobility is lower than during phases 1 and 2 (Figure 3e) resulting in a larger proportion of floodplain areas (Figure 3c). Channels also appear to cut into material previously deposited in flood plains (Figure 3c). This and the very high rugosity (close to 100 m) (Figure 3e) indicate higher reworking. Phase 3 also shows the highest variation in deposition rate in the upper reaches of the basin, i.e., near the fan (Figure 3d). All these factors indicate that autogenic processes are very active and of high amplitude. With high by-pass (high $F$) conditions, this suggests that the observed low to moderate grain size fining has become
totally autogenic-dominated. During phase 3, the system should be position in the low-$\beta$ vs.high $F$ (bottom-right) corner of Wild et al. (2024b)'s framework (also shown in figure 1).

## 4.2    Flexural foreland evolution under a high $\beta$ set-up

In Figure 4, we show the results of a model run identical to the model run shown in Figure 3 except for a higher $\beta$ value, obtained by increasing the precipitation rate in the mountain area to be ten times larger than in the basin area, such that $\beta = 2.5\alpha$, and a
slightly lower $K$ ($2e^{-5}\ m^{1-2m}yr^{-1}$), in order to produce a more comparable response time to figure 3 despite the increasing precipitation. Consequently, the size of the fan is larger than the wavelength of the resulting flexural defection and would extend beyond the model length, similar to a megafan or confined system as described in Braun (2022). Note that, based on (Wild et al., 2024b)'s framework (Figure 1), a $\beta/\alpha = 2.5$ model experiment should be subsidence dominated, regardless of the value of $F$.

We see (Figure 4) that the timing of phases 0 to 3, as defined for the low $\beta$ case, remain the same, except that the $F$ values at steady state are much higher in Figure 4 compared to Figure 3. Due to the enhanced precipitation rate in the orogen, the basin more rapidly reaches a state of high-bypass compared to the low $\beta$ scenario, but the sediment thickness accumulated in the basin is substantially reduced. Furthermore we see, in Figure 4B, that channel fining occurs early in the basin evolution, i.e., when $F < 10$, or early in phase 1. Once bypass is reached ($F > 10$) in the high $\beta$ scenario, i.e. during phases 2 and 3,
no further channel fining is observed (Figure 4B) and channel mobility (Figure 4E) is reduced. This is in contrast to results shown in Figure 3 where downstream fining in the channel is observed under high bypass $F$ values and downstream channel mobility occurs. Around and after orogen steady-state is reached, pulses of deposition and erosion result in unconformaties and stratigraphic layers that are highly truncated downstream (Figure 4D). In the final stages of the model simulation the main depositional depocenter migrates towards the downstream areas of the model.

## 5    Application to a case example: the Alberta Basin

Equipped with the results of this fully coupled numerical experiment (example in figure 3) in an unconstrained basin (low $\beta$), we now propose to re-analyze the stratigraphy of the Alberta foreland basin in Western Canada that has already been extensively studied and modeled (Catuneanu, 2004; Mossop and Shetsen, 1994; Leckie and Smith, 1992; Beaumont, 1981), to

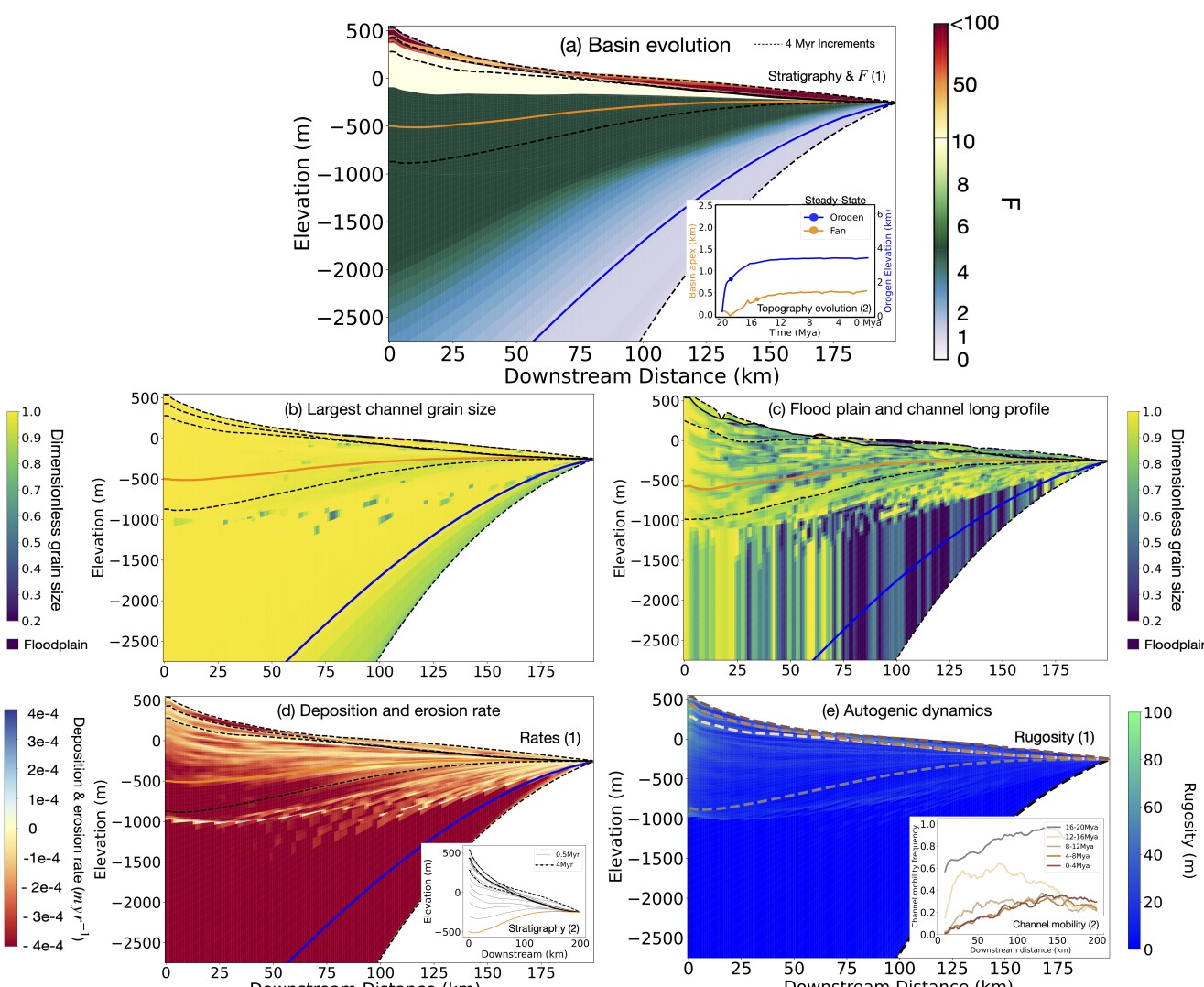

**Figure 4.** High $\beta/\alpha = 2.5$ ($\beta > \alpha$) foreland basin that falls within the local minima (low $F$) to subsidence dominated regime (Fig. 1). Panels show: basin evolution of $F$ (a), largest channel grain size (b), computed grain size at a given cross-cut through the basin ($y = 10$) (c), deposition and erosion (d), e) autogenic dynamics such as rugosity (e1) and channel mobility at 4Myr intervals (e2). In all panels, the blue line correspond to the time when the mountain height approaches its steady-state value; the orange line is the equivalent for the basin maximum height and the black dashed lines correspond to time markers every 4 Myrs in the evolution of the model. Early periods (e.g. around $F < 2$) of basin evolution are likely impacted by local minima. See Table1 for model input parameters.

assess whether we can use the framework developed in Wild et al. (2024b) to determine the relative importance of subsidence

vs autogenic processes in its stratigraphy and the potential impact it has had on grain size distribution within the basin.

## 5.1 General Evolution

The Alberta Basin formed by lithospheric flexure associated with the accretion of the Intermontane Superterrane against western Canada in the Jurassic to form the early Rocky Mountains/Columbian Orogen (Mossop and Shetsen, 1994). This basin should be an ideal example of a foreland system that has experienced a transition from subsidence- to autogenic-dominated control due a number of factors: 1) it is a retro-arc foreland basin (Price, 1974), which, therefore, according to Naylor et al. (2008) would tend to fill and reach high bypass over time; 2) it is underlain by a relatively thick elastic plate (the strong North American craton) (Gordy and Frey, 1977) that results in a very long flexural wavelength (Flemings and Jordan, 1989; Flück et al., 2003) and, therefore, a basin that is much wider than the orogen resulting in a small $\beta$ value; 3) the basin has been evolving gradually for well over 100 Myr (Price, 1974); 4) the orogen lithology consists of several erodible sedimentary terranes (Monger and Price, 2002), that are likely to generate high initial sedimentary flux. However, contrary to our synthetic example described in the previous section, the Alberta basin has experienced periods of both subsidence and uplift, and, recently, has been subjected to periods of rapid glacial isostatic adjustment (Mossop and Shetsen, 1994) causing non-monotonous variations in $F$.

It has been well described that the collision began much earlier in southern Alberta than in northern Alberta (Mossop and Shetsen, 1994). Consequently, at any given time in the evolution of the Alberta Basin, its northern and southern parts may have been in different states of by-pass (or characterized by different $F$ values), and, therefore, showed different level of autogenic vs.subsidence grain size fining during the same time period. We have focused our comparison of model predictions to the general evolution of the system in mid-southern Alberta (between Swan Hills and Cypress Hills area) from the mid-Jurassic onwards generally avoiding issues arising from the three dimensional nature of the collision or pre-existing basement structures, namely the underlying Peace and Sweetgrass arches (figure 5).

In Figure 5, we show a stratigraphic cross-section across the basin, as well as paleogeographic maps for selected geological times. Within the upper Jurassic (Figure 5 A (dark green) or B6), central-southern Alberta exhibited subsidence in response to early mountain building and was inundated by a shallow seaway accumulating thick marine shales. This would likely represent phase 0 in our model where the basin is under-filled (low $F$), its surface is below base-level, coarse fluvial sediments do not propagate very far, and the grain size fining regime is largely non-fluvial and therefore not subsidence-dominated.

After an initial phase of mountain building, there is a well-defined erosional unconformity, which corresponds to a tectonic quiescence and rebound episode in the Canadian Cordillera from 140 to 125 Ma (Leckie and Cheel, 1997). Previous work suggests that long-wavelength uplift may have occurred due to unloading of the orogenic front (Johnson and Dalrymple, 2019). Following this unconformity is an extensive alluvial facies unit called the Cadomin Formation, made of clast-supported, subrounded, dominantly coarse conglomerates (with local sandstone occurrences). It is hundreds of kilometers in spatial extent within a fluvial system flowing northward (Johnson and Dalrymple, 2019; Leckie and Cheel, 1997) to eastward (McLean, 1977) for 12–18 million years during roughly the Barremian–Aptian (White and Leckie, 1999) (Figure 5A, above the dark green, or B). As evidenced by the erosional unconformity at the base of the Cadomin Formation and the propagation of coarse fluvial grains (gravel conglomerate with interbedded sands and occasional finer clasts (McLean, 1977)) away from the orogenic front,

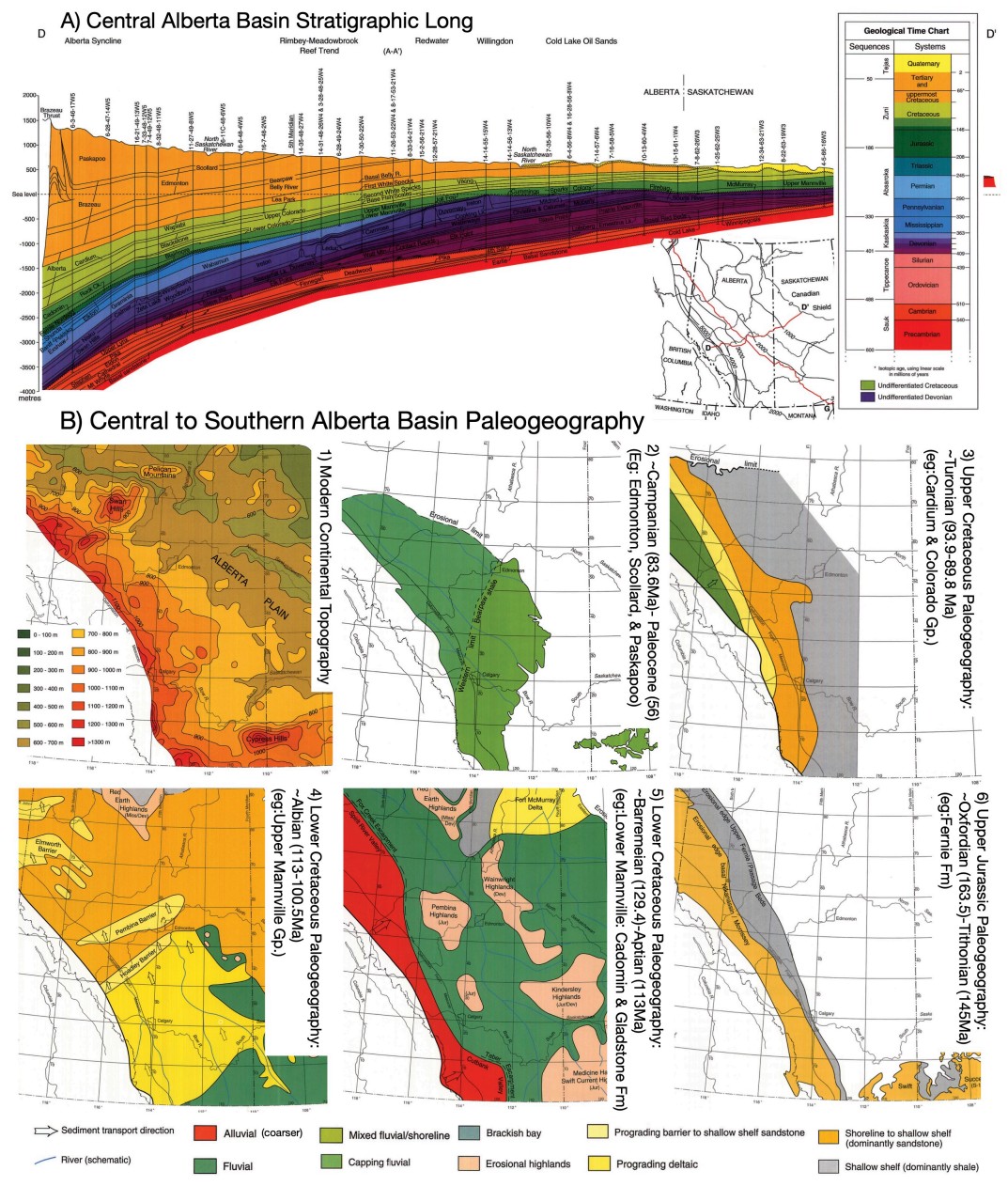

**Figure 5.** Images were compiled and modified from the Atlas of the Western Canada Sedimentary Foreland Basin (Mossop and Shetsen, 1994) under the Open Alberta Licence (*https://open.alberta.ca/licence*). Panel A shows shows the stratigraphic thicknesses and ages perpendicular (D to D') to the orogen and 'downstream' ($x$) within the central Alberta foreland system south of Edmonton. The exact stratigraphic D-D' transect location is shown on an inset map of western Canada in subset A. Subsets B show the modern topographic elevation (1) or paleogeography (2-6) with fluvial propagation vs marine inundation extents at several time intervals between the upper Jurassic (Figure B6) to present(Figure B1).

it is likely that during this phase, the basin had rapidly entered a state of high bypass and, thus, a highly autogenic-dominated regime (e.g., $F > 10$). The Cadomin Formation has a thickness of 1–200 m (McLean, 1977) accumulated over 12–18 Myr (White and Leckie, 1999), implying an extraction rate of approximately $1.5 \times 10^{-5}$ to $5.5 \times 10^{-8}$ $(m/yr)$, which we would consider low within our model simulations and indicative of a state of high bypass (e.g., Figure 2A had a subsidence rate of $1.5 \times 10^{-5}$ $(m/yr)$, resulting in $F \geq 100$).

Following this tectonic quiescence, mountain building reactivated in the Cretaceous when the Insular Superterrane collided with North America and reactivated the eastward thrusting of the Intermontane Superterrane. Subsidence resumed as well. Throughout the Albian, there are varying occurences of brackish bay, delta, and shallow marine shelf deposits, while any fluvial propagation of coarse grains that was observed in the Cadomin Fm is significantly reduced (eg: Figure 5 A (light green) or B4). The foreland trough, parallel to the Rocky Mountain front, substantially broadened and deepened during the 355 Cretaceous such that the stratigraphic preservation is much higher than in the underlying upper Jurassic interval. Based on the lack of fluvial propagation and high marine inundation, we are likely back in a phase 0 to 1 of basin filling. One could argue that a phase 1, early-filled basin (where $F$ approaches and even briefly exceeds 1) occurred during select pulses throughout the Cretaceous where the fluvial and delta facies propagated further (Figure 5B3) due to variations in uplift in time and space. But any grain size fining would likely still be subsidence-dominated on account of the low $F$ associated with active mountain 360 building. Mossop and Shetsen (1994) argues for high variability in the fluvial front and the extent of marine inundation events throughout the Cretaceous, which is likely linked to sea-level oscillations.

By the end of the Cretaceous (i.e., above the Bearpaw Formation in Figure 5 A (gold)), the marine seaway retreated due to the increased sediment flux and reduced subsidence resulting both from the effective erosion of the mountain belt that limited its height and thus the resulting flexure. However, the sediment flux from the mountain continued to be high and comparable 365 to the flexure-driven subsidence (Catuneanu, 2004). Thus, in the Uppermost Cretaceous and into the Tertiary (Figure 5A gold and B2), the basin entered phase 1-2, with the majority of the basin consistently above sea-level and fluvial (dark green-mainly sandstone) sediments transported over very long distances. Subsidence generally continued at a reduced pace during most of the Mesozoic and Cenozoic (see Figure 5A), most likely in response to the load of the sediment, and allowing for the preservation of strata (Mossop and Shetsen, 1994; Leckie and Smith, 1992) and thus, with $F$-values comprised between 1 and 10, indicating 370 a potential combination of subsidence and autogenic/transient control on fining.

Finally, during the Quaternary period the system entered what can be considered as phase 3 in the foreland basin evolution with strong bypass ($F > 10$) or erosion (figure 5A). Quaternary deposition is limited to the outer basin in Saskatchewan (Mossop and Shetsen, 1994). The modern topography of the Central-southern Alberta basin is high near the orogenic front (700-1000 m) with ample slope (promoting channel mobility and a thick active layer), which, with the small $\beta$ value, promotes 375 high autogenic influence on grain size fining. The modern fans developing at the surface of the Alberta basin have been described as dominated by internal processes (Campbell, 1998) and are therefore likely to be in a autogenic-dominated fining regime.

To examine a range of foreland basin evolution spanning from low to high bypass regimes ($F > 100$), we implemented a compressed (200 km), time-accelerated (20 Myr) basin evolution in our model simulations in which scaling $K$ and $U$ produced

orogen elevations from 1,000 to 10,000 m. While this approach offers an idealized overview of foreland basin evolution, it differs from the Alberta basin in a number of ways. Some of these include, changes in base level over time (Mossop and Shetsen, 1994), spatial and temporal variations in erodibility in the mountain area due to localized igneous and metamorphic lithologies (Monger and Price, 2002), and the N–S diachronous evolution of the collision (Mossop and Shetsen, 1994). Thus, simulation time (1 Myr) should not be directly equated to Alberta basin stratigraphy in terms of timing or thickness.

Keeping these complexities in mind, we can further estimate key model inputs ($G$, $K$, and flexure wavelength/$\alpha$) and parameters ($\beta$) for the Alberta Basin that have not already been as fully discussed as $F$. A global compilation by Guerit et al. (2019) suggests that systems tend to favor a slight transport limitation (laboratory median of $G = 0.7$), with common values of $G$ ranging from 0 to 3.1. The response time and $K$ are also difficult to constrain due to lateral growth of the mountain by terrane accretion of varied lithologies over the basin's long lifespan, exceeding 100 Myr (Mossop and Shetsen, 1994). Episodic
high uplift events, such as during the 100–99 Ma Viking Formation, show spatially variable sedimentary responses across the basin, with downstream sediment infilling in some areas but not others (Peper, 1993), indicating that the system's response time exceeds 1 Myr. The parameter $1/K$ is one of the dominant controls on response time (see equations in Braun (2022); Braun et al. (2015)), resulting in estimates of $K$ on the order of $10^{-7}$ to $10^{-8}$ $m^{1-2m}yr^{-1}$ with response time estimates of 10–100 Myr (based on the above descriptions). Elastic plate thickness in the Alberta foreland basin is estimated at 20–60 km,
yielding a flexural wavelength $\lambda$ of 274–739 km, assuming a lithosphere–asthenosphere density contrast ($\Delta p$) of 500 $\text{kg/m}^3$ (Flück et al., 2003). The adjacent craton, 60–150 km e-thickness, produces longer felxural wavelengths $> 739$ km (Flück et al., 2003). Early in basin evolution, limited sediment cover and a thick cratonic basement (Gordy and Frey, 1977) likely led to a rigid, long-wavelength system (Flemings and Jordan, 1989), suggesting a potential rigid-end $\alpha$ range (e.g. 0.2–0.5).

Evidence suggests a value of $\beta/\alpha < 1$ for the Western Canada foreland basin, more consistent with Fig. 3 than Fig. 4, due to
the rainshadow effect and the large downstream basin relative to the orogen. Recall the relationship $\beta/\alpha = \frac{v_M L_M}{v_B L_B}$ (Wild et al., 2024b) where $\beta$ can be derived from a ratio of orogen vs basin precipitation and length. Satellite imagery and paleogeographic maps (Mossop and Shetsen, 1994) show the Canadian Rockies as a series of ranges approximately 50–200 km wide, while the foreland basin spans on the order of 1000 km, bounded to the east by the Canadian Shield. Precipitation data (1971–2000) indicate 300–1500 mm/yr over the orogen and 300–500 mm/yr across the basin (Rivera and Calderhead, 2022) resulting in an
$\sim$2-3 times wetter orogen. Keep in mind, Cretaceous seaways (e.g., Fig. 5) may have increased basin moisture and reduced the rainshadow effect, while aridification may have enhanced precipitation gradients. Using modern averages, $\beta/\alpha$ can be estimated around 0.2 to 0.4.

### 5.2 Autogenic-dominated example: The Cadomin Formation

The early Cretaceous Cadomin Formation is described in areas as a megafan similar to the Kosi Fan, India, but with coarser
grain sizes (e.g. on average 1-3 cm gravels clasts that reach up to a maximum of 40cm (McLean, 1977)) and with some preserved evidence of autogenic dynamics (Leckie and Cheel, 1997). Amalgamation, frequent shifting, and bank cutting of channels have been described within the stratigaphy (Leckie and Cheel, 1997) match our stratigraphic results of an autogenic-dominated fining system under very high by-pass with reworking (Figure 2). However, recent work has identified terraces in the

upper basin that are preserved in the stratigraphy (Johnson and Dalrymple, 2019) and are indicative of aggrading and incision pulses (similar to our depositional divergence) during this period, that have been interpreted as related to climate cyclicity. This alternation of well sorted and poorly sorted episodes within the conglomerate has been described as a potential series of inter-flood episodes characterized by reworking and flood episodes without time for sorting (Johnson and Dalrymple, 2019). Banded argillic horizons within the soil deposits are also evidence of climate seasonality (Johnson and Dalrymple, 2019; Jablonski and Dalrymple, 2016) around this time that may impact the grain size signal.

We propose that the Cadomin Fm would be located in the lower right corner of the framework (Figure 1) in the region with a low $\beta$ and high $F$ where autogenic dynamics dominate the grain size record. We reiterate that the Cadomin Fm was deposited during a period of tectonic quiescence in the orogen collision and rebound in the basin (Leckie and Cheel, 1997; Johnson and Dalrymple, 2019) indicative of a high bypass $F$. We can also infer a generally low $\beta$ value for the Cadomin Fm from 1) the extent of the basin, i.e. the orogen existed several hundreds of kilometers from the shoreline (Smith and Leckie, 1990), which is likely to be larger than the size of the orogen resulting in a lower $\beta$ value; and 2) a generally semi-arid climate for western Canada around the Jurassic-early Cretaceous transition as indicated by the presence of silcrete in the paleosols (Leckie and Cheel, 1997; Leckie and Nadon, 1997; Ludvigson et al., 2015), which is the modern environments where steep fan to plain topography are observed (e.g. Mongolia). Furthermore, the system during the Cadomin clearly exhibits some reworking and variations in grain size (Johnson and Dalrymple, 2019) typical to the autogenic dominated regime described in Wild et al. (2024b) and shown in Figure 2A.

Seasonality could have also impacted climatic gradients between the newly formed orogen and basin. It is possible that the efficiency of the autogenic dominated grain size fining during the Cadomin Formation may have fluctuated under a cyclic climate, with periods of enhanced precipitation in the mountain causing variations in $\beta$. Periods of enhanced precipitation in the mountain would have caused an increase in $\beta$ and a decrease in surface slope, rugosity and magnitude of autogenic processes (Wild et al., 2024b). This, in turn, would have decreased the grain size fining and caused the propagation of the coarse grain front further into the basin, as observed by Johnson and Dalrymple (2019) and previously interpreted as periods of intense flooding. We have shown that our approach to modeling and interpreting foreland basin evolution, including identifying when such basins may record autogenic dynamics based on spatial gradients and sediment bypass, can be applied to real mountain belts. We suggest that this framework could be useful for analyzing other foreland basins at different stages of evolution.

## 6 Conclusions and Future Work

Our new GravelScape model can be used to help interpret grain size distributions within the stratigraphy of foreland basins. In particular, we have shown that:

- Grain size fining evolves in a foreland basin by transitioning from an under-filled regime (phase 0), to a subsidence-dominated regime (phase 1), a mixed (subsidence and transient/autogenic) regime (phase 2) and into a final bypass autogenic-dominated regime (phase 3).

- Grain size fining can tell us a lot about the basin dynamics and, in particular that (1) coarsening upwards with decreasing stratigraphic thickness (increasing $F$) is one of the clearest indications of a subsidence dominated basin (phase 1 in Figure 3; (2) a system that shows consistent fining with no coarsening upwards despite decreasing stratigraphic thickness (increasing $F$) is likely dominated by autogenic/transient fining or a combination of subsidence and autogenic fining (phase 2 and 3 in Figure 3); 3) a system with little to no fining under high bypass (Figure 4) is characterized by a long transport length of sediment (i.e. high $\beta$ or low $G$) and little autogenic dynamic induced grain size fining.

- Increasing depositional variance within the basin, reduced channel mobility, increased reworking/rugosity, the presence of unconformities, and increasing floodplain area all indicate a transition into a high bypass and autogenic-dominated grain size regime.

- During periods of subsidence-dominated grain size fining (phase 1 and 2), variations in sedimentary flux in response to external forcings are likely to cause spatial and temporal variations in grain size fining in response to a changing $F$ value (as demonstrated by Armitage et al. (2011); we have also show that in periods of autogenic-dominated grain size fining, climatic events can be recorded through the effect they will have on the value of $\beta$, which strongly influences the efficiency of autogenic-dominated grain size fining, as seen in the Cadomin Formation.

Further work is, however, needed to disentangle the relative importance of subsidence and autogenic processes in preserving or "shredding" the grain size information recorded in foreland basin stratigraphy.

## 7 Funding Declaration

This project received funding from the European Union's Horizon 2020 research and innovation program under the Marie Sklodowska-Curie grant agreement No 860383.

*Code availability.* Python Code for GravelScape will be made available upon paper acceptance. See the FastScape repository for further details: https://FastScape.readthedocs.io/en/latest/index.html

*Author contributions.*

. Amanda Wild: Conceptualization, formal analysis, investigation, methodology, software, validation, visualization, writing-original draft preparation, writing-review and editing

. Jean Braun: Supervision, resources, software, conceptualization, methodology, validation, visualization, writing-original draft preparation, and writing-review and editing

. Alexander Whittaker: Supervision, conceptualization, methodology, validation, and writing-review and editing

. Sebastien Castelltort: Supervision, conceptualization, and writing-review and editing

*Competing interests.* The authors declare that there are no competing interests.

*Acknowledgements.* The authors thank Benoit Bovy for general help with xarray-simlab and FastScape curation. We would also like to thank Charlotte Fillon for her comments during committee meetings and the earlier phases of this research. We would also like to thank scientists within the Earth Surface Process Modelling Section at the GFZ Potsdam and members of the S2S-Future Marie Curie ITN for their general feedback and discussions. Finally, we would like to thank Randy Enkin and members of the Pacific Geoscience Center for their general discussions and providing guided access to the Geological Atlas of the Western Canada Sedimentary Basin.

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
