# Peer review of "Grain size dynamics using a new planform model. Part 1: GravelScape description and validation"

_EGUsphere, 2024_

## Author Response (AR1)

**Reviewer Comments and Author Responses**

September 6, 2024

**1 RC1**

"This has been an extremely difficult paper to review. I found the large amount of information contained in the densely written 42 pages with 19 figures overwhelming. Five additional figures are in the appendix and 7 figures in the Supplementary Material. This is too much information for a single manuscript. The research summarized in the paper is thoroughly done and it has the potential to be a truly excellent contribution, but it needs to be presented in a more reader friendly way.

In the conclusion section the authors summarize main findings as follows:
- model formulation with the to incorporate a grain size fining model in a landscape evolution model;
- model testing and validation (not in the conclusion section, but must be written)
- model application to reproduce autogenic processes;
- analysis to show that grain size fining is controlled by a balance between external and internal forcing;
- applications to natural examples
identification of the stratigraphic signature
the case of a flexural foreland basin
- discussion on model applicability and limitation

This is clearly material for two solid, stand alone papers. The model, modeling challenges, limitations and verification can be presented in the first paper with some application to reproduce autogenic processes, if and how these processes depend on model parameters. The second paper can then clearly present main results and applications (control on fining, stratigraphic signatures....).

I hope this helps. "

- Thank you for stating the potential of contribution that warrants further expansion. Upon reflection on the reviewers' recommendations to split the manuscript, we have divided the original manuscript into three standalone papers, each addressing a unique aspect of our research on stratigraphic, fluvial grain size fining. The first paper introduces the new model, GravelScape, detailing its coupling method and validation against previous models, specifically focusing on the impact of altering two parameters ($F$ and $G$) that reflect subsidence and topography within the basin. This methods paper also identifies limitations in prior approaches, emphasizing the need to compute topography and deposition rates separately from subsidence. The second paper delves into the different factors the impact topography, internal variation, and subsequent grain size fining by varying further inputs ($G$, $K$, $F$, and $\beta$ that reflect fan extent, basin erodilbility, and more), and correlating between grain size fining and improved autogenic parameters. Paper 2 also proposes a novel framework for distinguishing whether basin dynamics are driven primarily by subsidence (mean deposition) or autogenic (variation) processes with natural examples to facilitate framework application. Out of all the papers, paper 2 underwent the most changes relative to the original manuscript although the core findings of the work remains the same (but improved in presentation and clarity). The third paper extends the model by incorporating flexure, exploring the evolution of foreland basins and stratigraphic profiles over time, with a case study comparison to the Alberta foreland basin. This paper builds on the previous two by considering the system as a dynamic, evolving entity rather than a series of static snapshots. While splitting the work into three standalone papers has led to some overlap in key model equations necessary for understanding the results, we believe the ample, quality content, novel findings, and new methods implemented throughout the works justify the division. A three paper approach allows us to provide further clarity and enhances the accessibility of the material.

**2    RC2**

" Synopsis

In this manuscript, the authors have undertaken an ambitious modeling exercise to answer an important geoscience problem. They base their modeling framework and plans on a few key tenets. First, they hold that sediment grain size is a primary observable parameter in the stratigraphic record. Thus, earth system models that produce stratigraphic volumes should be formulated to spit out grain size information as a primary output. Second, they assert that mass exchange in two dimensions is the main method by which autogenic noise arises in sedimentary environments. Third, coupling between sediment loading and flexure in the lithosphere should lead to a predictable life cycle of sedimentary basins that produce consistent grain size trends.

These main tenants come together in a set of equations that allow a two-dimensional morphodynamic model to efficiently and parsimoniously balance external (allogenic) and internal (autogenic) dynamics to produce self-consistent grain size trends in a stratigraphic volume. The authors use this modeling framework to examine when the grain size trends in sedimentary systems are dominated by internal dynamics versus external dynamics. They do so via a set of modeling experiments, and a comparison to the Alberta foreland basin.

- This is a good summary of the different components of the work presented. However, the results of the coupling of flexure and sediment load had slightly different intended connotations. We recognize that the life cycle of basins is not always so straight forward or predictable. We wanted to highlight that evolution of a filling basin and final state of grain size fining under bypass may differ between basin set-ups depending on if autogenic fining occurs. Thus, the same life cycle with different orogen-basin spatial configurations and topography can produce differing grain size trends through basin evolution. This deviates from previous assumptions that under the same flux and subsidence state of basin evolution, the same long-term fining trend should be observed across basin set-ups.

Overall comments

I agree strongly with the first reviewer. This manuscript was very challenging to review and to understand. I hope that my synopsis above summarizes the paper's goals and main ideas correctly. I also agree that I think that this piece of work could be quite impactful, and I think that the intellectual effort the authors have undertaken is very important. As far as I can tell, their modeling framework could represent a substantial step forward in our ability to model and understand the handoff between autogenic and allogenic forcing in sedimentary basins. "

- We greatly appreciate the recognition of the potential (with structure improved) impactful, ambitious, and important quality of the work that warrants clarification and expansion (into multiple works as suggested by RC1) opposed to simply cutting out content.

"As written and presented, I am not sure that this paper will have the impact that the authors intend, as I suspect it would not be widely read. My own attempt to connect with the meat of the paper is illustrative: I had a lot of questions about the specific ways that this model treats the internal dynamics of sedimentary systems. For instance, the authors assert in a few parts of the manuscript that fluctuations in the boundary conditions in their model are not shredded by the internal dynamics of the river system, and that information is recoverable (e.g. line 888). This is quite an interesting and exciting statement, but I found myself puzzling over it, because it seems to imply that in some parts of the parameter space, this model behaves like a linear transformer (that is, it adds random noise, but the signal remains recoverable). I struggled to understand why this might be the case, because from everything I know about sedimentary systems, if this model is going to capture those internal dynamics and feedbacks, it should produce specific kinds or colors of noise (mass or grain size fluctuations)."

- We more explicitly explain the autogenic dynamics within the model as physical phenomena and not numerical through including the spatial and temporal validation in paper 1 and through more robust correlations using a few key parameters (deposition, topography variation, and channel dynamics) explained in greater detail in paper 2. More specifically, we removed some of the autogenic parameters (e.g. depositional waves, channel mobility, and local minima) and instead focus primarily on depositional divergence and rugosity that had the strongest correlations to the autogenic grain size dynamics. We then relate the greater physical autogenic dynamics to the grain size fining through these more clearly correlated and defined parameters. We added a discussion paragraph on the physical nature of autogenic dynamics within the paper 2 " Links between $\Delta D$ and $\dot{d}_v$, $\eta$, and $S$" section.

"However, I was unable to really glean some of these big-picture aspects of the model, because the presentation quality is lacking. It is not just a matter of the material being overwhelming like the first reviewer mentioned. "

- To improve clarity and reduce density, we spent significant time improving the figures and writing. We decided to focus on only two parameters (one indicating subsidence changes and the other topography) within the first paper. This resulted in remaking the previous validation figure 12 (with $\beta$, $F$, and $G$) to only show changing $F$ and $G$ (paper 1 figure 5). In paper 2, we present the grain size results (paper 2 figures 2 3, and 4) of each parameters. Where previously, we only showed a subset of changing $G$, $F$, and $\beta$ (previous preprint figure 16). We also removed less well correlated autogenic parameters and presented only the strongest correlation (paper 2 figure 7) plots. Finally, we improved the general framework (paper 2 Figure 8), that was attempted in original figure 16, by explicitly defining an on average autogenically vs subsidence dominated regime on the figure.

"I'll highlight a couple of specific things about the communication that are unsuccessful, and offer a suggestion or two for each. First I have a suggestion for changes to the overall structure, then I have some ideas about how you could compose your sentences and paragraphs more clearly, and then I have a suggestion for how to make your terminology and other context information more approachable.

In broad structural terms, I agree with the first reviewer that the paper would benefit from being split in two parts."

- Upon reflection of the reviewers comments, we agree that splitting the work is necessary to improve accessibility and comprehension of the content. Please see the above blue comment to RC1 detailing how we split the work.

"Both the model description/experiment and the Alberta case study are dense and unreadable, and both would actually benefit from some expansion. The model description relies heavily on abbreviations and jargon that I suspect is common shorthand in the research team working on this project. I think the authors could make use of a standalone paper to explain each component of their model in plain language first, with lots of subsections and concrete examples. This basic idea also applies to the Alberta case study. "

- In the first paper, we spend more time to introduce the model and the key equations. This gives the space to introduce more parameters that impact topography ($\beta$) in paper 2 that also build on the findings from paper 1. In paper 2, we added more examples to the (Part 2-figure 8) framework. In paper 3 (alberta basin), we expanded the introduction and methods regarding foreland basin evolution and improved the paragraph structure and writing flow. We also believe that splitting the work, and any necessary repetition, substantially improved the accessibility of the content.

"Within each section or subsection, I found it very hard to relate individual paragraphs back to the larger purpose of the manuscript. Part of this is because sometimes the paragraphs lacked clear topic sentences or they encompassed several different ideas. The outcome is that longer passages started to read something like a stream of consciousness, and I would have to go back and reread the passage many times to get the meaning. The subject matter that you're trying to communicate in this manuscript is quite complicated, and multi-dimensional. Everything depends on everything else. You—the authors—have spent a long time thinking about and working with these equations and these model outputs. The reader though, is coming to this for the first time, and I had a really hard time holding all of the connections in my head simultaneously. "

- To address this lack of clarity, we both removed unnecessary content, shortened nearly all paragraphs, and only add content when it improved the flow or added clarity. For example, we decided to remove the autogenic recovery time parameter for a future work in order to only focused on the correlations that most impacted the grain size dynamics.

"I think you can make this easier for the reader by breaking up some of your model description and results into smaller self-contained chunks were you describe a single parameter and the influence it has on its own. I think that you could accomplish this through the use of extensive subsections within the sections you're using now. "

- Breaking up the parameter results over different papers and implementing more sections allowed us to have the space to more explicitly break down the impact of the different parameters. Then we only need to refresh the reader of the past results and build upon them. In paper 2, we often re-summarize (eg: start of the discussion) all model parameters and their impacts in plain language as well as produced figures that more clearly show the impacts of each model parameter.

"Concrete examples also help a great deal."

- We added images to the general framework (paper 2 Figure 8), that refer directly to concrete natural environmental examples where we suggest that their is a higher likelihood for autogenic vs subsidence dominated grain size fining within the system.

"The final thing that I will point out as a major way that you can improve communication is to simplify and streamline your terminology and to embrace restating key ideas in plain language throughout the manuscript. Once again you—the authors—have been working with these equations and parameters for months (if not years). We, the reader, have our own relationships with F, G, K, $\beta$, and $\mu$. I am perfectly willing to give up my relationship with $\beta$ temporarily and reassign it to something else while I am sitting down to read your manuscript, but it's hard to do that for 30 different constants and terms. You have helpfully provided a table for this, but even so, it's quite a lot to ask. The cost of relying on so many new terms is that the reading experience stops being frictionless. By the time that I got to page 25, I had forgotten the difference between F and G, and I was not so sure what $\beta$ referred to. In order me to understand what it is that you were saying, I had to flip back and forth to table 1. "

- Splitting the work inherently entailed that we needed to restate a few key parameters and equations. We also made sure to restate key parameters at the start of each new, major section heading (eg: between discussion and results) and kept this in mind when considering the flow of our work.

"Even then, I'm not sure if I really do understand what "Depositional dimensionless parameter" means."

- We spent more time defining this parameter within paper 2 and explaining its significance with more clarity.

"I think you can easily remedy this by adopting a short, crisp half-sentence that describes each parameter, and sprinkling that phrase throughout the manuscript. For every time that you've gone, say, two pages without restating the meaning of a parameter, just insert the phrase so that the reader is reoriented. There's actually a really good example of this near line 445. You describe the "grain size at a instantaneous time step (Dx)" and then just a page later say "deposited grain size, Dx". By restating in words what it is these parameters refer to, you can sign post for the reader so that they don't get lost."

- We have added more clarification of parameters throughout the manuscripts.

"Similarly, in your introduction you tend to refer to and engage with a large body of literature mainly through reference, and then later use previous author's names as a shorthand for the modeling framework that they developed. While this is customary, I think that there is a better way. I think you can make it a lot easier for your reader by giving these existing modeling frameworks short descriptive names. Thus, instead of saying "Fedele and Paola (2007)'s equations", you could say "1D self-similar grainsize sorting model" or something like that. While of course you should make clear the attribution, for somebody who has not been following the twists and turns of this body of literature, descriptive names will be a more helpful shorthand."

- We went into greater detail on the past approaches and their deviation from our work within paper 1. We then more explicitly define an equation for autogenic grain size fining or grain size deviation in paper 2 where we move away from referring to parameters solely based on past literature.

"Anyway, I think after substantial revision, or maybe reconsideration as two separate manuscripts, this could be a really valuable contribution. I look forward to learning more about it, and thinking about how this model and theoretical framework might apply to my own work."

- The key findings presented in the original manuscript remain constant after the revisions, but the structure, figures, and expansion of key concepts has been greatly improved. The changes described aim to greatly improve the clarity through substantial restructuring and improving the presentation of the correlations of grain size with the autogenic dynamics physically observed within the model. We hope that you agree and would like to thank the reviewers for their constructive comments.

---

## Author Response (AR2)

**Reviewer Comments and Author Responses**

May 26, 2025

**1 RC1**

"These series of papers by Wild and coauthors introduce GravelScape, a novel model for simulating sediment grain size dynamics in sedimentary basins. The model couples a landscape evolution model with a self-similar grain size fining model, allowing for the investigation of how both external factors, such as tectonics and climate, and internal processes, such as channel avulsions, influence sediment deposition. One study focuses on the model's validation and general behavior. Another explores the relative importance of autogenic processes versus external forcing and the final study examines the evolution and the stratigraphic record of flexural foreland basins using the model. The GravelScape model offers advancements over previous models, particularly in its ability to simulate grain size fining in multiple dimensions and account for autogenic processes. Overall, the manuscripts are well written, results are nicely demonstrated and discussed. I recommend it to be published after addressing a few minor comments. "

- Thank you for taking the time to read through, summarize, and review the articles.

"Regarding model validation, it is based on a comparison with Duller et al. (2010)'s single-channel model. However, GravelScape is a multi-dimensional model designed for complex sedimentary systems. The manuscript should provide more justification on that."

- Within the validation of Paper 1, we have separated Figure 4 into more sub-plots to more clearly show the two grain size validations that were made comparing the Duller et al. (2010) solutions with our GravelScape model. First we compared the Duller et al. (2010) results to our computation limiting the model to a single channel (2 cell wide) solution to validate our integration of the Fedele and Paola (2007) equation (subplots c and d in Figure 4). Then we include multiple-channel dynamics to the model and compare this to the single channel solution (subplots e and f in Figure 4).

- In the discussion of Paper 1, we added text (see the following bullets) to the discussion explicitly noting where the application of a single channel vs multi-channel grain size models can be justified:

- "...The results from our validation (Figure 4) indicate that single-channel solutions are most applicable under more uniform flow and early basin filling states (low $F$) where our GravelScape multi-channel solution showed little deviation. Such is likely the case in the Pobla Basin, Montsor Formation where Duller et al. (2010) applied the Fedele and Paola (2007) grain size fining model assuming subsidence is equal to deposition rate. The Montsor formation is described as a progradation of extensive alluvial fans filling a wedge top basin during a period of intense thrust activity and subsidence in the southern Pyrenees Axial zone (Duller et al., 2010). However, a more complex, multi-channel, lateral model that decouples deposition from subsidence rate is justified, if not necessary, to simulate grain size in systems in high bypass (high $F$), with steeper topography, or with more diverse geomorphology and stratigraphy (e.g. variations in channel dynamics, fan, and floodplain)..."

- "Additional factors (e.g. slope) that influence grain size fining in alluvial fans aside from subsidence and mean deposition rate, have long been debated in the literature (Stock et al., 2008)...D'Arcy et al. (2017)'s correction factor is one example that justifies the need for the multi-channel grain size model that can predict lateral depositional variations in real time, especially in systems where grain size fining cannot easily be explained through subsidence alone."

"The model assumes downstream deposition is the primary control on grain size fining, how would the pre-existing topography affect the grain size distribution? "

- We added a line to the discussion of Manuscript 1 to address the impact of pre-existing topography: "...This topographic influence on grain size implies that factors that can increase topography, such as initial topography or certain basin geometries, could impact the grain size fining when multi-channel solutions are considered. This warrants further study that we present in Wild et al. (2024) along with further applications of the multi-channel model....".

- We then expand on this in the appendix of Paper 2, where we discuss the impact of slope on grain size in greater detail in an alternative framework, stating: "Within the main text, we prioritized $\beta$ configurations as one approach to inducing higher slopes and more autogenically dominated conditions, due to $\beta$'s measurability at the landscape scale. However, our results also showed how transient conditions (lower $K$) and higher $G$ can increase slope and autogenic dynamics. With limited subsidence, any initial topography present within the basin could perpetuate increased slope, rugosity, and autogenic fining conditions. However, under high subsidence conditions, impacts of initial topography in a basin would likely be rapidly buried, leading to flatter slopes, low across basin topographic variability, and subsidence dominated fining conditions. There are many more scenarios that could impact slope and subsequent autogenic fining conditions that warrant further study."

- Within Paper 3, we also added a sentence addressing initial topography within the introduction: "Initial topography can impact the timing of basin infilling and, when initial conditions raise elevation, promote more continental opposed to marine dominated infilling conditions (Gérard et al., 2023)."; and methods: "We imposed a slight initial topography in the model to promote continental conditions in the foreland basin where we can compute grain size (see Gérard et al. (2023) for a description of how initial topography impacts foreland basin evolution)." of the paper clarifying the impact of initial topography within the context of foreland basin evolution modeling.

"Minor technique correction Line 4: change Fedele and Paola (2007) to (Fedele and Paola, 2007) "

- We have corrected the citation error from Fedele and Paola (2007) to (Fedele and Paola, 2007) and corrected other minor typos and errors within the text.

**2 RC2**

"In this contribution, the authors present a significant advancement in landscape and stratigraphy modeling by developing a new framework that integrates a planform grain size model with a landscape evolution model. The ability to simulate downstream grain size fining across multiple channels in a landscape offers valuable insights into the mechanisms (autogenic versus allogenic) driving the development of grain size trends in the stratigraphic record. The authors have taken great care in restructuring the original submission into three standalone yet closely related manuscripts. I found the revision to be well-organized, with the significance of the work clearly articulated."

- We thank the reviewer for their positive comments and for taking the time to read through the articles.

" I believe the manuscript meets the standards for publication, and I offer two major comments for the authors' consideration to further improve clarity: 1. The second manuscript specifically addresses disentangling the effects of autogenic processes and external controls on model outcomes. Although the introduction mentions autogenic processes, the only one explicitly modeled appears to be avulsion (line 51), if I have interpreted the manuscript correctly. Given that autogenic processes are central to this component of the study, I suggest adding a brief paragraph elaborating on the specific autogenic processes included in the model. For instance, what mechanisms of avulsion are being represented in the simulations? "

- We added a short section titled 'Modeled Autogenic Dynamics' where we specify the scale and specific autogenic dynamics included within the numerical model. In this section, we not only provide a clearer description of what is included in the model, but also refer to the work of Hajek and Straub (2017) where they described autogenic dynamics and the different scales at which they operate. In this way, we can describe our model within the context of work on real-world autogenic dynamics without needing to re-define already well-described processes/phenomena within the literature.

"2. In the third manuscript, the authors examine the Alberta Basin and interpret the stratigraphic and grain size trends in terms of their development conditions. If I understand the methods correctly, these interpretations are based on qualitative interpretation of key modeling parameters, such as F and $\beta$, derived from model outcomes and the basin's stratigraphic architecture and paleogeography. While this is not an inversion study, providing some back-of-the-envelope, quantitative constraints on these parameters would strengthen the interpretations and add credibility to the conclusions. "

- Within Manuscript 3, we added three paragraphs at the end of the "General Evolution" section addressing further quantitative constraints on model parameters within the context of the Alberta Basin.

- While addressing the reviewer's comment, we noticed a minor mistake between some of the $K$ and precipitation values (impacting our computations of $\beta$) in the input table of Paper 3 and those used in some of the model simulations. Also, since multiple modeling approaches (e.g. with and without flexure) were applied in paper 3 that used different input fields, we realized that the original table of inputs was a bit confusing or misleading. To address this, we separated the table of inputs into two columns adding more clarity regarding the two different (with and without flexure) modeling approaches used in Paper 3 and corrected the $\beta$ values in the table, text, and figures. These changes did not alter the interpretations or $F$ evolution (e.g., autogenic vs. subsidence-dominated) within the framework or text.

- Finally, without changing the interpretation or simulations used, we replotted the high $\beta$ foreland evolution (Figure 4 in Manuscript 3) in the exact same manner (e.g., using the same color scheme and same panel setup) as the low $\beta$ simulation in order to improve the comparison between the two basin simulations for the reader.

"Minor comments: 1. Manuscript 1, line 8-11 "we also show...": This sentence is too long and very hard to read. "

"We show that, when multi-channel dynamics (i.e. avulsions) are prevented, by reducing the planform model to a single downstream dimension, our new model can reproduce results obtained by other methods that assume that fining is controlled by subsidence only. We demonstrate that including across-basin (two-dimensional) effects can lead to deviations from previous subsidence predictions for grain size fining. The magnitude of these deviations correlates with the extent of sediment bypass and the configuration of surface topography, both of which influence the amplitude of across-basin variability within the sedimentary system."

"2. line 182 "by the flux of..": Do you mean "to the flux of..." "

- Yes, we changed line 182 to state: "to the flux of...".

"3. Figure 4 is hard to read, especially subplot c and d. Consider having more subplots or break this up into two figures. For example, subplot c and d showing multiple comparisons in one figure that are hard to follow "

- We added additional subplots to Figure 4 in Manuscript 1. More specifically, we split the original Figure 4 subplots c and d into subplots c,d,e, and f. We also removed the F=1000 scenario from all subplots in order to reduce the number of lines within the plots to improve clarity.

"4. Manuscript 2, line 120: could you provide some end member examples of this shape parameter. For example, something like "high catchment precipitation and low basin length will result in high beta value" or "high beta value typically will indicate small/large basins". The point is that, having some end member examples of low and high beta value, and how it is directly linked to the topography/size of the basin/catchment, will help reader to visualize the parameter, which will make reading the following discussions much easier. "

- We added three lines to Manuscript 2 summarizing the parameter end member scenarios: "In short,$\beta$ is a measure of the difference in area (extent) and precipitation rate between the orogen catchment and the sedimentary basin. Combinations of high precipitation and drainage area in the orogen with low basin length and basin aridity result in high, orogen dominant, $\beta$ values. Inversely, large basin areas, especially with higher precipitation relative to the orogen, result in low, basin dominant, $\beta$ values."

"5. line 125: shouldn't this be "alpha/LB"?"

- We corrected the sentence: "$\beta$ is in fact the ratio of the length/size of the fan, $\nu_M L_M/\nu_B$ to the size of the subsidence function, $\alpha L_B$." to read: "$\beta$ is in fact the ratio of the length/size of the fan, $\nu_M L_M/\nu_B$ to the size of the subsidence function, $\alpha/L_B$."

- We also tried to provide more clarification and consistency in the handling of $\beta$ throughout the text in all manuscripts.

"6. line 216-217 "To quantify this difference...": this sentence is hard to read and understand, consider break it up."

- We have rewritten the sentence: "To quantify this difference, we define a parameter that is sensitive to the amplitude of local deposition rate events independent of the deposition rate caused by the imposed basement subsidence." as: "To quantify this difference, we define a parameter sensitive to local deposition rate fluctuations. We explicitly remove the background mean deposition rate, induced by basement subsidence, to isolate the amplitude of depositional variability."

"7. Manuscript #3, line 24 "accommodation space": the word "accommodation" means "available space", so technically you can just say accommodation without "space". This term has been used multiple times, please consider changing it. "

- Although both the terms "accommodation" and "accommodation space" are used in the literature, we removed the "space" from line 24 and other areas of the text to reduce repetition and redundancy.